# Heterogeneous Neuronal and Synaptic Dynamics for Spike-Efficient Unsupervised Learning: Theory and Design Principles

**Biswadeep Chakraborty & Saibal Mukhopadhyay**
Department of Electrical and Computer Engineering
Georgia Institute of Technology
`{biswadeep,smukhopahyay6}@gatech.edu`

## Abstract

This paper shows that the heterogeneity in neuronal and synaptic dynamics reduces the spiking activity of a Recurrent Spiking Neural Network (RSNN) while improving prediction performance, enabling spike-efficient (unsupervised) learning. We analytically show that the diversity in neurons' integration/relaxation dynamics improves an RSNN's ability to learn more distinct input patterns (higher memory capacity), leading to improved classification and prediction performance. We further prove that heterogeneous Spike-Timing-Dependent-Plasticity (STDP) dynamics of synapses reduce spiking activity but preserve memory capacity. The analytical results motivate Heterogeneous RSNN design using Bayesian optimization to determine heterogeneity in neurons and synapses to improve $\mathcal{E}$, defined as the ratio of spiking activity and memory capacity. The empirical results on time series classification and prediction tasks show that optimized HRSNN increases performance and reduces spiking activity compared to a homogeneous RSNN.

## 1 Introduction

Spiking neural networks (SNNs) (Ponulak & Kasinski, 2011) use unsupervised bio-inspired neurons and synaptic connections, trainable with either biological learning rules such as spike-timing-dependent plasticity (STDP) (Gerstner & Kistler, 2002) or supervised statistical learning algorithms such as surrogate gradient (Neftci et al., 2019). Empirical results on standard SNNs also show good performance for various tasks, including spatiotemporal data classification, (Lee et al., 2017; Khoei et al., 2020), sequence-to-sequence mapping (Zhang & Li, 2020), object detection (Chakraborty et al., 2021; Kim et al., 2020), and universal function approximation (Gelenbe et al., 1999; Iannella & Back, 2001). An important motivation for the application of SNN in machine learning (ML) is the sparsity in the firing (activation) of the neurons, which reduces energy dissipation during inference (Wu et al., 2019). Many prior works have empirically shown that SNN has lower firing activity than artificial neural networks and can improve energy efficiency (Kim et al., 2022; Srinivasan & Roy, 2019). However, there are very few analytical studies on *how to reduce the spiking activity of an SNN while maintaining its learning performance*. Understanding and optimizing the relations between spiking activity and performance will be key to designing energy-efficient SNNs for complex ML tasks.

In this paper, we derive analytical results and present design principles from optimizing the spiking activity of a recurrent SNN (RSNN) while maintaining prediction performance. Most SNN research in ML considers a simplified network model with a homogeneous population of neurons and synapses (homogeneous RSNN (MRSNN)) where all neurons have uniform integration/relaxation dynamics, and all synapses use the same long-term potentiation (LTP) and long-term depression (LTD) dynamics in STDP learning rules. On the contrary, neurobiological studies have shown that a brain has a wide variety of neurons and synapses with varying firing and plasticity dynamics, respectively (Destexhe & Marder, 2004; Gouwens et al., 2019; Hansel et al., 1995; Prescott et al., 2008). We show that *optimizing neuronal and synaptic heterogeneity will be key to simultaneously reducing spiking activity while improving performance*.

Table 1: Notations Table

| Notation | Meaning | Notation | Meaning | Notation | Meaning | Notation | Meaning |
|---|---|---|---|---|---|---|---|
| HRSNN | Heterogeneous Recurrent Spiking Neural Network | $S_i(t)$ | Spike train from neuron i at time t | $z_j^t$ | Spike indicator Function | $\mathbf{r}(t)$ | States of RSNN |
| MRSNN | Homogeneous Recurrent Spiking Neural Network | $\Delta w$ | Synaptic Weight Update | $v_{th}$ | Threshold voltage | $\Sigma$ | Covariance Matrix of $\mathbf{r}(t)$ |
| $\tau_m$ | Membrane Time Constant | $\tau_\pm$ | Synaptic Time Constants | $\mathcal{C}$ | Memory Capacity | $x_i$ | Model Input |
| $v_i(t)$ | Membrane Potential | $\eta_\pm$ | Scaling Functions | $y_\tau$ | Model Output with delay $\tau$ | $\mathcal{E}$ | Spike Efficiency |
| $v_{rest}$ | Resting potential | $\alpha$ | Constant Decay Factor | $\mathcal{R}$ | Recurrent Layer in RSNN | $S_R, S_M$ | Total Spike Count HRSNN (R) , MRSNN (M) |
| $\tilde{S}_R, \tilde{S}_M$ | Avg. Spike Count per neuron of HRSNN (R), MRSNN (M) | $W_{ji}$ | Synapse weight from neuron $i$ to neuron $j$ | $\mathcal{H}$ | Heterogeneity in Neuronal Parameters | $\Phi_R, \Phi_M$ | Spike frequency of HRSNN (R), MRSNN (M) |

We define the spike efficiency $\mathcal{E}$ of an RSNN as the ratio of its memory capacity $\mathcal{C}$ and average spiking activity $\tilde{S}$. Given a fixed number of neurons and synapses, a higher $\mathcal{C}$ implies a network can learn more patterns and hence, perform better in classification or prediction tasks (Aceituno et al., 2020; Goldmann et al., 2020); a lower spiking rate implies that a network is less active, and hence, will consume less energy while making inferences (Sorbaro et al., 2020; Rathi et al., 2021). We analytically show that a **H**eterogeneous **R**ecurrent SNN (HRSNN) model leads to a more spike-efficient learning architecture by reducing spiking activity while improving $\mathcal{C}$ (i.e., performance) of the learning models. In particular, we make the following contributions to the theoretical understanding of an HRSNN.

- We prove that for a finite number of neurons, models with heterogeneity among the neuronal dynamics has higher memory capacity $\mathcal{C}$.

- We prove that heterogeneity in the synaptic dynamics reduces the spiking activity of neurons while maintaining $\mathcal{C}$. Hence, a model with heterogeneous synaptic dynamics has a lesser firing rate than a model with homogeneous synaptic dynamics.

- We connect the preceding results to prove that simultaneously using heterogeneity in neurons and synapses, as in an HRSNN, improves the spike efficiency of a network.

We empirically characterize HRSNN considering the tasks of (a) classifying time series ( Spoken Heidelberg Digits (SHD)) and (b) predicting the evolution of a dynamical system (a modified chaotic Lorenz system). The theoretical results are used to develop an HRSNN architecture where a modified Bayesian Optimization (BO) is used to determine the optimal distribution of neuron and synaptic parameters to maximize $\mathcal{E}$. HRSNN exhibits a better performance (higher classification accuracy and lower NRMSE loss) with a lesser average spike count $\tilde{S}$ than MRSNN.

**Related Works** Inspired by the biological observations, recent empirical studies showed potential for improving SNN performance with heterogeneous neuron dynamics(Perez-Nieves et al., 2021; Chakraborty & Mukhopadhyay, 2023). However, there is a lack of theoretical understanding of why heterogeneity improves SNN performance, which is critical for optimizing SNNs for complex tasks. She et al. (2022) have analytically studied the universal sequence approximation capabilities of a feed-forward network of neurons with varying dynamics. However, they did not consider heterogeneity in plasticity dynamics, and the results are applicable only for a feed-forward SNN and do not extend to recurrent SNNs (RSNN). The recurrence is not only a fundamental component of a biological brain (Soures & Kudithipudi, 2019), but as a machine learning (ML) model, RSNN also shows good performance in modeling spatiotemporal and nonlinear dynamics (Pyle & Rosenbaum, 2017; Gilra & Gerstner, 2017). Hence, it is critical to understand whether heterogeneity can improve learning in an RSNN. To the best of our knowledge, this is the first work that analytically studies the impact of heterogeneity in synaptic and neuronal dynamics in an RSNN. This work shows that only using neuronal heterogeneity improves performance and does not impact spiking activity. The number of spikes required for the computation increases exponentially with the number of neurons. Therefore, simultaneously analyzing and optimizing neuronal and synaptic heterogeneity, as demonstrated in this work, is critical to design an energy-efficient recurrent SNN.

## 2    PRELIMINARIES AND DEFINITIONS

We now define the key terms used in the paper. Table 1 summarizes the key notations used in this paper. Figure 1 shows the general structure of the HRSNN model with heterogeneity in both the LIF neurons and the STDP dynamics. It is to be noted here that there are a few assumptions we use for the rest of the paper: Firstly, the heterogeneous network hyperparameters are estimated before the training and inference. The hyperparameters are frozen after estimation and do not change during the model evaluation. Secondly, this paper introduces neuronal and synaptic dynamics heterogeneity by using a distribution of specific parameters. However, other parameters can also be chosen, which might lead to more interesting/better performance or characteristics. We assume a mean-field model where the synaptic weights converge for the analytical proofs. In addition, it must be noted that LIF neurons have been shown to demonstrate different states Brunel (2000). Hence, for the analytical study of the network, we use the mean-field theory to analyze the collective behavior of a dynamical system comprising many interacting particles.

**Heterogeneous LIF Neurons** We use the Leaky Integrate and Fire (LIF) neuron model in all our simulations. In this model, the membrane potential of the $i$-th neuron $u_i(t)$ varies over time as:

$$\tau_m \frac{dv_i(t)}{dt} = -\left(v_i(t) - v_{rest}\right) + I_i(t) \qquad (1)$$

where $\tau_{\mathrm{m}}$ is the membrane time constant, $v_{rest}$ is the resting potential and $I_{\mathrm{i}}$ is the input current. When the membrane potential reaches the threshold value $v_{\mathrm{th}}$ a spike is emitted, $v_i(t)$ resets to the reset potential $v_{\mathrm{r}}$ and then enters a refractory period where the neuron cannot spike. Spikes emitted by the $j$ th neuron at a finite set of times $\{t_j\}$ can be formalized as a spike train $S_i(t) = \sum \delta\left(t - t_i\right)$.

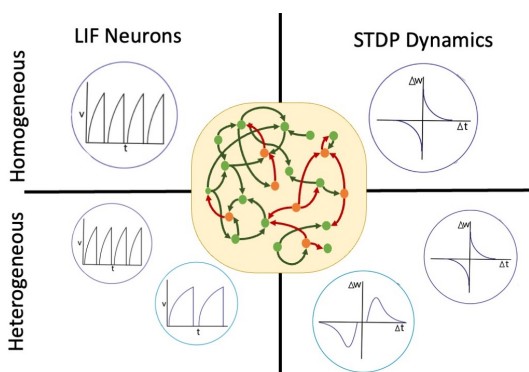

Figure 1: Concept of HRSNN with variable Neuronal and Synaptic Dynamics

Let the recurrent layer of an RSNN be $\mathcal{R}$. We incorporate heterogeneity in the LIF neurons by using different membrane time constants $\tau_{m,i}$ and threshold voltages $v_{th,i}$ for each LIF neuron $i$ in $\mathcal{R}$. This gives a distribution of time constants and threshold voltages of the LIF neurons in $\mathcal{R}$.

**Heterogeneous STDP:**

The STDP rule for updating a synaptic weight ($\Delta w$) is defined by Pool & Mato (2011):

$$\Delta w(\Delta t) = \begin{cases} A_+(w)e^{-\frac{|\Delta t|}{\tau_+}} & \text{if } \Delta t \geq 0 \\ -A_-(w)e^{-\frac{|\Delta t|}{\tau_-}} & \text{if } \Delta t < 0 \end{cases} \qquad s.t., A_+(w) = \eta_+\left(w_{\max} - w\right), A_-(w) = \eta_-\left(w - w_{\min}\right)$$

(2)

where $\Delta t = t_{\mathrm{post}} - t_{\mathrm{pre}}$ is the time difference between the post-synaptic spike and the pre-synaptic one, with synaptic time-constant $\tau_\pm$. In heterogeneous STDP, we use an ensemble of values from a distribution for $\tau_\pm$ and the scaling functions $\eta_\pm$.

**Heterogeneity:**

*We define heterogeneity as a measure of the variability of the hyperparameters in an RSNN that gives rise to an ensemble of neuronal dynamics.*

Entropy is used to measure population diversity. Assuming that the random variable for the hyperparameters $X$ follows a multivariate Gaussian Distribution ($X \sim \mathcal{N}(\mu, \Sigma)$), then the differential entropy of $x$ on the multivariate Gaussian distribution, is $\mathcal{H}(x) = \frac{n}{2}\ln(2\pi) + \frac{1}{2}\ln|\Sigma| + \frac{n}{2}$. Now, if we take any density function $q(\mathrm{x})$ that satisfies $\int q(\mathrm{x})x_i x_j d\mathrm{x} = \Sigma_{ij}$ and $p = \mathcal{N}(0, \Sigma)$, then $\mathcal{H}(q) \leq \mathcal{H}(p)$. (Proof in Suppl. Sec. A) The Gaussian distribution maximizes the entropy for a given covariance. Hence, the log-determinant of the covariance matrix bounds entropy. Thus, for the rest of the paper, we use the determinant of the covariance matrix to measure the heterogeneity of the network.

**Memory Capacity:** *Given an input signal x(t), the memory capacity $\mathcal{C}$ of a trained RSNN model is defined as a measure for the ability of the model to store and recall previous inputs fed into the network* (Jaeger, 2001; Jaeger et al., 2001).

In this paper, we use $\mathcal{C}$ as a measure of the performance of the model, which is based on the network's ability to retrieve past information (for various delays) from the reservoir using the linear combinations of reservoir unit activations observed at the output. Intuitively, HRSNN can be interpreted as a set of coupled filters that extract features from the input signal. The final readout selects the right combination of those features for classification or prediction. First, the $\tau$-delay $\mathcal{C}$ measures the performance of the RC for the task of reconstructing the delayed version of model input $x(t)$ at delay $\tau$ (i.e., $x(t-\tau)$ ) and is defined as the squared correlation coefficient between the desired output ( $\tau$-time-step delayed input signal, $x(t-\tau)$) and the observed network output $y_\tau(t)$, given as:

$$\mathcal{C} = \lim_{\tau_{\max}\to\infty} \sum_{\tau=1}^{\tau_{\max}} \mathcal{C}(\tau) = \lim_{\tau_{\max}\to\infty} \sum_{\tau=1}^{\tau_{\max}} \frac{\mathrm{Cov}^2\left(x(t-\tau), y_\tau(t)\right)}{\mathrm{Var}(x(t))\,\mathrm{Var}\left(y_\tau(t)\right)}, \tau \in \mathbb{N}, \tag{3}$$

where $\mathrm{Cov}(\cdot)$ and $\mathrm{Var}(\cdot)$ denote the covariance function and variance function, respectively. The $y_\tau(t)$ is the model output in this reconstruction task. $\mathcal{C}$ measures the ability of RSNN to reconstruct precisely the past information of the model input. Thus, increasing $\mathcal{C}$ indicates the network is capable of learning a greater number of past input patterns, which in turn, helps in increasing the performance of the model. For the simulations, we use $\tau_{\max} = 100$.

**Spike-Efficiency:** *Given an input signal x(t), the spike-efficiency ($\mathcal{E}$) of a trained RSNN model is defined as the ratio of the memory capacity $\mathcal{C}$ to the average total spike count per neuron $\tilde{S}$.*

$\mathcal{E}$ is an analytical measure used to compare how $\mathcal{C}$ and hence the model's performance is improved with per unit spike activity in the model. Ideally, we want to design a system with high $\mathcal{C}$ using fewer spikes. Hence we define $\mathcal{E}$ as the ratio of the memory capacity using $N_\mathcal{R}$ neurons $\mathcal{C}(N_\mathcal{R})$ to the average number of spike activations per neuron ($\tilde{S}$) and is given as:

$$\mathcal{E} = \frac{\mathcal{C}(N_\mathcal{R})}{\frac{\sum_{i=1}^{N_\mathcal{R}} S_i}{N_\mathcal{R}}}, \qquad S_i = \int_0^T s_i(t)dt \approx N_{\mathrm{post}} \frac{T}{\int_{t_{ref}}^\infty t\Phi_i dt} \tag{4}$$

where $N_{\mathrm{post}}$ is the number of postsynaptic neurons, $\Phi_i$ is the inter-spike interval spike frequency for neuron $i$, and $T$ is the total time. It is to be noted here that the total spike count $S$ is obtained by counting the total number of spikes in all the neurons in the recurrent layer until the emission of the first spike at the readout layer.

## 3 Heterogeneous RSNN: Analytical Results

We present three main analytical findings. Firstly, neuronal dynamic heterogeneity increases memory capacity by capturing more principal components from the input space, leading to better performance and improved $\mathcal{C}$. Secondly, STDP dynamic heterogeneity decreases spike activation without affecting $\mathcal{C}$, providing better orthogonalization among the recurrent network states and a more efficient representation of the input space, lowering higher-order correlation in spike trains. This makes the model more spike-efficient since the higher-order correlation progressively decreases the information available through neural population (Montani et al., 2009; Abbott & Dayan, 1999). Finally, incorporating heterogeneity in both neuron and STDP dynamics boosts the $\mathcal{C}$ to spike activity ratio, i.e., $\mathcal{E}$, which enhances performance while reducing spike counts.

**Memory Capacity:** The performance of an RSNN depends on its ability to retain the memory of previous inputs. To quantify the relationship between the recurrent layer dynamics and $\mathcal{C}$, we note that extracting information from the recurrent layer is made using a combination of the neuronal states. Hence, more linearly independent neurons would offer more variable states and, thus, more extended memory.

***Lemma 3.1.1:*** *The state of the neuron can be written as follows:* $r_i(t) = \sum_{k=0}^{N_R}\sum_{n=1}^{N_R} \lambda_n^k \left\langle v_n^{-1}, \mathbf{w}^{\mathrm{in}} \right\rangle (v_n)_i x(t-k),$ *where* $\mathbf{v}_n, \mathbf{v}_n^{-1} \in \mathbf{V}$ *are, respectively, the left and right eigenvectors of* $\mathbf{W}$, $\mathbf{w}^{in}$ *are the input weights, and* $\lambda_n^k \in \lambda$ *belongs to the diagonal matrix containing*

*the eigenvalues of* $\mathbf{W}$; $\mathbf{a}_i = [a_{i,0}, a_{i,1}, \ldots]$ *represents the coefficients that the previous inputs* $\mathbf{x}_t = [x(t), x(t-1), \ldots]$ *have on* $r_i(t)$.

**Short Proof:** *(See Suppl. Sec. B for full proof)* As discussed by Aceituno et al. (2020), the state of the neuron can be represented as $\mathbf{r}(t) = \mathbf{W}\mathbf{r}(t-1) + \mathbf{w}^{\text{in}} x(t)$, where $\mathbf{w}^{\text{in}}$ are the input weights. We can simplify this using the coefficients of the previous inputs and plug this term into the covariance between two neurons. Hence, writing the input coefficients $\mathbf{a}$ as a function of the eigenvalues of $\mathbf{W}$,

$$\mathbf{r}(t) = \sum_{k=0}^{\infty} \mathbf{W}^k \mathbf{w}^{\text{in}} x(t-k) = \sum_{k=0}^{\infty} \left( \mathbf{V}\mathbf{\Lambda}^k \mathbf{V}^{-1} \right) \mathbf{w}^{\text{in}} x(t-k) \Rightarrow r_i(t) = \sum_{k=0}^{N_R} \sum_{n=1}^{N_R} \lambda_n^k \left\langle v_n^{-1}, \mathbf{w}^{\text{in}} \right\rangle (v_n)_i \, x(t-k) \blacksquare$$

**Theorem 1:** *If the memory capacity of the HRSNN and MRSNN networks are denoted by* $\mathcal{C}_H$ *and* $\mathcal{C}_M$ *respectively, then,* $\mathcal{C}_H \geq \mathcal{C}_M$, *where the heterogeneity in the neuronal parameters* $\mathcal{H}$ *varies inversely to the correlation among the neuronal states measured as* $\sum_{n=1}^{N_R} \sum_{m=1}^{N_R} \text{Cov}^2 \left( x_n(t), x_m(t) \right)$ *which in turn varies inversely with* $\mathcal{C}$.

**Intuitive Proof:** *(See Suppl. Sec. B for full proof)* Aceituno et al. (2020) showed that the $\mathcal{C}$ increases when the variance along the projections of the input into the recurrent layer are uniformly distributed. We show that this can be achieved efficiently by using heterogeneity in the LIF dynamics. More formally, let us express the projection in terms of the state space of the recurrent layer. We show that the raw variance in the neuronal states $\mathcal{J}$ can be written as $\mathcal{J} = \dfrac{\sum_{n=1}^{N_{\mathcal{R}}} \lambda_n^2(\mathbf{\Sigma})}{\left( \sum_{n=1}^{N_{\mathcal{R}}} \lambda_n(\mathbf{\Sigma}) \right)^2}$ where $\lambda_n(\mathbf{\Sigma})$ is the $n$th eigenvalue of $\mathbf{\Sigma}$. We further show that with higher $\mathcal{H}$, the magnitude of the eigenvalues of $\mathbf{W}$ decreases and hence leads to a higher $\mathcal{J}$. Now, we project the inputs into orthogonal directions of the network state space and model the system as $\mathbf{r}(t) = \sum_{\tau=1}^{\infty} \mathbf{a}_\tau x(t-\tau) + \varepsilon_r(t)$ where the vectors $\mathbf{a}_\tau \in \mathbb{R}^N$ are correspond to the linearly extractable effect of $x(t-\tau)$ onto $\mathbf{r}(t)$ and $\varepsilon_r(t)$ is the nonlinear contribution of all the inputs onto the state of $\mathbf{r}(t)$. First, we show that $\mathcal{C}$ increases when the variance along the projections of the input into the recurrent layer is more uniform. Intuitively, the variances at directions $\mathbf{a}_\tau$ must fit into the variances of the state space, and since the projections are orthogonal, the variances must be along orthogonal directions. Hence, we show that increasing the correlation among the neuronal states increases the variance of the eigenvalues, which would decrease our memory bound $\mathcal{C}^*$. We show that heterogeneity is inversely proportional to $\sum_{n=1}^{N_{\mathcal{R}}} \text{Cov}^2 \left( x_n(t), x_m(t) \right)$.

We see that increasing the correlations between neuronal states decreases the heterogeneity of the eigenvalues, which reduces $\mathcal{C}$. We show that the variance in the neuronal states is bounded by the determinant of the covariance between the states; hence, covariance increases when the neurons become correlated. As $\mathcal{H}$ increases, neuronal correlation decreases. Aceituno et al. (2020) proved that the neuronal state correlation is inversely related to $\mathcal{C}$. Hence, for HRSNN, with $\mathcal{H} > 0$, $\mathcal{C}_H \geq \mathcal{C}_M$. $\blacksquare$

**Spiking Efficiency** We analytically prove that the average firing rate of HRSNN is lesser than the average firing rate of the MRSNN model by considering a subnetwork of the HRSNN network and modeling the pre-and post-synaptic spike trains using a nonlinear interactive Hawkes process with inhibition, as discussed by Duval et al. (2022). The details of the model are discussed in Suppl. Sec. B.

**Lemma 3.2.1:** *If the neuronal firing rate of the HRSNN network with only heterogeneity in LTP/LTD dynamics of STDP is represented as* $\Phi_R$ *and that of MRSNN represented as* $\Phi_M$, *then the HRSNN model promotes sparsity in the neural firing which can be represented as* $\Phi_R < \Phi_M$.

**Short Proof:** (*See Suppl. Sec. B for full proof*) We show that the average firing rate of the model with heterogeneous STDP (LTP/LTD) dynamics (averaged over the population of neurons) is lesser than the corresponding average neuronal activation rate for a model with homogeneous STDP dynamics. We prove this by taking a sub-network of the HRSNN model. Now, we model the input spike trains of the pre-synaptic neurons using a multivariate interactive, nonlinear Hawkes process with multiplicative inhibition. Let us consider a population of neurons of size $N$ that is divided into population $A$ (excitatory) and population $B$ (inhibitory). We use a particular instance of the model given in terms of a family of counting processes $\left( Z_t^1, \ldots, Z_t^{N_A} \right)$ (population $A$) and $\left( Z_t^{N_A+1}, \ldots, Z_t^N \right)$ (population

$B$ ) with coupled conditional stochastic intensities given respectively by $\lambda^A$ and $\lambda^B$ as follows:

$$\lambda_t^{A,N} := \Phi_A \left( \frac{1}{N} \sum_{j \in A} \int_0^{t^-} h_1(t-u) dZ_u^j \right) \Phi_{B \to A} \left( \frac{1}{N} \sum_{j \in B} \int_0^{t^-} h_2(t-u) dZ_u^j \right)$$

$$\lambda_t^{B,N} := \Phi_B \left( \frac{1}{N} \sum_{j \in B} \int_0^{t^-} h_3(t-u) dZ_u^j \right) + \Phi_{A \to B} \left( \frac{1}{N} \sum_{j \in A} \int_0^{t^-} h_4(t-u) dZ_u^j \right) \quad (5)$$

where $A, B$ are the populations of the excitatory and inhibitory neurons, respectively, $\lambda_t^i$ is the intensity of neuron $i$, $\Phi_i$ a positive function denoting the firing rate, and $h_{j \to i}(t)$ is the synaptic kernel associated with the synapse between neurons $j$ and $i$. Hence, we show that the heterogeneous STDP dynamics increase the synaptic noise due to the heavy tail behavior of the system. This increased synaptic noise leads to a reduction in the number of spikes of the post-synaptic neuron. Intuitively, a heterogeneous STDP leads to a non-uniform scaling of correlated spike-trains leading to de-correlation. Hence, we can say that heterogeneous STDP models have learned a better-orthogonalized subspace representation, leading to better encoding of the input space with fewer spikes. ∎

**Theorem 2:** *For a given number of neurons $N_{\mathcal{R}}$, the spike efficiency of the model $\mathcal{E} = \frac{\mathcal{C}(N_{\mathcal{R}})}{\tilde{S}}$ for HRSNN ($\mathcal{E}_R$) is greater than MRSNN ($\mathcal{E}_M$) i.e., $\mathcal{E}_R \geq \mathcal{E}_M$*

**Short Proof:** *(See Suppl. Sec. B for full proof)* First, using Lemma 3.2.1, we show that the number of spikes decreases when we use heterogeneity in the LTP/LTD Dynamics. Hence, we compare the efficiencies of HRSNN with that of MRSNN as follows:

$$\frac{\mathcal{E}_R}{\mathcal{E}_M} = \frac{\mathcal{C}_R(N_{\mathcal{R}}) \times \tilde{S}_M}{\tilde{S}_R \times \mathcal{C}_M(N_{\mathcal{R}})} = \frac{\sum_{\tau=1}^{N_{\mathcal{R}}} \frac{\text{Cov}^2(x(t-\tau), \mathbf{a}_\tau^R \mathbf{r}_R(t))}{\text{Var}(\mathbf{a}_\tau^R \mathbf{r}_R(t))} \times \int_{t_{ref}}^{\infty} t \Phi_R dt}{\sum_{\tau=1}^{N_{\mathcal{R}}} \frac{\text{Cov}^2(x(t-\tau), \mathbf{a}_\tau^M \mathbf{r}_M(t))}{\text{Var}(\mathbf{a}_\tau^M \mathbf{r}_M(t))} \times \int_{t_{ref}}^{\infty} t \Phi_M dt} \quad (6)$$

Since $S_R \leq S_M$ and also,the covariance increases when the neurons become correlated, and as neuronal correlation decreases, $\mathcal{H}_X$ increases (Theorem 1), we see that $\mathcal{E}_R/\mathcal{E}_M \geq 1 \Rightarrow \mathcal{E}_R \geq \mathcal{E}_M$ ∎

**Optimal Heterogeneity using Bayesian Optimization for Distributions** To get optimal heterogeneity in the neuron and STDP dynamics, we use a modified Bayesian Optimization (BO) technique. However, using BO for high-dimensional problems remains a significant challenge. In our case, optimizing HRSNN model parameters for 5000 neurons requires the optimization of two parameters per neuron and four parameters per STDP synapse, where standard BO fails to converge to an optimal solution. However, the parameters to be optimized are correlated and can be drawn from a probability distribution as shown by Perez-Nieves et al. (2021). Thus, we design a modified BO to estimate parameter distributions instead of individual parameters for the LIF neurons and the STDP synapses, for which we modify the BO's surrogate model and acquisition function. This makes our modified BO highly scalable over all the variables (dimensions) used. The loss for the surrogate model's update is calculated using the Wasserstein distance between the parameter distributions. We use the modified Matern function on the Wasserstein metric space as a kernel function for the BO problem. The detailed BO methods are discussed in Suppl. Sec. A. BO uses a Gaussian process to model the distribution of an objective function and an acquisition function to decide points to evaluate. For data points $x \in X$ and the corresponding output $y \in Y$, an SNN with network structure $\mathcal{V}$ and neuron parameters $\mathcal{W}$ acts as a function $f_{\mathcal{V}, \mathcal{W}}(x)$ that maps input data $x$ to $y$. The optimization problem can be defined as: $\min_{\mathcal{V}, \mathcal{W}} \sum_{x \in X, y \in Y} \mathcal{L}(y, f_{\mathcal{V}, \mathcal{W}}(x))$ where $\mathcal{V}$ is the set of hyperparameters of the neurons in $\mathcal{R}$ and $\mathcal{W}$ is the multi-variate distribution constituting the distributions of: (i) the membrane time constants $\tau_{m-E}, \tau_{m-I}$ of LIF neurons, (ii) the scaling function constants $(A_+, A_-)$ and (iii) the decay time constants $\tau_+, \tau_-$ for the STDP learning rule in $\mathcal{S}_{\mathcal{R}\mathcal{R}}$.

## 4 EXPERIMENTAL RESULTS

**Model and Architecture** We empirically verify our analytical results using HRSNN for classification and prediction tasks. Fig. 2 shows the overall architecture of the prediction model. Using a rate-encoding methodology, the time-series data is encoded to a series of spike trains. This high-dimensional spike train acts as the input to HRSNN. The output spike trains from HRSNN act as the

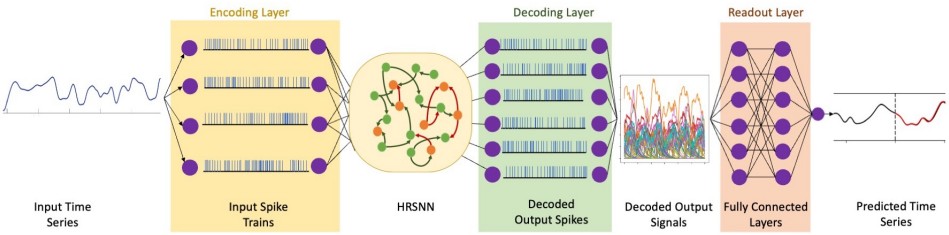

Figure 2: Block Diagram showing the methodology using HRSNN for prediction

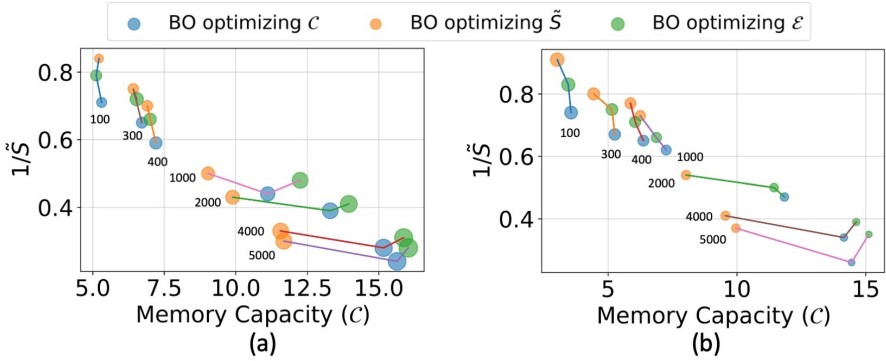

Figure 3: Figure showing the results of ablation studies of BO for the following three cases: (a) for prediction problem - the radius indicates the normalized NRMSE loss with a smaller radius indicating lower (better) NRMSE (b) for classification problem - the radius indicates the normalized accuracy with larger radius indicating higher (better) accuracy. The numbers represent the number of neurons used in each model, and the line joins the three corresponding models with the same model size. The detailed results are shown in Suppl. Sec. A

input to a decoder and a readout layer that finally gives the prediction results. For the classification task, we use a similar method. However, we do not use the decoding layer for the signal but directly feed the output spike signals from HRSNN into the fully connected layer. The complete details of the models used and description of the different modules used in Fig. 2 is discussed in Suppl. Sec. A.

**Datasets:** *Classification:* We use the Spoken Heidelberg Digits (SHD) spiking dataset to benchmark the HRSNN model with other standard spiking neural networks (Cramer et al., 2020).
*Prediction:* We use a multiscale Lorenz 96 system (Lorenz, 1996) which is a set of coupled nonlinear ODEs and an extension of Lorenz's original model for multiscale chaotic variability of weather and climate systems which we use as a testbed for the prediction capabilities of the HRSNN model (Thornes et al., 2017). Further details on both datasets are provided in Suppl. Sec. A.

**Bayesian Optimization Ablation Studies:** First, we perform an ablation study of BO for the following three cases: (i) Using Memory Capacity $\mathcal{C}$ as the objective function (ii) Using Average Spike Count $\tilde{S}$ as the objective function and (iii) Using $\mathcal{E}$ as the objective function. We optimize both LIF neuron parameter distribution and STDP dynamics distributions for each. We plot $\mathcal{C}$, $\tilde{S}$, the empirical spike efficiency $\hat{\mathcal{E}}$, and the observed RMSE of the model obtained from BO with different numbers of neurons. The results for classification and prediction problems are shown in Fig. 3(a) and (b), respectively. Ideally, we want to design networks with high $\mathcal{C}$ and low spike count, i.e., models in the upper right corner of the graph. The observed results show that BO using $\mathcal{E}$ as the objective gives the best accuracy with the fewest spikes. Thus, we can say that this model has learned a better-orthogonalized subspace representation, leading to better encoding of the input space with fewer spikes. Hence, for the remainder of this paper, we focus on this BO model, keeping the $\mathcal{E}$ as the objective function. This Bayesian Optimization process to search for the optimal hyperparameters of the model is performed before training and inference using the model and is generally equivalent to the network architecture search process used in deep learning. Once we have these optimal hyper-parameters, we freeze these hyperparameters, learn (unsupervised) the network parameters

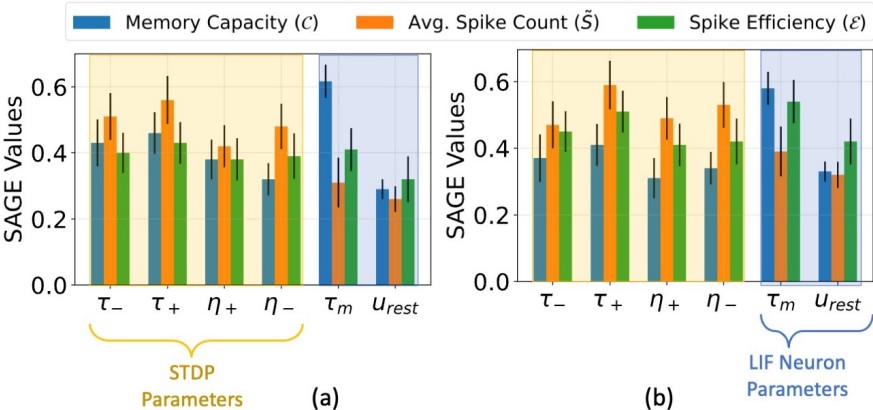

Figure 4: Bar chart showing the global importance of different heterogeneous parameters using HRSNN on the dataset. The experiments were repeated five times with different parameters from the same distribution (a) Classification (b) Prediction

(i.e., synaptic weights) of the HRSNN while using the frozen hyperparameters, and generate the final HRSNN model for inference. In other words, the hyperparameters, like the distribution of membrane time constants or the distribution of synaptic time constants for STDP, are fixed during the learning and inference. Further details of the Bayesian Optimization procedure, including the parameterized variables and the final searched distributions of the hyperparameters, are shown in Suppl. Sec. A, where we also discuss the convergence analysis of the three different BOs discussed above.

**Heterogeneity Parameter Importance:** We use SAGE (Shapley Additive Global importancE) (Covert et al., 2020), a game-theoretic approach to understand black-box models to calculate the significance of adding heterogeneity to each parameter for improving $\mathcal{C}$ and $\tilde{S}$. SAGE summarizes the importance of each feature based on the predictive power it contributes and considers complex feature interactions using the principles of Shapley value, with a higher SAGE value signifying a more important feature. We tested the HRSNN model using SAGE on the Lorenz96 and the SHD datasets. The results are shown in Fig. 4. We see that $\tau_m$ has the greatest SAGE values for $\mathcal{C}$, signifying that it has the greatest impact on improving $\mathcal{C}$ when heterogeneity is added. Conversely, we see that heterogeneous STDP parameters (viz., $\tau_{\pm}, \eta_{\pm}$) play a more critical role in determining the average neuronal spike activation. Hence, we confirm the notions proved in Sec. 3 that heterogeneity in neuronal dynamics improves the $\mathcal{C}$ while heterogeneity in STDP dynamics improves the spike count. Thus, we need to optimize the heterogeneity of both to achieve maximum $\mathcal{E}$.

**Results:** We perform an ablation study to evaluate the performance of the HRSNN model and compare it to standard BP-based spiking models. We study the performances of both the SHD dataset for classification and the Lorenz system for prediction. The results are shown in Table 2. We compare the Normalized Root Mean Squared Error (NRMSE) loss (prediction), Accuracy (classification), Average Spike Count $\tilde{S}$ and the application level empirical spiking efficiency $\hat{\mathcal{E}}$ calculated as $\dfrac{1}{\text{NRMSE} \times \tilde{S}}$ (prediction) and $\dfrac{\text{Accuracy}}{\tilde{S}}$ (classification). We perform the experiments using 5000 neurons in $\mathcal{R}$ on both classification and prediction datasets. We see that the HRSNN model with heterogeneous LIF and heterogeneous STDP outperforms other HRSNN and MRSNN models in terms of NRMSE scores while keeping the $\tilde{S}$ much lower than HRSNN with heterogeneous LIF and homogeneous STDP. From the experiments, we can conclude that the heterogeneous LIF neurons have the greatest contribution to improving the model's performance. In contrast, heterogeneity in STDP has the most significant impact on a spike-efficient representation of the data. HRSNN with heterogeneous LIF and STDP leverages the best of both worlds by achieving the best RMSE with low spike activations, as seen from Table 2. Further detailed results on limited training data are added in Suppl. Sec. A. We also compare the generalizability of the HRSNN vs. MRSNN models, where we empirically show that the heterogeneity in STDP dynamics helps improve the overall model's generalizability. In addition, we discuss how HRSNN reduces the effect of higher-order correlations, thereby giving rise to a more efficient representation of the state space.

Table 2: Table showing the comparison of the Accuracy and NRMSE losses for the SHD Classification and Lorenz System Prediction tasks, respectively. We show the average spike rate, calculated as the ratio of the moving average of the number of spikes in a time interval $T$. For this experiment, we choose $T = 4ms$ and a rolling time span of $2ms$, which is repeated until the first spike appears in the final layer. Following the works of Paul et al. (2022), we show that the normalized average spike rate is the total number of spikes generated by all neurons in an RSNN averaged over the time interval $T$. The results marked with * denotes we implemented the open-source code for the model and evaluated the given results.

| | | | SHD (Classification) | | | Chaotic Lorenz System (Prediction) | | |
|---|---|---|---|---|---|---|---|---|
| | Method | Accuracy $(A)$ | Normalized Avg. Firing Rate $(\frac{\tilde{S}}{T})$ | Efficiency $(\hat{\mathcal{E}} = \frac{A}{\tilde{S}})$ | NRMSE | Normalized Avg. Firing Rate $(\frac{\tilde{S}}{T})$ | Efficiency $\hat{\mathcal{E}} = \frac{1}{\text{NRMSE} \times \tilde{S}}$ |
| **Unsupervised RSNN** | MRSNN (Homogeneous LIF, Homogeneous STDP) | 73.58 | -0.508 | $18.44 \times 10^{-3}$ | 0.395 | -0.768 | $0.787 \times 10^{-3}$ |
| | HRSNN (Heterogeneous LIF, Homogeneous STDP) | 78.87 | 0.277 | $17.19 \times 10^{-3}$ | 0.203 | -0.143 | $1.302 \times 10^{-3}$ |
| | HRSNN (Homogeneous LIF, Heterogeneous STDP) | 74.03 | -1.292 | $22.47 \times 10^{-3}$ | 0.372 | -1.102 | $0.932 \times 10^{-3}$ |
| | HRSNN (Heterogeneous LIF, Heterogeneous STDP) | 80.49 | -1.154 | $24.35 \times 10^{-3}$ | 0.195 | -1.018 | $1.725 \times 10^{-3}$ |
| **RSNN with BP** | MRSNN-BP (Homogeneous LIF, BP) | 81.42 | 0.554 | $16.9 \times 10^{-3}$ | 0.182 | 0.857 | $1.16 \times 10^{-3}$ |
| | HRSNN-BP (Heterogeneous LIF, BP) | 83.54 | 1.292 | $15.42 \times 10^{-3}$ | 0.178 | 1.233 | $1.09 \times 10^{-3}$ |
| | Adaptive SRNN (Yin et al., 2020) | 84.46 | $0.831^*$ | $17.21^* \times 10^{-3}$ | $0.174^*$ | $0.941^*$ | $1.19^* \times 10^{-3}$ |

## 5 CONCLUSION

This paper analytically and empirically proved that heterogeneity in neuronal (LIF) and synaptic (STDP) dynamics leads to an unsupervised RSNN with more memory capacity, reduced spiking count, and hence, better spiking efficiency. We show that HRSNN can achieve similar performance as an MRSNN but with sparse spiking leading to the improved energy efficiency of the network. In conclusion, this work establishes important mathematical properties of an RSNN for neuromorphic machine learning applications like time series classification and prediction. However, it is interesting to note that the mathematical results from this paper also conform to the recent neurobiological research that suggests that the brain has large variability between the types of neurons and learning methods. For example, intrinsic biophysical properties of neurons, like densities and properties of ionic channels, vary significantly between neurons where the variance in synaptic learning rules invokes reliable and efficient signal processing in several animals (Marder & Taylor, 2011; Douglass et al., 1993). Experiments in different brain regions and diverse neuronal types have revealed a wide range of STDP forms with varying neuronal dynamics that vary in plasticity direction, temporal dependence, and the involvement of signaling pathways (Sjostrom et al., 2008; Korte & Schmitz, 2016). Thus, heterogeneity is essential in encoding and decoding stimuli in biological systems. In conclusion, this work establishes connections between the mathematical properties of an RSNN for neuromorphic machine learning applications like time series classification and prediction with neurobiological observations. There are some key limitations to the analyses in this paper. First, the properties discussed are derived independently. An important extension will be to consider all factors simultaneously. Second, we assumed an idealized spiking network where the memory capacity is used to measure its performance, and the spike count measures the energy. Also, we mainly focused on the properties of RSNN trained using STDP. An interesting connection between synchronization and heterogeneous STDP remains a topic that needs to be studied further - whether we can optimally engineer the synchronization properties to improve the model's performance. Finally, the empirical evaluations were presented for the prediction task on a single dataset. More experimental evaluations, including other tasks and datasets, will strengthen the empirical validations.

ACKNOWLEDGEMENT

This work is supported by the Army Research Office and was accomplished under Grant Number W911NF-19-1-0447. The views and conclusions contained in this document are those of the authors and should not be interpreted as representing the official policies, either expressed or implied, of the Army Research Office or the U.S. Government.

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

# Supplementary Section

## Table of Contents

# A SUPPLEMENTARY SECTION A

## A.1 EXPERIMENTAL SETUP

### A.1.1 HRSNN AND MRSNN

Table 3: Table showing the description of the models described in the paper

|  | **LIF Neurons** | **STDP parameters** |
|---|---|---|
| **HRSNN** | Heterogeneous | Heterogeneous |
| **MRSNN** | Homogeneous | Homogeneous |

The models used in this paper were the Heterogeneous Recurrent SNN (HRSNN) and the Homogeneous Recurrent SNN (MRSNN). Both models use STDP as the learning method. For MRSNN, we use STDP with uniform parameters for all the synapses. However, for HRSNN, we use a distribution for each parameter to get a rich class of diverse LTP/LTD dynamics. But, at the core, all the training is done using STDP.

### A.1.2 LIF NEURON NUMERICAL IMPLEMENTATION

To implement the LIF model, we discretize time into multiples of a small-time step $\Delta t$ so that spikes can only happen at multiples of $\Delta t$. (Cramer et al., 2020; Perez-Nieves et al., 2021) Thus, we can approximately solve Eq. 1 as

$$v_i(t+\Delta t) = v_i[t+1] = \beta \left( v_i[t] - v_0 \right) + v_0 + (1-\beta) I_i[t] - (v_{th} - v_r) S_i[t] \quad \text{s.t.} \ \beta = \exp^{(-\Delta t/\tau_m)} \quad (7)$$

It is to be noted here that we use this approximation for numerically solving the LIF neurons. Hence, although we use continuous notations for the remainder of the paper, it is to be noted that we use the discrete form discussed here for numerical solutions.

### A.1.3 SPIKE CODING

**Encoding:** For the RSNN to process our time series, the signal must be represented as spikes. We use a temporal encoding technique for representing signals in this paper. The spikes are only generated whenever the signal changes in value. The implementation of the temporal encoding used in this research is based on the Step-Forward (SF) algorithm (Petro et al., 2019). The percentage of neurons to input the spikes to ($\alpha$) is also chosen to provide good recurrent layer dynamics.

**Decoding:** To represent the recurrent state, we use an exponentially decreasing rate decoding strategy by taking the sum of all the spikes $s$ over the last $\tau$ timesteps into account as follows:

$$x_i^X(t) = \sum_{n=0}^{\tau} \gamma^n s_i(t-n) \quad \forall i \in E$$

where X denotes the model representation. The parameters $\tau$ and $\gamma$ are balanced to optimize the memory size of the stored data (e.g., $\tau \le 50$ ) and its containment of information, which includes adjusting $\tau$ to the pace at which the temporal data is presented and processed. The state of the recurrent layer will be only based on the output of excitatory neurons. Thus, it is crucial for the discount $\gamma$ not to be too small, as it possibly flattens older values in the window to 0, making part of the sliding window unusable. Recent spikes hardly affect the recurrent layer state when setting $\gamma$ too high in combination with a large window size. This causes the decoder to react too late to recent information provided by the recurrent layer and complicates the learning process of the readout layer.

### A.1.4 READOUT

After the initialization of the recurrent layer, the readout is the only component of the LSM with trainable parameters. It consists of a single fully connected layer for regression or classification. The readout does not have to be any deeper, as the output of the recurrent layer is already a high-dimensional representation of the processed input. $x$ and $y$ present the continuous signals of the time series $T$ and model representation $X$.

$$x^T(t+k) = y^T(t) \approx \hat{y}^X(t) = f_\theta\left(x^X(t)\right)$$

The mean squared error (MSE) is used as the loss function to train the readout, and the network was trained using the stochastic optimizer Adam (Kingma & Ba, 2014).

$$\mathcal{L}\left(y^T, \hat{y}^X\right) = \frac{1}{n}\sum_{i=0}^{n}\left(y_i^T - \hat{y}_i^X\right)^2$$

### A.1.5 DATASETS

**Lorenz96: (Lorenz, 1996)**Our objective is more clearly demonstrated using the canonical chaotic system we will use as a test bed for the prediction capabilities of the HRSNN model. We use a multiscale Lorenz 96 system which is a set of coupled nonlinear ODEs and an extension of Lorenz's original model (Thornes et al., 2017), (Chattopadhyay et al., 2020).

$$\begin{aligned}
\frac{\mathrm{d}X_k}{\mathrm{d}t} &= X_{k-1}\left(X_{k+1} - X_{k-2}\right) + F - \frac{hc}{b}\Sigma_j Y_{j,k} \\
\frac{\mathrm{d}Y_{j,k}}{\mathrm{d}t} &= -cbY_{j+1,k}\left(Y_{j+2,k} - Y_{j-1,k}\right) - cY_{j,k} + \frac{hc}{b}X_k - \frac{he}{d}\Sigma_i Z_{i,j,k} \\
\frac{\mathrm{d}Z_{i,j,k}}{\mathrm{d}t} &= edZ_{i-1,j,k}\left(Z_{i+1,j,k} - Z_{i-2,j,k}\right) - geZ_{i,j,k} + \frac{he}{d}Y_{j,k}
\end{aligned} \qquad (8)$$

This set of coupled nonlinear ordinary differential equations (ODEs) is a three-tier extension of Lorenz's original model (Lorenz, 1963) and has been proposed by Thornes et al.Thornes et al. (2017) as a fitting prototype for multiscale chaotic variability of the weather and climate system and a useful test bed for novel methods. In these equations, $F = 20$ is a large-scale forcing that makes the system highly chaotic, and $b = c = e = d = g = 10$ and $h = 1$ are tuned to produce appropriate spatiotemporal variability. For this paper, we focus on predicting $Y$ axes, which have relatively moderate amplitudes compared to $X, Z$ and demonstrate high-frequency variability and intermittency, which makes the

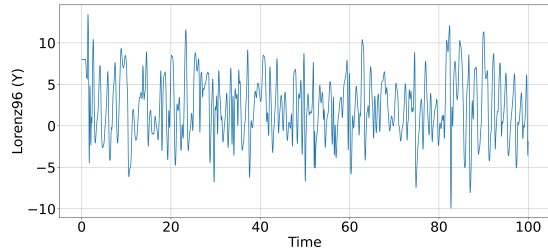

Figure 5: Figure showing a snippet for the Y dimension of the Lorenz96 time series used for the prediction problem

prediction problem difficult. It is to be noted here that the Lorenz 96 is a complex, difficult dataset for climate modeling. A snippet of the time series is shown in Fig. 5.

**SHD dataset:** We use the Spoken Heidelberg Digits spiking dataset to benchmark the HRSNN model with other standard spiking neural networks (Cramer et al., 2020). It was created based on the Heidelberg Digits (HD) audio dataset which comprises 20 classes of spoken digits from zero to nine in English and German, spoken by 12 individuals. For training and evaluation, the dataset (10420 samples) is split into a training set (8156 samples) and test set (2264 samples). To apply our RSNNs, we converted all audio samples into 250- by-700 binary matrices. For this, all samples fit within a 1 the second window, shorter samples were padded with zeros, and longer samples were cut by removing the tail. Spikes were then binned in time bins, both of sizes 10ms and 4ms; for the RSNNs, the presence or non-presence of any spikes in the time bin is noted as a single binary event.

### A.1.6 HYPERPARAMETERS

The hyperparameters used in this paper are summarized in Table 4

Table 4: Table showing the hyperparameters used in the experiments and their values

| Parameter | Value | Description | | | |
|---|---|---|---|---|---|
| \|E\|/\|N\| | 80% | Excitatory/inhibitory ratio | | | |
| $\lambda$ | 2 | Leak Exponent | | | |
| $\tau$ | 50 | Sliding window size | | | |
| $\gamma^{\tau-1}$ | 0.02 | Sliding Window Leak | | | |
| **SHD Parameters** | | | **Lorenz-63 Parameters** | | |
| **Parameter** | **Value** | **Description** | **Parameter** | **Value** | **Description** |
| $\tau$ | 17 | Time delay | $\rho$ | 28.0 | $\rho$-parameter |
| $a$ | 0.2 | a parameter | $\sigma$ | 10.0 | $\sigma$-parameter |
| $b$ | 0.1 | b parameter | $\beta$ | 8/3 | $\beta$-parameter |
| n | 10 | n parameter | $x_0$ | [1.0,1.0,1.0] | Initial Condition |
| $x_0$ | 1.2 | Initial Condition | h | 0.03 | Time delta between two discrete timesteps |
| h | 1.0 | Time delta between two discrete timesteps | | | |

### A.2 MAXIMUM ENTROPY DISTRIBUTION

This subsection proves that the maximum entropy distribution with a fixed covariance matrix is Gaussian.

**Lemma:** *Let $q(\mathbf{r})$ be any density satisfying $\int q(\mathbf{r})x_i x_j d\mathbf{r} = \Sigma_{ij}$. Let $p = \mathcal{N}(\mathbf{0}, \boldsymbol{\Sigma})$. Then $h(q) \leq h(p)$*
**Proof.**

$$0 \leq \mathbb{KL}(q\|p) = \int q(\mathbf{r}) \log \frac{q(\mathbf{r})}{p(\mathbf{r})} d\mathbf{r}$$

$$= -h(q) - \int q(\mathbf{r}) \log p(\mathbf{r}) d\mathbf{r}$$

$$= -h(q) - \int p(\mathbf{r}) \log p(\mathbf{r}) d\mathbf{r}$$

$$= -h(q) + h(p)$$

since $q$ and $p$ yield the same moments for the quadratic form encoded by $\log p(\mathbf{r})$.

### A.3 OPTIMAL HYPERPARAMETER SELECTION USING BAYESIAN OPTIMIZATION

Most recent research in Bayesian Optimization (BO) applications is limited to low-dimensional problems, as BO fails catastrophically when generalizing to high-dimensional problems (Frazier, 2018). However, in this paper, we aim to use BO to optimize the neuronal and synaptic parameters of a heterogeneous RSNN model. This BO problem thus entails a huge number of hyperparameters to be optimized; hence, using standard BO algorithms remains a significant challenge. Hence, to overcome this issue, we used a novel BO algorithm based on the assumption that our hyperparameters to be optimized are not completely random and uncorrelated but can be thought of as being drawn from a probability distribution as shown by Perez et al.Perez-Nieves et al. (2021). Thus, instead of searching for the individual parameters themselves, we use a modified BO to estimate *parameter distributions* for the LIF neurons and the STDP dynamics. After learning the optimal distributions, we simply sample from the distribution to get the distribution of hyperparameters used in the model. To learn the probability distribution of the data, we modify BO's surrogate model and acquisition function to treat the parameter distributions instead of individual variables. This makes our modified BO highly scalable over all the variables (dimensions) used. The loss for the surrogate model's update is calculated using the Wasserstein distance between the parameter distributions.

BO uses a Gaussian process to model the distribution of an objective function and an acquisition function to decide on points to evaluate. For data points in a target dataset $x \in X$ and the corresponding label $y \in Y$, an SNN with network structure $\mathcal{V}$ and neuron parameters $\mathcal{W}$ acts as a function $f_{\mathcal{V},\mathcal{W}}(x)$ that maps input data $x$ to predicted label $\tilde{y}$. The optimization problem in this work is defined as

$$\min_{\mathcal{V},\mathcal{W}} \sum_{x \in X, y \in Y} \mathcal{L}\left(y, f_{\mathcal{V},\mathcal{W}}(x)\right) \tag{9}$$

where $\mathcal{V}$ is the set of hyperparameters of the neurons in $\mathcal{R}$ (Details of hyperparameters given in the Supplementary) and $\mathcal{W}$ is the multi-variate distribution constituting the distributions of (i) the membrane time constants $\tau_{m-E}, \tau_{m-I}$ of the LIF neurons, (ii) the scaling function constants $(A_+, A_-)$ and (iii) the decay time constants $\tau_+, \tau_-$ for the STDP learning rule in $\mathcal{S}_{\mathcal{R}\mathcal{R}}$.

Again, BO needs a prior distribution of the objective function $f(\vec{x})$ on the given data $\mathcal{D}_{1:k} = \{\vec{x}_{1:k}, f(\vec{x}_{1:k})\}$. In the Gaussian Process (GP)-based BO, we assume that the prior distribution of $f(\vec{x}_{1:k})$ follows the multivariate Gaussian distribution, which follows a GP with mean $\vec{\mu}_{\mathcal{D}_{1:k}}$ and covariance $\vec{\Sigma}_{\mathcal{D}_{1:k}}$. Thus, we estimate $\vec{\Sigma}_{\mathcal{D}_{1:k}}$ using the modified Matern kernel function. We use the loss function as $d(x, x')$, which is the Wasserstein distance between the multivariate distributions of the different parameters. That is, given two distributions of hyperparameters $x_1, x_2$, the distance between these two distributions (given as $d(x_1, x_2)$ is used as the loss function in the Matern kernel for the modified BO. We want to learn the optimal distribution of hyperparameters $x'$, which maximizes the performance. It is to be noted here that for higher-dimensional metric spaces, we use the Sinkhorn distance as a regularized version of the Wasserstein distance to approximate the Wasserstein distance (Feydy et al., 2019).

$\mathcal{D}_{1:k}$ are the points evaluated by the objective function. The GP will estimate the mean $\vec{\mu}_{\mathcal{D}_{k:n}}$ and variance $\vec{\sigma}_{\mathcal{D}_{k:n}}$ for the rest unevaluated data $\mathcal{D}_{k:n}$. The acquisition function used in this work is the expected improvement (EI) of the prediction fitness as:

$$EI\left(\vec{x}_{k:n}\right) = \left(\vec{\mu}_{\mathcal{D}_{k:n}} - f\left(x_{\text{best}}\right)\right) \Phi(\vec{Z}) + \vec{\sigma}_{\mathcal{D}_{k:n}} \phi(\vec{Z}) \tag{10}$$

where $\Phi(\cdot)$ and $\phi(\cdot)$ denote the probability distribution function and the cumulative distribution function of the prior distributions, respectively. $f(x_{\text{best}}) = \max f(\vec{x}_{1:k})$ is the maximum value that

Table 5: The list of parameter settings for the Bayesian Optimization-based hyperparameter search

| Parameter | Initial Value | Range |
|---|---|---|
| $\eta$ | 10 | (0,50) |
| $\gamma$ | 5 | (0,10) |
| $\zeta$ | 2.5 | (0,10) |
| $\eta^*$ | 1 | (0,3) |
| $g$ | 2 | (0,10) |
| $\omega$ | 0.5 | (0,1) |
| $k$ | 50 | (0,100) |
| $\lambda$ (SHD) | 1 | (0,2) |
| $\lambda$ (Lorenz) | 1.5 | (0,4) |
| $P_{IR}$ | 0.05 | (0,0.1) |
| $\tau_{n-E}, \tau_{n-I}$ (SHD) | $50ms$ | $(0ms, 100ms)$ |
| $\tau_{n-E}, \tau_{n-I}$ (Lorenz) | $100ms$ | $(0ms, 300ms)$ |
| $A_{en-R}, A_{EE}, A_{EI}, A_{IE}, A_{II}$ | 30 | (0,60) |

Table 6: Table showing the average final distributions of the hyperparameters

| | Parameter | Distribution | |
|---|---|---|---|
| STDP Parameter | $\tau_+$ | Normal | $\bar{\mu} = 18.235$ 
 $\bar{\sigma} = 1.522$ |
| | $\tau_-$ | Normal | $\bar{\mu} = 22.382$ 
 $\bar{\sigma} = 1.768$ |
| | $\eta_+$ | Normal | $\bar{\mu} = 0.516$ 
 $\bar{\sigma} = 0.0055$ |
| | $\eta_-$ | Normal | $\bar{\mu} = 0.448$ 
 $\bar{\sigma} = 0.0057$ |
| LIF Parameter | $\tau_m^{(e)}$ | Gamma | $\bar{\alpha} = 2.89$ 
 $1/\bar{\beta} = 0.248$ |
| | $\tau_m^{(i)}$ | Gamma | $\bar{\alpha} = 5.14$ 
 $1/\bar{\beta} = 0.313$ |

has been evaluated by the original function $f$ in all evaluated data $\mathcal{D}_{1:k}$ and $\vec{Z} = \frac{\vec{\mu}_{\mathcal{D}_{k:n}} - f(x_{\text{best}})}{\vec{\sigma}_{\mathcal{D}_{k:n}}}$. BO will choose the data $x_j = \text{argmax}\left\{EI\left(\vec{x}_{k:n}\right); x_j \subseteq \vec{x}_{k:n}\right\}$ as the next point to be evaluated using the original objective function.

### A.3.1 OPTIMIZED HYPERPARAMETERS

The list of the hyperparameters optimized using the Bayesian Optimization technique is shown in Table 5. We also show the range of the hyperparameters used and the initial values. In addition to this, Table 6 enlist the final optimized distributions of the STDP and the LIF parameters obtained using BO.

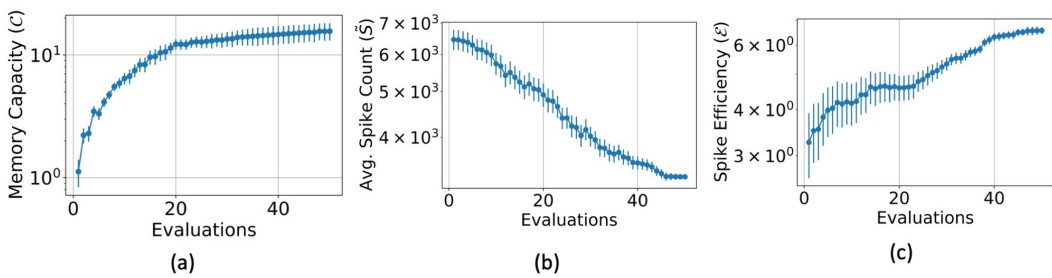

Figure 6: Figure Showing the convergence behaviors of the three types of BO described in the paper (a) BO optimizing the memory capacity $\mathcal{C}$ (b) BO optimizing the average spike count $\tilde{S}$ and (c) BO optimizing the spike efficiency $\mathcal{E}$

Table 7: Table showing the performance of the Bayesian Optimization on the SHD Classification dataset for the three different cases where BO 1 optimizes $\mathcal{C}$ , BO 2 optimizes $\tilde{S}$ and BO 3 optimizes $\mathcal{E}$

| N_R | BO 1 Memory Capacity | BO 1 Average Spike Count | BO 1 Accuracy | BO 2 Memory Capacity | BO 2 Average Spike Count | BO 2 Accuracy | BO 3 Memory Capacity | BO 3 Average Spike Count | BO 3 Accuracy |
|---|---|---|---|---|---|---|---|---|---|
| 100 | $5.31 \pm 0.36$ | $1399.14 \pm 113.66$ | $67.41 \pm 4.86$ | $5.22 \pm 0.58$ | $1189.37 \pm 86.95$ | $66.23 \pm 5.74$ | $5.12 \pm 0.31$ | $1268.86 \pm 91.15$ | $68.67 \pm 4.97$ |
| 200 | $5.45 \pm 0.39$ | $1479.86 \pm 133.72$ | $67.85 \pm 4.16$ | $5.38 \pm 0.53$ | $1233.19 \pm 122.56$ | $67.21 \pm 5.49$ | $5.28 \pm 0.34$ | $1328.18 \pm 131.45$ | $69.54 \pm 4.27$ |
| 300 | $6.72 \pm 0.4$ | $1541.14 \pm 148.83$ | $68.41 \pm 4.87$ | $6.42 \pm 0.57$ | $1325.25 \pm 128.47$ | $67.95 \pm 5.11$ | $6.54 \pm 0.36$ | $1391.68 \pm 140.37$ | $70.1 \pm 4.69$ |
| 400 | $7.21 \pm 0.44$ | $1682.68 \pm 239.95$ | $70.11 \pm 4.33$ | $6.91 \pm 0.51$ | $1425.69 \pm 129.57$ | $68.23 \pm 5.27$ | $7.01 \pm 0.35$ | $1511.79 \pm 200.96$ | $71.54 \pm 4.06$ |
| 500 | $8.59 \pm 0.48$ | $1768.24 \pm 287.94$ | $71.05 \pm 3.98$ | $7.69 \pm 0.46$ | $1555.29 \pm 139.17$ | $69.31 \pm 5.03$ | $8.63 \pm 0.38$ | $1621.8 \pm 250.46$ | $73.05 \pm 3.87$ |
| 1000 | $11.22 \pm 0.46$ | $2251.17 \pm 319.75$ | $72.93 \pm 3.38$ | $9.03 \pm 0.44$ | $2015.24 \pm 147.44$ | $70.89 \pm 4.91$ | $12.25 \pm 0.39$ | $2102.59 \pm 279.86$ | $75.32 \pm 3.44$ |
| 2000 | $13.3 \pm 0.51$ | $2566.21 \pm 348.68$ | $75.36 \pm 3.29$ | $9.89 \pm 0.46$ | $2314.59 \pm 151.18$ | $72.33 \pm 4.88$ | $13.95 \pm 0.37$ | $2410.08 \pm 301.57$ | $77.25 \pm 3.17$ |
| 3000 | $14.47 \pm 0.53$ | $2825.47 \pm 355.87$ | $77.14 \pm 3.38$ | $10.48 \pm 0.41$ | $2623.41 \pm 177.94$ | $74.63 \pm 4.93$ | $14.88 \pm 0.42$ | $2708.52 \pm 315.34$ | $78.21 \pm 3.24$ |
| 4000 | $15.17 \pm 0.52$ | $3551.07 \pm 366.19$ | $78.05 \pm 3.25$ | $11.57 \pm 0.45$ | $3045.28 \pm 225.53$ | $75.15 \pm 4.85$ | $15.87 \pm 0.38$ | $3218.42 \pm 328.19$ | $79.36 \pm 3.13$ |
| 5000 | $15.64 \pm 0.57$ | $4186.49 \pm 383.09$ | $78.92 \pm 3.31$ | $11.68 \pm 0.48$ | $3294.62 \pm 241.14$ | $75.87 \pm 4.81$ | $16.03 \pm 0.41$ | $3573.51 \pm 331.18$ | $80.49 \pm 3.15$ |

Table 8: Table showing the performance of the Bayesian Optimization on the Lorenz System Prediction dataset for the three different cases where BO 1 optimizes $\mathcal{C}$ , BO 2 optimizes $\tilde{S}$ and BO 3 optimizes $\mathcal{E}$

| N_R | BO 1 Memory Capacity | BO 1 Average Spike Count | BO 1 RMSE | BO 2 Memory Capacity | BO 2 Average Spike Count | BO 2 RMSE | BO 3 Memory Capacity | BO 3 Average Spike Count | BO 3 RMSE |
|---|---|---|---|---|---|---|---|---|---|
| 100 | $3.56 \pm 0.36$ | $1354.35 \pm 108.96$ | $0.617 \pm 0.019$ | $3.02 \pm 0.52$ | $1101.25 \pm 75.93$ | $0.684 \pm 0.026$ | $3.45 \pm 0.3$ | $1207.52 \pm 67.26$ | $0.654 \pm 0.0207$ |
| 200 | $4.12 \pm 0.39$ | $1443.59 \pm 127.23$ | $0.587 \pm 0.02$ | $3.89 \pm 0.48$ | $1207.35 \pm 118.57$ | $0.639 \pm 0.027$ | $4.01 \pm 0.33$ | $1302.47 \pm 96.05$ | $0.613 \pm 0.0218$ |
| 300 | $5.26 \pm 0.37$ | $1499.62 \pm 141.73$ | $0.503 \pm 0.027$ | $4.44 \pm 0.55$ | $1257.26 \pm 1257.26$ | $0.558 \pm 0.034$ | $5.15 \pm 0.34$ | $1335.81 \pm 168.02$ | $0.531 \pm 0.0287$ |
| 400 | $6.37 \pm 0.41$ | $1528.73 \pm 228.87$ | $0.459 \pm 0.033$ | $5.87 \pm 0.49$ | $1304.35 \pm 126.18$ | $0.467 \pm 0.04$ | $6.05 \pm 0.35$ | $1415.32 \pm 213.49$ | $0.482 \pm 0.0347$ |
| 500 | $7.25 \pm 0.45$ | $1601.27 \pm 277.97$ | $0.389 \pm 0.036$ | $6.25 \pm 0.45$ | $1365.35 \pm 137.04$ | $0.421 \pm 0.043$ | $6.87 \pm 0.36$ | $1507.29 \pm 219.58$ | $0.411 \pm 0.0377$ |
| 1000 | $10.12 \pm 0.43$ | $1868.14 \pm 301.17$ | $0.316 \pm 0.04$ | $7.41 \pm 0.51$ | $1563.25 \pm 146.76$ | $0.396 \pm 0.047$ | $9.03 \pm 0.37$ | $1699.27 \pm 275.79$ | $0.332 \pm 0.0417$ |
| 2000 | $11.84 \pm 0.48$ | $2105.95 \pm 331.54$ | $0.293 \pm 0.045$ | $8.02 \pm 0.5$ | $1854.35 \pm 150.28$ | $0.351 \pm 0.052$ | $11.44 \pm 0.39$ | $2014.12 \pm 280.03$ | $0.301 \pm 0.0467$ |
| 3000 | $13.71 \pm 0.51$ | $2408.35 \pm 348.26$ | $0.258 \pm 0.042$ | $8.94 \pm 0.53$ | $2195.82 \pm 179.75$ | $0.335 \pm 0.058$ | $13.87 \pm 0.4$ | $2236.59 \pm 281.05$ | $0.241 \pm 0.0482$ |
| 4000 | $14.15 \pm 0.52$ | $2951.56 \pm 352.66$ | $0.242 \pm 0.063$ | $9.55 \pm 0.55$ | $2445.31 \pm 217.73$ | $0.326 \pm 0.07$ | $14.63 \pm 0.41$ | $2546.25 \pm 289.81$ | $0.227 \pm 0.0649$ |
| 5000 | $14.45 \pm 0.54$ | $3784.44 \pm 353.51$ | $0.203 \pm 0.064$ | $9.96 \pm 0.58$ | $2684.59 \pm 234.63$ | $0.302 \pm 0.071$ | $15.12 \pm 0.42$ | $2898.27 \pm 307.14$ | $0.195 \pm 0.0655$ |

### A.3.2 Convergence Analysis

We compare the convergence analysis of the three Bayesian Optimization techniques and the results are shown in Fig. 6. Each of the experiments was repeated five times and the mean and variance of the observations are shown in the Figure. It is to be noted here that since we define the BO as a minimization principle, we minimize $\frac{1}{\mathcal{C}}$, $\tilde{S}$ and $\frac{1}{\mathcal{E}}$.

### A.4 Comparing Bayesian Optimization Objective Functions

We show the results of Bayesian Optimization results for the three cases we are considering in this paper for both the classification and prediction problems. The results for the classification problem are shown in Table 7. We tabulate the memory capacity, the average spike count and the observed accuracy for the three BO cases. Similarly, the results for the prediction problem are shown in Table 8. In that case, we tabulate the memory capacity, the average spike count and the observed NRMSE for the three BO cases. We rerun each of the experiments 5 times and report the mean and standard deviation of the results obtained.

### A.5 Comparing the Generalizability

We observed that increasing the neuronal heterogeneity increases the memory capacity of the network. However, this increment in the memory capacity might lead to a model which overfits the training data. However, the heterogeneous STDP model with varying synaptic dynamics gives rise to a heavy-tailed Feller process. Recent works Simsekli et al. (2020), Chakraborty & Mukhopadhyay (2021) show that the Hausdorff dimension of the trajectories of the sample paths of the learning algorithm can control the generalization error. This is intimately linked to the tail behavior of the driving process. The authors showed that heavier-tailed processes achieve better generalization. Thus, the tail index of the process can be used as a notion of capacity metric that estimates the generalization error, which does not necessarily grow with the number of parameters. The authors discuss that the stochastic process for the synaptic weights behaves like a Lévy motion around a local point. Because of this locally regular behavior, the Hausdorff dimension can be bounded by the Blumenthal-Getoor (BG) index (Blumenthal & Getoor, 1960), which depends on the tail behavior of the Lévy process. Thus, we can use the BG index as a bound for the Hausdorff dimension of the trajectories from the STDP

Table 9: Table showing the Ablation Study for the comparison of the Generalizability of heterogeneous networks

| | BG Index | Training Accuracy (A) | Testing Accuracy (B) | Generalization Error (|A-B|) |
|---|---|---|---|---|
| Hom LIF Hom STDP | 1.522 | 87.33 | 73.58 | 13.75 |
| Hom LIF Het STDP | 1.438 | 85.31 | 74.03 | 11.28 |
| Het LIF Hom STDP | 1.835 | 95.29 | 78.87 | 16.42 |
| Het LIF Het STDP | 1.711 | 94.32 | 80.49 | 13.83 |

Table 10: Table showing results with limited training data

| Percentange Training Data | Train Accuracy (A) | Test Accuracy (B) | Generalization Error |A-B| | Train Accuracy (A) | Test Accuracy (B) | Generalization Error |A-B| |
|---|---|---|---|---|---|---|
| | Heterogeneous LIF, Heterogeneous STDP | | | Homogeneous LIF, Homogeneous STDP | | |
| 100 | 94.32 | 80.49 | 13.83 | 87.33 | 73.58 | 13.75 |
| 90 | 94.89 | 78.34 | 16.55 | 87.83 | 67.84 | 19.99 |
| 80 | 95.47 | 76.72 | 18.75 | 88.86 | 65.86 | 23 |
| 70 | 96.15 | 74.92 | 21.23 | 89.95 | 62.19 | 27.76 |
| 60 | 96.92 | 70.34 | 26.58 | 91.58 | 61.25 | 30.33 |
| 50 | 97.69 | 69.44 | 28.25 | 94.38 | 59.51 | 34.87 |
| 40 | 98.21 | 63.76 | 34.45 | 96.85 | 55.93 | 40.92 |
| 30 | 98.43 | 54.01 | 44.42 | 98.43 | 45.86 | 52.57 |
| 20 | 99.43 | 43.87 | 55.56 | 99.49 | 42.68 | 56.81 |
| 10 | 100 | 31.43 | 68.57 | 100 | 30.18 | 69.82 |
| 5 | 100 | 15.32 | 84.68 | 100 | 14.38 | 85.62 |
| | Heterogeneous LIF, Homogeneous STDP | | | Homogeneous LIF, Heterogeneous STDP | | |
| 100 | 97.29 | 78.87 | 18.42 | 86.31 | 74.03 | 12.28 |
| 90 | 97.41 | 77.48 | 19.93 | 86.94 | 68.59 | 18.35 |
| 80 | 97.65 | 76.32 | 21.33 | 87.75 | 67.58 | 20.17 |
| 70 | 97.95 | 74.03 | 23.92 | 88.17 | 65.25 | 22.92 |
| 60 | 98.03 | 71.16 | 26.87 | 89.52 | 63.11 | 26.41 |
| 50 | 98.43 | 68.48 | 29.95 | 90.48 | 60.86 | 29.62 |
| 40 | 98.79 | 61.93 | 36.86 | 93.15 | 57.31 | 35.84 |
| 30 | 99.56 | 51.68 | 47.88 | 96.34 | 48.41 | 47.93 |
| 20 | 100 | 44.52 | 55.48 | 98.43 | 43.59 | 54.84 |
| 10 | 100 | 30.68 | 69.32 | 99.56 | 32.57 | 66.99 |
| 5 | 100 | 14.15 | 85.85 | 100 | 18.48 | 81.52 |

learning process. Now, as the Hausdorff dimension is a measure of the generalization error and is also controlled by the tail behavior of the process, heavier tails imply less generalization error. In this paper, we empirically study the generalization ability of the HRSNN network using the BG index as a metric. We did the experiments on the 4 ablation study models for the classification task on the SHD dataset, and the results are reported in Table 9. From the table, we see that the heterogeneity in STDP improves the generalization error the most, while the heterogeneity in the LIF neurons increases the training and testing accuracies.

## A.6 RESULTS ON LIMITED TRAINING DATA

We have trained the models with limited training data. We observe that the HRSNN model with heterogeneous LIF and STDP dynamics not only has better testing accuracy but also shows better generalization behavior when compared to other homogeneous RSNN or the other ablation heterogeneous models (with heterogeneity in only one of them). Also, we see that the HRSNN model with heterogeneous STDP shows distinctly better generalization ability than the generalization ability of

| | SHD | | | SSC | | | CIFAR10 DVS | | |
|---|---|---|---|---|---|---|---|---|---|
| | Training Accuracy (A) | Testing Accuracy (B) | Generalization Error $|A-B|$ | Training Accuracy (A) | Testing Accuracy (B) | Generalization Error $|A-B|$ | Training Accuracy (A) | Testing Accuracy (B) | Generalization Error $|A-B|$ |
| Hom LIF Hom STDP | $86.92 \pm 1.35$ | $72.89 \pm 1.85$ | $14.03 \pm 1.67$ | $74.69 \pm 1.72$ | $47.94 \pm 1.94$ | $26.75 \pm 1.42$ | $82.41 \pm 1.8$ | $65.33 \pm 3.41$ | $17.08 \pm 1.35$ |
| Hom LIF Het STDP | $85.76 \pm 1.27$ | $73.91 \pm 1.49$ | $11.85 \pm 1.25$ | $76.79 \pm 1.58$ | $52.96 \pm 1.73$ | $23.86 \pm 1.29$ | $83.48 \pm 1.52$ | $67.06 \pm 2.97$ | $16.42 \pm 1.24$ |
| Het LIF Hom STDP | $95.29 \pm 1.16$ | $78.36 \pm 1.42$ | $16.93 \pm 1.13$ | $84.26 \pm 1.33$ | $55.11 \pm 1.65$ | $29.15 \pm 1.12$ | $86.93 \pm 1.79$ | $68.37 \pm 3.05$ | $18.56 \pm 1.42$ |
| Het LIF Het STDP | $94.07 \pm 1.03$ | $80.01 \pm 1.13$ | $14.06 \pm 1.02$ | $86.41 \pm 1.49$ | $59.28 \pm 1.35$ | $27.13 \pm 0.97$ | $87.49 \pm 1.76$ | $70.54 \pm 1.82$ | $16.95 \pm 1.38$ |

HRSNN with heterogeneous LIF neurons. On the other hand, the latter showcases significantly higher training and testing accuracy compared to the former model. This can be interpreted as follows: since heterogeneous LIF dynamics increase the memory capacity, it leads to an overfitting of the data. Heterogeneous STDP dynamics help in obtaining more generalizable solutions from this. Each has its own downsides; however, using HRSNN with both heterogeneous LIF and STDP dynamics shows better performance and generalization abilities, as seen from Table 10.

## A.7 Further Evaluations

In Section B, we argued that as the heterogeneity in the neuronal parameters increases, the covariance decreases; hence the neurons become less correlated. In this section, we give empirical results to support the theory. We tested the model on more complex datasets - (i) The Spiking Heidelberg Digits (SHD) dataset (ii) the Spiking Speech Command (SSC) dataset are both audio-based classification datasets for which input spikes and output labels are provided Cramer et al. (2020) and (iii) CIFAR10 DVS dataset Li et al. (2017).

- **Impact of Heterogeneity on Covariance:** We plot the covariance matrices for different levels of heterogeneity $\mathcal{J}$ (Eq. 46) for a small network with 50 neurons. The covariance matrix is calculated by taking the average neuronal states before the appearance of the first spike in the final layer. We see that as the heterogeneity in the neuronal parameters increases, the correlation between the neurons decreases. The results are shown in Fig. 7

- **Impact of Heterogeneity on Principal Components:** From the covariance plots, we see that increasing $\mathcal{J}$. reduces the correlation between neurons. We also plot the probability density functions of the eigenvalues of the covariance matrix of the neurons with increasing heterogeneity in the neuronal parameters. We see that with higher heterogeneity in the neuronal parameters $\mathcal{J}$, the distribution of the eigenvalues of the covariance becomes flatter. This signifies that the covariance matrix has a lower variance for higher $\mathcal{J}$. A flatter distribution also indicates that a larger number of principal components are active. This supports our hypothesis that heterogeneity in the neuronal parameters increases the number of principal components and helps increase the model's memory capacity. The result is shown in this Fig. 8

- **Impact of Heterogeneity in STDP on Firing Rate:** We plot the mean firing rate of the neurons for the four types of HRSNNs and MRSNN with homogeneous LIF and STDP dynamics. We plot the results for a smaller network with 100 neurons and a Poisson input process. The MRSNN model shows a much higher firing rate, especially at a higher frequency, demonstrating that MRSNN requires significantly more spikes than the HRSNN model. The result is shown in Fig. 9

- **Coupling Strength:** We note here that in this paper, we use (homogeneous or heterogeneous) STDP to learn the synaptic conductance connecting various neurons in the SNN. Therefore, we do not control the synaptic coupling strength as independent variables and hence, cannot perform control experiments with various extents of coupling strength. An interesting future extension of the results will be quantifying the coupling strength for HRSNN with heterogeneity in LIF and STDP dynamics. We can leverage McKenzie et al.McKenzie et al. (2021), where the authors proposed statistical tools to estimate synaptic coupling dynamics from spike-spike correlations.

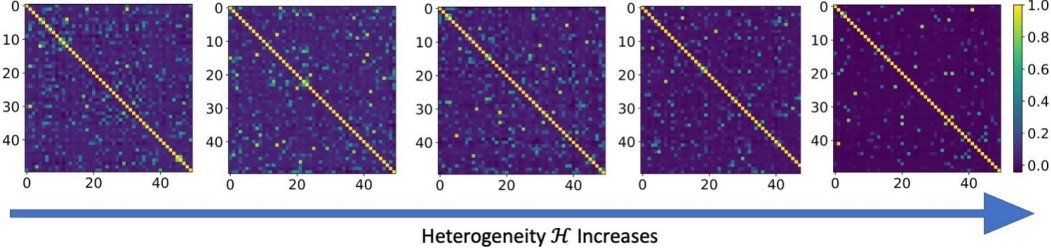

Figure 7: Figure showing as the heterogeneity in the neuronal parameters increases, the covariance between the neurons decreases

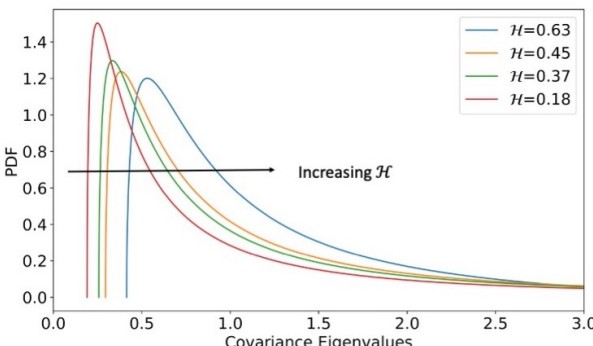

Figure 8: With higher heterogeneity in the neuronal parameters $\mathcal{H}$, the distribution of the eigenvalues of the covariance becomes flatter. This signifies that the covariance matrix has a lower variance for higher $\mathcal{H}$. A flatter distribution also signifies a greater number of principal components are active, which supports our hypothesis that heterogeneity in the neuronal parameters increases the number of principal components and helps in increasing the memory capacity of the model.

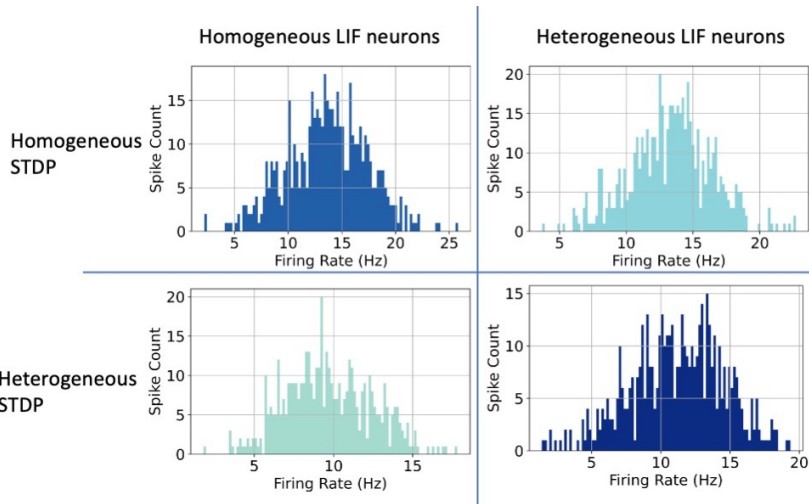

Figure 9: Figure showing the histograms of the firing rates for the four kinds of heterogeneous RSNN with homogeneous /heterogeneous neurons and synapses

# B  SUPPLEMENTARY SECTION B

## B.1  APPROXIMATIONS AND ASSUMPTIONS

We make several approximations and assumptions for this section's theoretical analysis of the heterogeneous RSNN networks. Firstly, it must be noted that in this paper, the analytical relations are derived by taking the heterogeneity individually. i.e., when we consider heterogeneity in the neuronal parameters, we consider homogeneous STDP dynamics and vice-versa. In addition to this, we assume *diffusion approximation*. That is, if a neuron receives Poissonian uncorrelated input spike trains and the contribution of a single synaptic connection is small compared to the distance between reset and threshold $w \ll (V_\Theta - V_0)$, the random input can be approximated by Gaussian white noise with mean $\mu$ and noise intensity $\sigma^2$. This approximation does not hold if the network features highly correlated activity or receives strong external input common to many neurons. Also, we assume a fast/slow synaptic regime in which the synaptic time constant $\tau_s$ is much shorter/longer than the membrane time constant $\tau_m$. In this work, we consider a mean-field approximation of the HRSNN network with heterogeneity in the parameters of the LIF neurons and the STDP dynamics independently.

## B.2  MEAN-FIELD REDUCTION MODEL OF HRSNN

In this section, we model the HRSNN network using heterogeneity in only the LIF neuron parameters. Following the works of Ly et al.Ly (2015), we can write the equations for the excitatory neurons indexed by $j \in \{1, 2, \ldots, N_e\}$ are:

$$\tau_m \frac{dv_j}{dt} = -v_j - g_{ie}(t)(v_j - \mathcal{E}_I) - g_{ee}(t)(v_j - \mathcal{E}_E) + \sigma_E \eta_j(t) \tag{11}$$

$$v_j(t^*) \geq \theta_j(\text{ refractory period }) \Rightarrow v_j(t^* + \tau_{ref}) = 0 \tag{12}$$

$$\tau_n \frac{d\eta_j}{dt} = -\eta_j + \sqrt{\tau_n \xi_j(t)} \tag{13}$$

$$g_{ee}(t) = q_j \frac{\gamma_{ee}}{p_{ee}N_e} \sum_{j' \in \{\text{ presyn E cells}\}} G_{j'}(t) \tag{14}$$

$$g_{ei}(t) = \frac{\gamma_{ei}}{p_{ei}N_i} \sum_{k' \in \{\text{ presyn I cells}\}} G_{k'}(t) \tag{15}$$

$$\tau_d \frac{dG_j}{dt} = -G_j + A_j \tag{16}$$

$$\tau_r \frac{dA_j}{dt} = -A_j + \tau_r \alpha \sum_l \delta(t - t_l) \tag{17}$$

where the inhibitory and excitatory reversal potentials are $\mathcal{E}_I$, and $\mathcal{E}_E$, respectively, with $\mathcal{E}_I < 0 < \mathcal{E}_E$. $\xi_j(t)$ are uncorrelated white noise processes, $p_{xy}$ is the proportion of neuron type $y$ (randomly chosen) that provides presynaptic input to neuron type $x$ $(x, y \in \{e, i\})$. The second line in the equations describes the refractory period at spike time $t^*$. When the neuron's voltage crosses threshold $\theta_j$, the neuron goes into a refractory period for $\tau_{ref}$ where the voltage is undefined, after which we set the neuron's voltage to 0. In the last equation, $t_l$ denotes the spike times of the $j$ th excitatory neuron. Now, for the mean-field analysis, we use $q_{ji}$ to model the synaptic heterogeneity between the pre-and post-synaptic neurons by modulating the synaptic conductance for both the excitatory and inhibitory neurons.

We note here the numerical assumptions for the mean-field analysis:

1. finite size effects are negligible (N e/i $\gg$ 1 )

2. the firing rate of presynaptic neurons is governed by a Poisson process

3. the population firing rate averaged over $q$ and $\tau_m$ is a good approximation to the average presynaptic input rate and

4. a single p.d.f. function is sufficient to describe the population behavior) (finite $N$)

Similarly, for the inhibitory neurons indexed by $k \in \{1, 2, \ldots, N_i\}$, the equations are:

$$\tau_m \frac{dv_k}{dt} = -v_k - g_{ii}(t)(v_k - \mathcal{E}_I) - g_{ei}(t)(v_k - \mathcal{E}_E) + \sigma_I \eta_k(t) \tag{18}$$

$$v_k(t^*) \geq 1 (\text{ refractory period }) \Rightarrow v_j(t^* + \tau_{ref}) = 0 \tag{19}$$

$$\tau_n \frac{d\eta_k}{dt} = -\eta_k + \sqrt{\tau_n \xi_k(t)} \tag{20}$$

$$g_{ie}(t) = q_j \frac{\gamma_{ie}}{p_{ie} N_e} \sum_{k' \in \{ \text{ presyn I cells}\}} G_{k'}(t) \tag{21}$$

$$g_{ii}(t) = \frac{\gamma_{ii}}{p_{ii} N_i} \sum_{k' \in \{\text{presyn I cells}\}} G_{k'}(t) \tag{22}$$

$$\tau_d \frac{dG_k}{dt} = -G_k + A_k \tag{23}$$

$$\tau_r \frac{dA_k}{dt} = -A_k + \tau_r \alpha \sum_l \delta(t - t_l) \tag{24}$$

Please refer to the paper by Ly et al. Ly (2015) for details regarding the equations. Since the recurrent coupled stochastic network is difficult to describe theoretically, we use population density methods, where an equation determines the probability of a neuron being in a particular state. The variables in the populations are determined using distribution functions. The two forms of heterogeneity introduce a large number of dimensions. For simplicity, one can track a family of probability density functions for each $(q_j, \tau_j)$ pair for each neuron. The subsequent equations are a good approximation to the HRSNN network with the following assumptions: (i) finite size effects are negligible $(N_{e/i} \gg 1)$ (ii) the firing rate of presynaptic neurons is governed by a Poisson process (iii) the population firing rate averaged over $q$ and $\tau_m$ is a good approximation to the average presynaptic input rate (iv) a single p.d.f. function is sufficient to describe the population behavior, and the heterogeneity is driven by $(q_j, \tau_m, j)$ For each pair of values $(q_j, \tau_m, j)$, the probability density function $\rho$ is defined by:

$$\int_\Omega \rho(v_E, \mathbf{w}_E, v_I, \mathbf{w}_I, t) \, dv_E d\mathbf{w}_E dv_I d\mathbf{w}_I = \Pr((v_E(t), \mathbf{w}_E(t), v_I(t), \mathbf{w}_I(t)) \quad \in \Omega) \tag{25}$$

where $\mathbf{w}_X$ denotes the other states variables of the corresponding neuron type $X \in \{E, I\}$, consisting of conductance, colored noise: $\mathbf{w}_X = (g_X, a_X, \eta_X)$. The evolution of the p.d.f.'s is governed by a continuity equation and boundary conditions:

$$\frac{\partial \rho}{\partial t} = -\nabla \cdot \mathbf{J} \tag{26}$$

$$\mathbf{J} := (J_{v_E}, J_{g_E}, J_{a_E}, J_{\eta_E}, J_{v_I}, J_{g_I}, J_{a_I}, J_{\eta_I}) \tag{27}$$

$$J_{v_E} := -\frac{1}{\tau_m} [v_E + q\gamma_{ei} g_I (v_E - \mathcal{E}_I) + q\gamma_{ee} g_E (v_E - \mathcal{E}_E) + \sigma_E \eta_E] \rho \tag{28}$$

$$J_{v_I} := -\frac{1}{\tau_m} [v_I + \gamma_{ii} g_I (v_I - \mathcal{E}_I) + \gamma_{ie} g_E (v_I - \mathcal{E}_E) + \sigma_I \eta_I] \rho \tag{29}$$

$$J_{g_X} := -\frac{1}{\tau_d} [g_X - a_X] \rho \tag{30}$$

$$J_{a_X} := -\frac{a_X}{\tau_r} + v_X(t) \int_{a_X - \alpha_X}^{a_X} \rho(\ldots, a'_X, \ldots) \, da'_X \tag{31}$$

$$J_{\eta_X} := -\frac{1}{\tau_n} \eta_X \rho + \frac{1}{\tau_n} \frac{\partial^2 \rho}{\partial \eta_X^2} \tag{32}$$

$$v_X(t) := \iiint \frac{1}{\tau_m} J_{v_X} \, d\mathbf{w}_X dq d\tau_m \tag{33}$$

$$J_{\mathbf{w}_X} \mid \partial \mathbf{w}_X = 0 \tag{34}$$

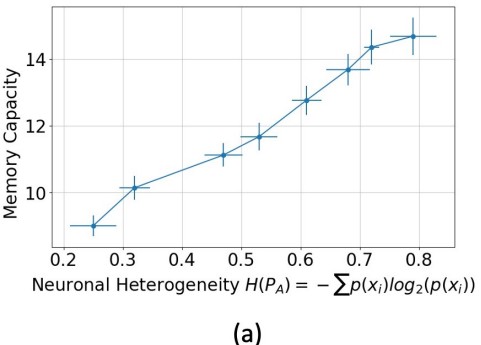 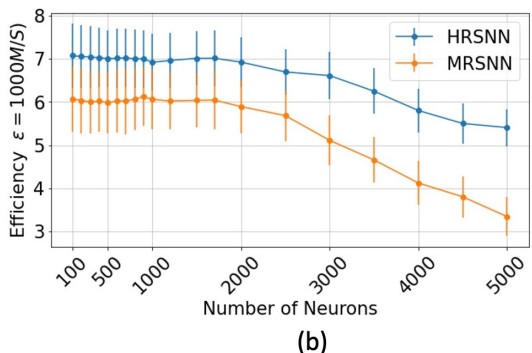

(a)                                                  (b)

Figure 10: (a)Figure showing the variation of memory capacity with neuronal heterogeneity (b) Figure showing the variation of efficiency $\mathcal{E}$ with number of neurons $N_{\mathcal{R}}$

The definitions of $g_{XY}$ in the LIF neuron equations defined above result in a total conductance of $\gamma_{XY} g_Y$ on average.

We describe an insightful analytic reduction that captures how the range of excitatory firing rates changes in different regimes. We focus on only the excitatory neurons, which have fewer state variables if the inhibitory population is ignored or assumed to be known.

Let us denote the approximate excitatory firing rate(s) $v_E$ as $r$. The deterministic firing rate of the equation

$$\tau_m \frac{dv_E}{dt} = -v_E - q\tilde{g_I}\left(v_E - \mathcal{E}_I\right) - q\tilde{g_E}\left(v_E - \mathcal{E}_E\right) + \tilde{\eta_E} \tag{35}$$

is given by

$$r_0\left(q, \tau_m; \tilde{\mathbf{w}}_E\right) = \begin{cases} 0, & \text{if } \frac{q(\tilde{g_E}\mathcal{E}_E + \tilde{g_I}\mathcal{E}_I) + \tilde{\eta_E}}{1 + q(\tilde{g_E} + \tilde{g_I})} \leq \theta \\ \frac{1 + q(\tilde{g_E} + \tilde{g_I})}{\tau_m\left(g_E^* \mathcal{E}_E + \tilde{g_I}\mathcal{E}_I\right) + \tilde{\eta_E}} & \text{if } \frac{q(\tilde{g_E}\mathcal{E}_E + \tilde{g_I}\mathcal{E}_I) + \tilde{\eta_E}}{1 + q(\tilde{g_E} + \tilde{g_I})} > \theta \end{cases} \tag{36}$$

We define: $\tilde{g_E} := \gamma_{ee} g_E, \tilde{g_I} := \gamma_{ei} g_I, \tilde{\eta_E} := \sigma_E \eta_E$. Finally, the given state variables are integrated against their marginal density to get:

$$r(q, \theta) = \mathbb{E}\left[\frac{r_0}{1 + r_0 \tau_{ref}}\right] = \int \frac{r_0}{1 + r_0 \tau_{ref}} \tilde{\rho}\left(\tilde{g_E}, \tilde{g_I}, \tilde{\eta_E}\right) d\tilde{\mathbf{w}}_E \tag{37}$$

There is a slight abuse of notation because the auxiliary variables $a_X$ effect the conductances but are not written in the previous equation; the emphasis is on how $(\tilde{g_E}, \tilde{g_I}, \tilde{\eta_E})$ directly effects $r$. Since the external noise is applied indiscriminately, $\tilde{\eta_E}$ is independent of the other variables and the marginal density factors into:

$$\tilde{\rho}\left(\tilde{g_E}, \tilde{g_I}, \tilde{\eta_E}\right) = \tilde{\rho}\left(\tilde{g_E}, \tilde{g_I}\right) \frac{e^{-(\tilde{\eta_E}/\sigma_E)^2}}{\sigma_E \sqrt{\pi}} \tag{38}$$

However, $\tilde{\rho}\left(\tilde{g_E}, \tilde{g_I}\right)$ is still not analytically tractable, leading us to rely on Monte Carlo simulations to numerically estimate $\tilde{\rho}\left(\tilde{g_E}, \tilde{g_I}\right)$.

It must be noted here that this is a reduction model for the HRSNN network with many simplifying assumptions. It is not a complete mean-field derivation of the HRSNN model with heterogeneous LIF neurons, and heterogeneous STDP dynamics is a fascinating research question but beyond the scope of this paper.

## B.3 ANALYTICAL RESULTS

**Analytical Results of Memory Capacity** Neuroscience networks of spiking neurons are increasingly used to understand mechanisms underlying phenomena observed in electrophysiological recordings.

There are two complementary strategies for studying such a recurrent network of spiking neurons - (a) numerical simulations and (b) analytical methods using mean field models. With numerical simulations, we can simulate any network model without any approximation. However, this method typically works in high dimensional parameter space and is, thus, hard to interpret. Also, it is generally hard to characterize parameter regions where specific behaviors are found using numerical simulations. On the other hand, with analytical calculations, we obtain deeper insights into mechanisms underlying specific behaviors and can obtain critical parameters that control specific behaviors. So, now, we analytically study the variance of the estimated memory capacity with the change in the heterogeneity of neuronal parameters. We plot the change in the estimated memory capacity $\mathcal{C}$, calculated using Eq. 3. We plot this with respect to the neuronal heterogeneity $\mathcal{H}$, measured using the entropy of the neuronal parameters for the HRSNN model. The result is plotted in Fig. 10(a). We use a HRSNN model with $N_{\mathcal{R}}$ = 1000 and sequences of 4,000 random inputs chosen from $\mathcal{U}[-1;1]$. We see that, as predicted, the memory capacity of the model increases linearly with the increase in heterogeneity within the limits of the application, as proved in Theorem 1. The error bars in Fig. 10(a) represent the standard deviation of the observations.

**Analytical Study of Spike Efficiency** We calculate the average firing rate of the heterogeneous spiking neural network for the prediction task during inference, and the results are shown in Fig 10(b). Using heterogeneity in the STDP parameters reduces the average number of spiking activations while keeping the memory capacity almost equal. This result shows that Heterogeneous STDP leads to sparse activation of neurons, as proved in Theorem 2.

**Comparison with Neuroscience Works:** We compare the analytical results obtained with some of the standard recurrent LIF network models in the literature. Brunel et al.Brunel (2000) analytically study the dynamics of sparsely connected a network of sparsely connected excitatory and inhibitory integrate-and-fire neurons. The authors showed the existence of a diverse set of states, including synchronous states in which neurons fire regularly; asynchronous states with stationary global activity and very irregular individual cell activity; and states in which the global activity oscillates but individual cells fire irregularly, typically at rates lower than the global oscillation frequency. In this paper, we use heterogeneity in the LIF neurons. This leads to a diverse set of states for the neurons, which consequently helps orthogonalize the state space dynamics to increase the information stored in the memory of the network.

Deneve et al. Denève & Machens (2016) discussed the inefficiency of irregular Poisson rate encoding in the brain. The authors argue that the Poisson point process, which we use to model the spike firing rate, is extremely inefficient as it exponentially increases the number of spikes required to convey information. The authors further discuss that a continuum exists between loosely balanced and tightly balanced spike-coding networks in neuroscience. Though loosely balanced networks are inefficient, they are cheap in terms of the number of connections per neuron and structure (Boerlin et al., 2013; Boerlin & Denève, 2011; Bourdoukan et al., 2012). On the other hand, tightly-balanced spike-coding networks are highly efficient but extremely structured, dense connections that STDP rules must constantly maintain. For the HRSNN model, since we are engineering an artificial spiking neural network model, our network is highly structured and constantly updated using the heterogeneous STDP rules. Thus, we might say that the HRSNN model is a tightly-coupled network that helps in an efficient transfer of information. This hypothesis is supported by the results shown in Table 2, where the HRSNN model shows a higher performance using a lesser number of spikes.

### B.4 MEMORY CAPACITY

Let $x(t) \in U$ (where $-\infty < t < +\infty$ and $U \subset \mathbb{R}$ is a compact interval) be a single-channel stationary input signal. Assume that we have an RSNN, specified by its internal weight matrix $\mathbf{W}$, its input weight vector $\mathbf{w}^{\text{in}}$ and the unit output functions $\mathbf{f}, \mathbf{f}^{\text{out}}$. The network receives $x(t)$ at its input unit. For a given delay $\tau$ and an output unit $y_\tau$ with connection weight vector $\mathbf{w}_\tau^{\text{out}}$ we consider the determination coefficient

$$d\left[\mathbf{w}_\tau^{\text{out}}\right](x(t-\tau), y_\tau(t)) =$$

$$= d\left(x(t-\tau), \mathbf{w}_\tau^{\text{out}} \begin{pmatrix} x(t) \\ \mathbf{r}(t) \end{pmatrix}\right)$$

$$= \frac{\text{Cov}^2(x(t-\tau), y_\tau(t))}{\sigma^2(x(t))\sigma^2(y_\tau(t))}$$

where Cov denotes covariance and $\sigma^2$ variance. The $\tau$-delay Memory capacity of the network is defined by $C_\tau = \max_{\mathbf{w}_\tau^{\text{out}}} d\left[\mathbf{w}_\tau^{\text{out}}\right](x(t-\tau), y_\tau(t))$. The Memory capacity of the network is $C = \sum_{\tau=1}^{\infty} C_\tau$.

The determination coefficient of two signals is the squared correlation coefficient. It ranges between 0 and 1 and represents the fraction of variance explainable in one signal by the other. Thus, the Memory capacity measures how much variance of the delayed input signal can be recovered from optimally trained output units, summed over all delays. Note that the output units do not interfere; arbitrarily, many output units $y_\tau$ can be attached to the same network.

The performance of the heterogeneous network model derives from its ability to retain the memory of previous inputs. To quantify the relationship between the recurrent layer dynamics and the memory capacity, we note that the extraction of information from the recurrent layer is made through a linear combination of the neurons' states. Hence, more linearly independent neurons would offer more variable states and, thus, more extended memory.

For reservoir computing (RC), Jaeger et al. Jaeger (2002) shows that $C$ is bounded by the reservoir network size of the linear RC with the identity activation function and the independent and identically distributed (i.i.d.) model input. Memory capacity ($C$) is used to quantify the memory of RSNN. Such memory capacity measures the ability of RC to reconstruct precisely the past information of the model input. Also, the network's structural properties can greatly impact the $C$ of the linear RC. Now, the question arises what is the need to maximize the memory capacity of the network? The $C$ normally serves as a global index to quantify the memory property of the network. To comprehensively examine the memory property deeply, the local measurement of its memory property is indispensable. Thus, maximizing the $C$ acts as an estimator for better prediction results of the trained network.

Since the first-order approximation of the model is linear, the heterogeneity between state variables depends on all the eigenvalues of the adjacency matrix, with a larger mean eigenvalue meaning higher heterogeneity. Hence we can use the eigenvalues $\{\lambda_i\}$ of the weight matrix $W$ to quantify approximately how fast the input decays in the recurrent layer. In other words, the eigenvalues of $W$ should be related to the memory capacity of the heterogeneous neural network model. Indeed, we find that the average eigenvalue modulus: $\langle|\lambda|\rangle = 1/N_R \sum_{i=1}^{N_R} |\lambda_i|$ strongly correlates with $\mathcal{H}$ and therefore with $C$ as well. Note that, instead of $C$ and $\mathcal{H}$, $\langle|\lambda|\rangle$ is much easier to compute and is solely determined by the recurrent layer network.

The memory capacity reflects the precision with which previous inputs can be recovered. The nonlinearity of the recurrent layer and other far-in-the-past inputs induce noise that complicates recovery. Thus, similar to the analysis done by Aceituno et al.Aceituno et al. (2020) for Echo state networks, the variance of the linear part of the recurrent layer is placed to maximize the recoverable information. Thus, the inputs are projected into orthogonal directions of the recurrent layer state space to not add noise to each other. The variance spread across the different dimensions should be evenly distributed within those orthogonal directions, quantified by the neurons' covariance.

We start by noticing that the linear nature of the projection vector $\mathbf{w}_{\text{out}}$ implies that we are treating the system as

$$\mathbf{r}(t) = \sum_{\tau=0}^{\infty} \mathbf{a}_\tau x(t-\tau) + \varepsilon(t) \tag{39}$$

where the vectors $\mathbf{a}_\tau \in \mathbb{R}^{N_R}$ correspond to the linearly extractable effect of $x(t-\tau)$ onto $\mathbf{r}(t)$ and $\varepsilon(t)$ is the nonlinear contribution of all the inputs onto the state of $\mathbf{r}(t)$.

Previous works have shown that linear recurrent layers have more extended memory, but nonlinearity is needed to perform interesting computations. Here we show that for a fixed ratio of the nonlinearity, greater heterogeneity leads to a lesser neuronal correlation, leading to a higher memory capacity.

To maintain this trade-off between linear and non-linear behavior, we will assume that linear and non-linear strengths distribution is fixed. This can be achieved if we impose that the probabilities of the neuron states do not change, meaning that the mean, variance, and other moments of the neuron outputs are unchanged; hence, the strength of the non-linear effects is unchanged. A first constraint can also be obtained from the maintained strength of the linear side of Eq.39

$$\text{Var}\left(\sum_{\tau=1}^{\infty} \mathbf{a}_\tau x(t-\tau)\right) = c \tag{40}$$

where $c$ is a constant.

**Lemma 3.1.1:** *The state of the neuron can be written as follows:*

$$r_i(t) = \sum_{k=0}^{N_R} \sum_{n=1}^{N_R} \lambda_n^k \left\langle v_n^{-1}, \mathbf{w}^{\text{in}} \right\rangle (v_n)_i \, x(t-k) \tag{41}$$

*where* $\mathbf{v}_n, \mathbf{v}_n^{-1} \in \mathbf{V}$ *are, respectively, the left and right eigenvectors of* $\mathbf{W}$, *and* $\lambda_n^k \in \lambda$ *belongs to the diagonal matrix containing the eigenvalues of* $\mathbf{W}$; $\mathbf{a}_i = [a_{i,0}, a_{i,1}, \ldots]$ *represents the coefficients that the previous inputs* $\mathbf{x}_t = [x(t), x(t-1), \ldots]$ *have on* $r_i(t)$.

**Proof:** We build on the work of Aceituno et al.Aceituno et al. (2020) where they showed that higher heterogeneity among the neuronal states implies higher memory capacity. Here we aim to show that as the number of neurons $N_R$ in the recurrent layer decreases, heterogeneity increases the spectral radius. More formally, the spectral radius $|\lambda_n|$ is directly proportional to $\mathcal{H}$ as $N_R$ decreases. We express the state of a neuron $r_i(t)$ as

$$r_i(t) = \sum_{k=0}^{\infty} \left( W^k \mathbf{w}^{\text{in}} \right)_i x(t-k) = \sum_{k=0}^{\infty} a_{i,k} x(t-k) = \langle \mathbf{a}_i, \mathbf{x}_t \rangle \tag{42}$$

where the vector $\mathbf{a}_i = [a_{i,0}, a_{i,1}, \ldots]$ represents the coefficients that the previous inputs $\mathbf{x}_t = [x(t), x(t-1), \ldots]$ have on $r_i(t)$. We can then plug this into the covariance between two neurons,

$$\begin{aligned}
\text{Cov}\,(r_i, r_j) &= \lim_{T \to \infty} \frac{1}{T} \sum_{q=t}^{t+T} \langle \mathbf{a}_i, \mathbf{x}_q \rangle \langle \mathbf{a}_j, \mathbf{x}_q \rangle \\
&= \langle \mathbf{a}_i, \mathbf{a}_j \rangle \lim_{T \to \infty} \frac{1}{T} \sum_{q_i=0}^{T} \sum_{q_j=0}^{T} \left\langle \mathbf{x}_{q_i}, \mathbf{x}_{q_j} \right\rangle \\
&= \langle \mathbf{a}_i, \mathbf{a}_j \rangle \lim_{T \to \infty} \frac{1}{T} \sum_{q=0}^{T} \langle \mathbf{x}_q, \mathbf{x}_q \rangle \\
&= \langle \mathbf{a}_i, \mathbf{a}_j \rangle \times \mathbb{E}\left[ x^2(t) \right] \\
&= \langle \mathbf{a}_i, \mathbf{a}_j \rangle
\end{aligned} \tag{43}$$

Now we write $\mathbf{a}_i$ as a function of the eigenvalues of $\mathbf{W}$. Using the eigenvalue decomposition of the weight matrix $\mathbf{W}$, we rewrite the state of the neuron as follows:

$$r_i(t) = \sum_{k=0}^{N_R} \sum_{n=1}^{N_R} \lambda_n^k \left\langle v_n^{-1}, \mathbf{w}^{\text{in}} \right\rangle (v_n)_i \, x(t-k) \tag{44}$$

where $\mathbf{v}_n, \mathbf{v}_n^{-1} \in \mathbf{V}$ are, respectively, the left and right eigenvectors of $\mathbf{W}$, and $\lambda_n^k \in \lambda$ belongs to the diagonal matrix containing the eigenvalues of $\mathbf{W}$; $\mathbf{a}_i = [a_{i,0}, a_{i,1}, \ldots]$ represents the coefficients that the previous inputs $\mathbf{x}_t = [x(t), x(t-1), \ldots]$ have on $r_i(t)$. $\blacksquare$

**Theorem 1:** *If the memory capacity of the HRSNN and MRSNN networks are denoted by* $\mathcal{C}_H$ *and* $\mathcal{C}_M$ *respectively, then,* $\mathcal{C}_H \geq \mathcal{C}_M$, *where the heterogeneity in the neuronal parameters* $\mathcal{H}$ *varies inversely to the correlation among the neuronal states measured as* $\sum_{n=1}^{N_\mathcal{R}} \sum_{m=1}^{N_\mathcal{R}} \text{Cov}^2\left( x_n(t), x_m(t) \right)$ *which in turn varies inversely with* $\mathcal{C}$.

**Proof:** As shown by Aceituno et al.Aceituno et al. (2020), the memory capacity increases when the variance along the projections of the input into the recurrent layer state has higher heterogeneity. This can be expressed in terms of the state space of the recurrent layer. Now, we aim to project the inputs into orthogonal directions of the network state space. Thus, we model the system as

$$\mathbf{r}(t) = \sum_{\tau=1}^{\infty} \mathbf{a}_\tau x(t-\tau) + \varepsilon(t) \tag{45}$$

where the vectors $\mathbf{a}_\tau \in \mathbb{R}^N$ correspond to the linearly extractable effect of $x(t-\tau)$ onto $\mathbf{r}(t)$ and $\varepsilon(t)$ is the nonlinear contribution of all the inputs onto the state of $\mathbf{r}(t)$.

Since our goal is to have a variance as homogeneous as possible along with the directions of $a_\tau$, we need a variance that is as homogeneous along with orthogonal directions, where the vectors $\mathbf{a}_\tau \in \mathbb{R}^N$ correspond to the linearly extractable effect of the input variable $x(t)$ onto the states of the neurons ($\mathbf{r}(t)$). Since the eigenvectors of $\boldsymbol{\Sigma}$ preserve orthogonality across the covariance matrix $\boldsymbol{\Sigma}$, the new variances are given by the eigenvalues of the covariance matrix, $\lambda_n(\boldsymbol{\Sigma})$. Thus, we work on the distribution of the eigenvalues of the covariance matrix. Specifically, we want to show that increasing the heterogeneity in the neuronal membrane time constants decreases the correlation between the neuron states, which decreases the variance of the neuronal states of the eigenvalues, which would increase the memory capacity $\mathcal{C}$. We quantify the heterogeneity using the mean with respect to the square root of the raw variance of the eigenvalues of the covariance matrix given by

$$\mathcal{J} = \frac{\sum_{n=1}^{N_\mathcal{R}} \lambda_n^2(\boldsymbol{\Sigma})}{\left(\sum_{n=1}^{N_\mathcal{R}} \lambda_n(\boldsymbol{\Sigma})\right)^2} \tag{46}$$

where $\lambda_n(\boldsymbol{\Sigma})$ is the $n$th eigenvalue of $\boldsymbol{\Sigma}$. To get an intuition of how this metric reflects the heterogeneity in the neuronal parameters, consider the case of two eigenvalues $\lambda_1, \lambda_2$; when $\lambda_1 = \lambda_2$-very homogeneous – then $\mathcal{J} = \frac{1}{2}$, but when $\lambda_1 > 0, \lambda_2 = 0-$ heterogeneity is more and hence, $\mathcal{J} = 1$. The membrane time constant is given by the product of the membrane resistance $R_m$ and membrane capacitance $C_m$, such that $\tau_m = R_m C_m$. $R_m$ is the inverse of the permeability; the higher the permeability, the lower the resistance, and vice versa. Thus, the lower the time constant, the faster or more rapidly a membrane will respond to a stimulus. The effects of the time constant on propagation velocity will become clear below. Hence, variability in the membrane time constants will lead to variability in the propagation velocity of action potentials.

Now,

$$\left(\sum_{n=1}^{N_\mathcal{R}} \lambda_n(\boldsymbol{\Sigma})\right)^2 = (\mathrm{tr}[\boldsymbol{\Sigma}])^2 = \left(\sum_{n=1}^{N_\mathcal{R}} \mathrm{Var}\left(r_n(t)\right)\right)^2 \tag{47}$$

which is constant by the assumption that the probability distributions of the neuron activities are fixed. Hence we can focus on the value of $\sum_{n=1}^{N_\mathcal{R}} \lambda_n^2(\boldsymbol{\Sigma})$ which is true since

$$\boldsymbol{\Sigma}^k \mathbf{e}_n(\boldsymbol{\Sigma}) = \lambda_n(\boldsymbol{\Sigma}) \boldsymbol{\Sigma}^{k-1} \mathbf{e}_n(\boldsymbol{\Sigma}) = \lambda_n^k(\boldsymbol{\Sigma}) \mathbf{e}_n(\boldsymbol{\Sigma}) \Rightarrow \sum_{n=1}^{N_\mathcal{R}} \lambda_n^2(\boldsymbol{\Sigma}) = \mathrm{tr}\left[\boldsymbol{\Sigma}^2\right] \tag{48}$$

where $\mathbf{e}_n(\boldsymbol{\Sigma})$ and $\lambda_n(\boldsymbol{\Sigma})$ are, resp. the $n$th eigenvector and eigenvalue of $\boldsymbol{\Sigma}$. Hence, we can compute this by decomposing the square of the covariance matrix as follows:

$$\sum_{n=1}^{N_\mathcal{R}} \lambda_n^2(\boldsymbol{\Sigma}) = \sum_{n=1}^{N_\mathcal{R}} \sum_{m=1}^{N_\mathcal{R}} \boldsymbol{\Sigma}_{nm} \boldsymbol{\Sigma}_{mn} = \sum_{n=1}^{N_\mathcal{R}} \sum_{m=1}^{N_\mathcal{R}} \mathrm{Cov}^2\left(x_n(t), x_m(t)\right) \tag{49}$$

where $\boldsymbol{\Sigma}_{ij}$ are the factor matrices obtained using Cholesky decomposition of $\boldsymbol{\Sigma}$. Thus, $\sum_{n=1}^{N_\mathcal{R}} \lambda_n^2(\boldsymbol{\Sigma})$ increases as the neurons become more correlated; hence heterogeneity decreases.

Thus, from Eqs. 46, 49 we can write the heterogeneity as inversely proportional to $\sum_{n=1}^{N_\mathcal{R}} \mathrm{Cov}^2\left(x_n(t), x_m(t)\right)$. We see that increasing the correlations between neuronal states decreases the heterogeneity of the eigenvalues, which would reduce the memory capacity of the model. We show that the determinant of the covariance between neuronal parameters bounds the heterogeneity. Thus, as $\mathcal{H}$ increases → covariance decreases → neurons become less correlated. Aceituno et al.Aceituno et al. (2020) proved that the neuronal state correlation is inversely related to the memory capacity of the network. Hence, we claim that as $\mathcal{H}$ increases, the memory capacity $\mathcal{C}$ also increases. Hence, for HRSNN, with $\mathcal{H} > 0$, $\mathcal{C}_H \geq \mathcal{C}_M$. ∎

## B.5 Spiking Efficiency

In this section, we model the spiking activity using a point process called the multivariate Point process model. A point process is a collection of random points on some underlying mathematical space, such as the real line, the Cartesian plane, or more abstract spaces.

The notion of using point process models, especially the interactive Hawkes processes, to model the spiking dynamics of LIF network dynamics has been studied in the literature previously (Löcherbach,

2017; Galves & Löcherbach, 2016; Mascart, 2021; Pfaffelhuber et al., 2022). We leverage these results to prove that heterogeneity in the synaptic dynamics can help reduce the spike count, as already discussed in the paper. We highlighted the key assumptions used in deriving the results in the Suppl. Sec. C. We apologize if there is still confusion, and we will add more in-depth discussion in the final manuscript as discussed below. In their paper, Locherbach et al.Löcherbach (2017) survey some aspects of the study of Hawkes processes in high dimensions to model biological neural systems and study their long-term behavior. Galves et al.Galves & Löcherbach (2016) provided an overview of point processes used as stochastic models for interacting neurons in discrete and continuous time. Similarly, Hawkes processes have met a recent interest in the mathematical neuroscience literature for their ability to model the dependence of a neuron's activity in the network's history (Mascart, 2021; Pfaffelhuber et al., 2022; Galves & Löcherbach, 2016; Gerhard et al., 2017; Zhou et al., 2020; Duval et al., 2022). Other works have also used a nonlinear interactive Hawkes process to model spiking neural networks with excitatory and inhibitory neurons (Chevallier et al., 2015; Chornoboy et al., 1988; Hansen et al., 2015; Reynaud-Bouret et al., 2014). Drawing from these works, we use a microscopic model describing a large network of interacting neurons that can generate oscillations in a macroscopic frame. In the model, the activity of each neuron is represented by a point process indicating the successive times at which the neuron emits a spike, where each realization of this point process is the spike train. We take the spiking intensity of a neuron as the probability of emitting a spike during the next instant, depending on the history of the neuron and the activity of other neurons in the network. The neurons interact through their synapses. This means that a spike of a pre-synaptic neuron leads to an increase of the membrane potential of the post-synaptic neuron if the synapse is excitatory or a decrease if the synapse is inhibitory, possibly after some delay, like the process of synaptic integration. The neuron fires a spike when the membrane potential reaches a certain upper threshold. Thus, excitatory inputs from the neurons in the network increase the firing intensity, and inhibitory inputs decrease it. Hawkes processes provide good models of this synaptic integration phenomenon by the structure of their intensity processes. This paper uses a general class of mean-field interacting Hawkes processes, modeling the reciprocal interactions between a population of excitatory neurons and a population of inhibitory neurons.

Let us consider a subsection of the HRSNN network as shown in Fig. 11 denoted by $N_x$. We use the multivariate Point process model to create a probabilistic model that relates the inner structure of the sub-network and its spiking activity. In this model, each neuron $i$ has a background spiking intensity $\nu_i$ caused by neurons outside the network. We know that when a neuron spikes, it impacts its spiking activity and the spiking activity of its output neurons. The impact of a neuron $j$ on neuron $i$ is modeled by a real function $h_{j \to i}(t)$. This impact can be excitatory or inhibitory depending on whether the pre-synaptic neuron is excitatory or inhibitory, as shown in Fig. 11. While the spikes from excitatory neurons try to excite another spike, spikes originating from inhibitory neurons try to inhibit the spiking of the cascading neuron.

A Hawkes process is a point process in which each point is commonly associated with event occurrences in time, where every event time impacts the probability that other events will take place subsequently. These processes are characterized by the conditional intensity function, seen as an instantaneous measure of the probability of event occurrences. A Hawkes process is a point process in which each point is commonly associated with event occurrences. In this past-dependent model, every event time impacts the probability that other events take place subsequently. These processes are characterized by the conditional intensity function, seen as an instantaneous measure of the probability of event occurrences. Although the self-exciting Hawkes process remains widely studied, there has been a growing interest in modeling the opposite effect, known as inhibition, in which the apparition of certain events lowers the probability of observing an event. In practice, this amounts to considering negative kernel functions. To maintain the positivity of the intensity function, a non-linear operator is added to the expression, which in turn entails the loss of the cluster representation. This model is known as the non-linear Hawkes process, where the existence of such processes was proved via construction using bi-dimensional marked Poisson processes. The general Hawkes framework can be written as:

$$\lambda_t^i = \Phi_i \left( \sum_{j \in \mathcal{S}_{i,E}} \int_0^t h_{j \to i}(t-u) \mathrm{d}Z_u^j \right), \tag{50}$$

where $\lambda_t^i$ is the intensity of neuron $i$, $\Phi_i$ a positive function, $Z_{j,t}$ is the counting process associated with neuron $j$, $h_{j \to i}(t)$ is the synaptic kernel associated with the synapse between neurons $j$ and $i$.

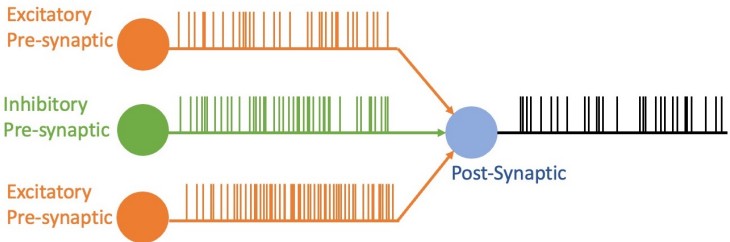

Figure 11: Figure showing the excitatory and inhibitory pre-synaptic neurons with excitatory and inhibitory spikes respectively incident on the post-synaptic neuron. We use this model to model the nonlinear interacting Hawkes process with inhibition.

To simplify the notation, we can rewrite Eq. 50 as

$$\lambda_i(t) = \Phi_i \left( \sum_{k \in I} \int_{(0,t)} h_{ki}(t-s) dZ_k(s) \right). \tag{51}$$

where $h_{ik}(t-s)$ measures the influence of neuron $k$ on neuron $i$ and how this influence vanishes with the time. More precisely, $h_{ik}(t-s)$ describes how a spike of neuron $k$ lying back $t-s$ time units in the past influences the present spiking rate at time $t$.

The goal of using heterogeneity in the STDP dynamics is to get better orthogonalization among the recurrent network states to lower higher-order correlations in spike trains. Studies have shown that the correlation of higher order progressively decreases the information available through neural population (Montani et al., 2009; Abbott & Dayan, 1999). Since we are trying to engineer a spike-efficient model, we leverage the heterogeneity in the STDP dynamics to reduce the higher-order correlations. The hypothesis is that using heterogeneity in STDP helps us orthogonalize the recurrent layer that can help us achieve an *efficient representation* of the input spike patterns with fewer spikes. This may be interpreted as the recurrent layer acting as an orthogonal bases function where inputs are projected onto these bases. Thus, having orthogonal bases can efficiently map inputs without much loss. While heterogeneous LIF neurons help us increase the number of principal components, thereby enabling us to store a greater subclass of features, heterogeneous STDP helps us efficiently encode this orthogonalization of the recurrent layer, resulting in fewer spikes compared to a homogeneous RSNN. Thus, in effect, heterogeneous STDP parameters can learn the output more precisely, which is projected back into the recurrent network. One of the primary reasons why heterogeneous STDP helps project the input to orthogonal activations of the recurrent network can be attributed to the distribution of LTD dynamics, as this increases the competition and helps distribute the input projection to multiple principal components. We discuss that the heterogeneous LTP/LTD dynamics in STDP lead to fewer spikes in the transmission of information.

***Lemma 3.2.1:*** *If the neuronal firing rate of the HRSNN network with only heterogeneity in LTP/LTD dynamics of STDP is represented as $\Phi_R$ and that of MRSNN represented as $\Phi_M$, then the HRSNN model promotes sparsity in the neural firing which can be represented as $\Phi_R < \Phi_M$.*

**Proof:** In this lemma, we show that the average firing rate of the model with heterogeneous STDP (LTP/LTD) dynamics (averaged over the population of neurons) is lesser than the corresponding average neuronal activation rate for a model with homogeneous STDP dynamics. We prove this by taking a sub-network of the HRSNN model as illustrated by Fig. 11. Now, we model the input spike trains of the pre-synaptic neurons using a multivariate interactive, nonlinear Hawkes process with multiplicative inhibition (Duval et al., 2022).

We consider a population of neurons of size $N$ that is divided into population $A$ (excitatory) with size $N_A := \alpha N$ and a population $B$ (inhibitory) with size $N_B = (1-\alpha)N$. A particular instance of the model is then given in terms of a family of counting processes $\left( Z_t^1, \ldots, Z_t^{N_A} \right)$ (population $A$) and $\left( Z_t^{N_A+1}, \ldots, Z_t^N \right)$ (population $B$) with coupled conditional stochastic intensities given respectively by $\lambda^A$ and $\lambda^B$. Consider on a filtered probability space $\left( \Omega, \mathcal{F}, (\mathcal{F}_t)_{t \geq 0}, \mathbf{P} \right)$ an independent family of i.i.d. Poisson measures $(\pi_i(\,ds,\,dz), i \in \{1, \ldots, N\})$ with intensity measure $ds \times dz$ on $[0, \infty) \times [0, \infty)$. Let $(x, y) \mapsto F(x, y)$ and $(x, y) \mapsto G(x, y)$ two nonnegative functions defined on $(0, \infty)^2$.

We assume that $F$ and $G$ satisfy

$$F(x, y) = \Phi_A(x)\Phi_{B \to A}(y), G(x, y) = \Phi_B(x) + \Phi_{A \to B}(y),$$

where $\Phi_A, \Phi_{B \to A}, \Phi_B$ and $\Phi_{A \to B}$ are nonnegative functions, each of them globally Lipschitz with $\Phi_{B \to A}$ bounded (and with no loss of generality we assume $0 \le \Phi_{B \to A} \le 1$).

Let us consider the family of càdlàg $(\mathcal{F}_t)_{t \ge 0}$ point processes $\left(Z_t^i\right)_{t \ge 0, i=1,\dots,N}$ given by

$$Z_t^i = \int_0^t \int_0^\infty \mathbf{1}_{z \le \lambda_s^i} \pi_i(\,\mathrm{d}s,\,\mathrm{d}z), i = 1, \dots, N,$$

where the intensity $\lambda^i, i = 1, \dots, N$, is given as:

$$\lambda_t^{A,N} := \Phi_A\left(\frac{1}{N}\sum_{j \in A}\int_0^{t^-} h_1(t-u)\mathrm{d}Z_u^j\right)\Phi_{B \to A}\left(\frac{1}{N}\sum_{j \in B}\int_0^{t^-} h_2(t-u)\mathrm{d}Z_u^j\right) \quad (52)$$

$$\lambda_t^{B,N} := \Phi_B\left(\frac{1}{N}\sum_{j \in B}\int_0^{t^-} h_3(t-u)\mathrm{d}Z_u^j\right) + \Phi_{A \to B}\left(\frac{1}{N}\sum_{j \in A}\int_0^{t^-} h_4(t-u)\mathrm{d}Z_u^j\right) \quad (53)$$

, where $A\&B$ are the populations of the excitatory and inhibitory neurons, respectively.

The dynamics given by Eq. 53 is of Hawkes type: each particle's intensity depends on the whole system's history, through memory kernels $h_i, i = 1, \dots, 4$ and firing rate functions $\Phi_A$ and $\Phi_B$. The multiplicative influence of inhibitory population $B$ onto population $A$, is represented using the inhibition kernel $\Phi_{B \to A}$ which is a decreasing nonnegative function on $[0, +\infty)$, with $\Phi_{B \to A}(0) = 1$ and $\Phi_{B \to A}(x) \xrightarrow[x \to \infty]{} 0$- i.e., activity of population $A$ should decrease as activity of population $B$ rises. The model secondly incorporates retroaction from population $A$ onto population $B$, which is supposed to be mostly additive, although possibly modulated by a nonlinear feedback kernel $\Phi_{A \to B}$.

Now, without loss of generality we assume that $\Phi_A$ and $\Phi_B$ are linear - i.e., $\Phi_A(x) = \mu_A + x, \Phi_B(x) = \mu_B + x, x \ge 0$, where $\mu_A, \mu_B \ge 0$, and $h_i \ge 0$ for $i = 1, \dots, 4$.

Hence, Eq. 53 becomes

$$\begin{cases} \lambda_t^A = \left(\mu_A + \alpha \int_0^t h_1(t-u)\lambda_u^A\,\mathrm{d}u\right)\Phi_{B \to A}\left((1-\alpha)\int_0^t h_2(t-u)\lambda_u^B\,\mathrm{d}u\right), \\ \lambda_t^B = \mu_B + (1-\alpha)\int_0^t h_3(t-u)\lambda_u^B\,\mathrm{d}u + \Phi_{A \to B}\left(\alpha\int_0^t h_4(t-u)\lambda_u^A\,\mathrm{d}u\right). \end{cases} \quad (54)$$

For heterogeneous neuron populations, there exists an asymmetry of the weights. Based on balanced spiking neural networks with heterogeneous connection strengths, previous works have revealed that such heterogeneous networks possess heavy-tailed Lévy fluctuations (Shlesinger et al., 1987; Mantegna & Stanley, 1995; Cossell et al., 2015). The heterogeneous heavy-tailed distributions of synaptic weights have been fitted to lognormal distributions (Buzsáki & Mizuseki, 2014; Kuśmierz et al., 2020). We model the inputs to neuron $i \in E$ as:

$$l_i(t) = W_{EX}\tau_{mE}\sum_{j \in X} c_{ij}s_j(t) + W_{EE}\tau_{mE}\sum_{j \in E} c_{ij}s_j(t) - W_{EI}\tau_{mE}\sum_{j \in I} c_{ij}s_j(t) \quad (55)$$

$$= \mu_1 E + \Delta\mu_i + \eta_i(t) \quad (56)$$

where $\mu$ denotes the mean inputs such that $\mu_E = K\tau_{mE}(W_{EX}r_X + W_{EE}r_E - W_{EI}r_l)$; $\Delta\mu_i = $ 'quenched' fluctuations (from neuron to neuron) with variance $\left\langle\Delta\mu^2\right\rangle_E = K\tau_{mE}^2\left(W_{EX}^2\left(r_X^2 + \Delta r_X^2\right) + W_{EE}^2\left(r_E^2 + \Delta r_E^2\right) + W_{EI}^2\left(r_I^2 + \Delta r_I^2\right)\right)$ due to random connectivity. Finally, $\eta_i$ denotes temporal fluctuations due to spiking activity. We assume that the pre-synaptic neurons fire as using the interactive Hawkes process described above.

Consider a case where $\mu_A \gg 1, \mu_B = 0$ and $h = 1_{[0,\theta]}$

$$\begin{cases} \lambda_t^A := \left(\mu_A + \alpha \int_0^t h_1(t-u)\mathrm{d}\lambda_u^A\right)\Phi_{B \to A}\left((1-\alpha)\int_0^t h_2(t-u)\mathrm{d}\lambda_u^B\right), \\ \lambda_t^B := (1-\alpha)\int_0^t h_3(t-u)\mathrm{d}\lambda_u^B + \alpha\int_0^t h_4(t-u)\mathrm{d}\lambda_u^A. \end{cases} \quad (57)$$

In a normal case, the excitatory and inhibitory populations follow the following steps: (1) $t \approx 0, \lambda_t^A \approx \mu_A$ is high and $\lambda_t^B \approx 0$ is small (2) Feedback from $A$ to $B : \lambda_t^B$ increases (3) Inhibition of $B$ to $A$ : when $\lambda_t^B$ gets high, $\Phi_{B \to A}$ reduces $\lambda_t^A$ (4) $h_4$ has compact support: after a time $\theta_4$, $B$ no longer feels the influence of $A$ : intensity of $B$ is back to $\mu_B \approx 0$ and $A$ to its normal high activity $\mu_A$ (State 1)

This leads to oscillations which lead to spikes. However, heterogeneity in the synaptic dynamics increases the stochasticity of the pre-synaptic spike arrival. Thus, due to the heterogeneity, $\Phi_{B \to A}$ promotes the system in the inhibition state (state 3) and inhibits the system's movement to system 4 and system 1, thereby creating a spike. Hence, $\Phi_R^A < \Phi_M^A$. Similarly, for the inhibitory neurons, we can show that $\Phi_R^B < \Phi_M^B$. Thus, we get $\Phi_R < \Phi_M$ ∎

This lemma might be interpreted as the heterogeneous STDP dynamics increasing the synaptic noise, which reduces the number of spikes of the post-synaptic neuron. A heterogeneous STDP leads to a non-uniform scaling of correlated spike trains leading to de-correlation. Hence, we can say that heterogeneous STDP models have learned a better-orthogonalized subspace representation, leading to a better encoding of the input space with fewer spikes.

It is to be mentioned here that the synaptic noise might be thought of as analogous to the stochasticity in the gradient descent algorithm. As recently proved by Simsekli et al. (Simsekli et al., 2020; 2019), stochasticity plays an important role in the generalization ability of the model. We might interpret the synaptic noise in the heterogeneous STDP to play a similar role and helps in better generalizability of the HRSNN model. This hypothesis is empirically proven in Supplementary Section A. However, a detailed theoretical analysis would be a very interesting direction for future work.

**Theorem 2:** *For a given number of neurons $N_\mathcal{R}$, the spike efficiency of the model $\mathcal{E} = \frac{\mathcal{C}(N_\mathcal{R})}{S}$ for HRSNN ($\mathcal{E}_R$) is greater than MRSNN ($\mathcal{E}_M$) i.e., $\mathcal{E}_R \geq \mathcal{E}_M$*

**Proof:** To study the effect of the spike time when the weight $w_k$ changes, we look into the expected value of the time difference in the post-synaptic spikes, which is given as:

$$\mathbb{E}\left[\Delta t_{\text{post}}\right] = \mathbb{E}\left[t_{\text{post}} - t_{\text{post}}\right] = \left(\mathbb{E}\left[t_{\text{post}}\right] - t_{\text{post}}\right)\Pr[s] \tag{58}$$

where $\Pr[\exists s]$ is the probability of occurrence of the post-synaptic spike. Thus, the expected input to the neuron at time $t(\mathbb{E}\left[i(t)\right])$, which comprises of its excitatory and inhibitory components $\mathbb{E}\left[i_e(t)\right], \mathbb{E}\left[i_i(t)\right]$ can be expressed as:

$$\mathbb{E}\left[\Delta i(t)\right] = \Delta\mathbb{E}\left[i_e(t)\right] - \Delta\mathbb{E}\left[i_i(t)\right] \quad \text{for } t < t_{\text{post}} \tag{59}$$

$$\text{where} \quad \mathbb{E}\left[i_e(t)\right] = \rho_e \int_0^\infty \mu_{w_e}(w, t)dw \quad ; \quad \mathbb{E}\left[i_i(t)\right] = \rho_i \int_0^\infty \mu_{w_i}(w, t)dw \tag{60}$$

where $\rho_e, \rho_i$ are the rates of incoming spikes and $\mu_{w_e}(w, t), \mu_{w_i}(w, t)$ the probabilities of the weights associated to time $t$. Now, considering the case for RSNNs with homogeneous STDP ($M$) and with heterogeneous STDP ($R$), the difference in the variances of the two populations is given as:

$$\Delta\text{Var}[V_M] - \Delta\text{Var}[V_R] = \Delta \int_{-\infty}^t [\mathbb{E}\left[i_M^2(t)\right] - \left(\mathbb{E}\left[i_R^2(t)\right] - \mathbb{E}[i_R(t)]^2\right)]dt \tag{61}$$

Since $t < t_{\text{post}}$, STDP potentiates both inhibitory and excitatory synapses, so $\Delta\mathbb{E}\left[i_i^2(t)\right] > 0, \Delta\mathbb{E}\left[i_e^2(t)\right] > 0$. The term $\mathbb{E}[i_M(t)]^2 = 0$ by the symmetry of the weights, and it is maintained at zero by the symmetry of the STDP. But for heterogeneous neuron populations, as described above, there exists an asymmetry of the weights. Based on balanced spiking neural networks with heterogeneous connection strengths, previous works have revealed that such heterogeneous networks possess heavy-tailed, Lévy fluctuations (Shlesinger et al., 1987; Mantegna & Stanley, 1995; Cossell et al., 2015). This implies $\mathbb{E}[i_R(t)]^2 > 0 \Rightarrow \Delta\text{Var}[V_R] < \Delta\text{Var}[v(t)_M]$ We calculate the number of post-synaptic spikes triggered when the stimulus is present. Now, representing the spike rate of the HRSNN and the MRSNN as $\Phi_R, \Phi_M$ resp.,

$$\int_0^t \Phi_R(t)dt \leq \int_0^t \Phi_M(t) \Rightarrow S_R = N_\mathcal{R}\frac{T}{t_R^{ISI}} \leq N_\mathcal{R}\frac{T}{t_M^{ISI}} = S_M \tag{62}$$

Thus, spikes decrease when we use heterogeneity in the LTP/LTD Dynamics. Hence, we compare the efficiencies of the HRSNN with that of MRSNN as follows:

$$\frac{\mathcal{E}_R}{\mathcal{E}_M} = \frac{M_R(N_{\mathcal{R}}) \times S_M}{S_R \times M_M(N_{\mathcal{R}})} = \frac{\sum_{\tau=1}^{N_{\mathcal{R}}} \frac{\text{Cov}^2(x(t-\tau), \mathbf{a}_\tau^R \mathbf{r}_R(t))}{\text{Var}(\mathbf{a}_\tau^R \mathbf{r}_R(t))} \times \int_{t_{ref}}^{\infty} t\Phi_R dt}{\sum_{\tau=1}^{N_{\mathcal{R}}} \frac{\text{Cov}^2(x(t-\tau), \mathbf{a}_\tau^M \mathbf{r}_M(t))}{\text{Var}(\mathbf{a}_\tau^M \mathbf{r}_M(t))} \times \int_{t_{ref}}^{\infty} t\Phi_M dt} \tag{63}$$

Since $S_R \le S_M$ and also, the covariance increases when the neurons become correlated, and as neuronal correlation decreases, $\mathcal{H}$ increases (Theorem 1), we see that $\frac{\mathcal{E}_R}{\mathcal{E}_M} \ge 1 \Rightarrow \mathcal{E}_R \ge \mathcal{E}_M$ ∎

# C  SUPPLEMENTARY SECTION C

## C.1  HIGHER ORDER CORRELATION

In this paper, we took inspiration from results in reservoir computing, which show that we can maximize memory capacity using orthogonalization among reservoir states in the case of reservoir computers (Farkaš & Gergel', 2017; Farkaš et al., 2016). The goal of using heterogeneous STDP dynamics is to get better orthogonalized recurrent network states to achieve more efficient information transfer with lower higher-order correlations in spike trains. Recent studies (Montani et al., 2009; Abbott & Dayan, 1999) have shown that the correlation of higher order progressively decreases the information available through the neural population. The decrease in information becomes larger as the interaction order grows. Since we are trying to engineer a spike-efficient model, we leverage the heterogeneity in neuronal parameters to reduce the higher-order correlations. The hypothesis is that an orthogonal recurrent layer can help us efficiently represent the input spike patterns with fewer spikes. This may be interpreted as the recurrent layer acting as an orthogonal bases function where the inputs are projected onto these bases. Thus, having orthogonal bases can efficiently map the inputs without much loss. The heterogeneous STDP helps us efficiently achieve this orthogonalization of the recurrent layer, resulting in a lesser voltage variance across the neuron population. This leads to fewer spikes (since the mean is constant) compared to a homogeneous RSNN. Thus, in effect, heterogeneous STDP parameters can learn the output more precisely, which is projected back into the recurrent network. Hence, using heterogeneous STDP parameters leads to a better orthogonalization among the neuronal states and hence, a higher $\mathcal{C}$.

In this paper, we show that using a distribution of LTP/LTD dynamics in the STDP parameters helps us in mappings the input onto the orthogonal activations of the recurrent network to capture the principal components of the input signal. The LTD dynamics play an important role in determining the orthogonality of neuronal activations. LTD windows of the STDP rules enable robust sequence learning amid background noise in cooperation with a large signal transmission delay between neurons and a theta rhythm (Hayashi & Igarashi, 2009). The LTD window in the range of positive spike-timing plays an important role in preventing noise influences with sequence learning. Oja (Oja, 1982; 1989) showed that the LIF neuron's time constant is very fast compared to the time constant of learning in which the weights $w_{ji}$ change. The learning is assumed to take place according to the STDP type conjunction of the inputs $\xi_i$ and the integrated effect of the inputs, $\nu_j$, with an additional forgetting term attributed to the LTD dynamics: $\frac{dw_{ji}}{dt} = \alpha\nu_j\xi_i - f(\nu_j, \xi_i, w_{ji})$ In the case of homogeneous STDP, $f(.)$ is a constant; hence, the model can only efficiently learn the first principal component of the input. However, quite interesting functions emerge when considering STDP to have a distribution. This also helps us determine the next principal components other than the first one. Hence the diversity in the different LTD dynamics increases the competition and helps that not all inputs are mapped to the first principle component. Thus, the diversity in the LTD dynamics helps in projecting the input to orthogonal activations of the recurrent network.

Now, for homogeneous RSNNs, several higher-order correlations, which according to our hypothesis, arise because of the poor orthogonalization among the network states. This results in the redundancies of spikes for encoding the same information. In this paper, we use heterogeneous STDP dynamics to learn an efficient orthogonal representation of the state space, which result in the network learning the same patterns but using fewer spikes. (theorem: 3) We also show that heterogeneity in the neuronal parameters decreases the neuronal correlation (theorem 1 And fig 2a). Thus, since heterogeneity results in better orthogonalization among the neuronal states, it results in fewer higher-order correlations. Moreover, recent studies have shown that the correlation of higher order progressively

Table 11: Table Showing the estimated highest order of correlation for HRSNN vs. MRSNN using CuBIC

|  | $\hat{\xi}$ | p-value | Estimated Highest-order of correlation ($\hat{\xi} + 1$) |
|---|---|---|---|
| HRSNN | 2 | 0.423 | 3 |
| MRSNN | 5 | 0.358 | 6 |

decreases the information available through the neural population, and the decrease in information becomes larger as the interaction order grows. Since we are trying to engineer an efficient model, we aim to reduce the higher-order correlations using heterogeneity in neuronal parameters (as shown in Theorem 1). In addition to this, to verify this, we used CuBIC (Staude et al., 2010), a cumulant-based inference of higher-order correlations in massively parallel spike trains. The details of the experimental methodology are given in Supplementary Section C. The outcome of CuBIC is a lower bound $\hat{\xi}$ on the order of correlation in the spiking activity of large groups of simultaneously recorded neurons. CuBIC can provide statistical evidence for large correlated groups without the discouraging requirements on a sample size that direct tests for higher-order correlations have to meet. This is achieved by exploiting constraining relations among correlations of different orders. However, it must be noted that CuBIC is not designed to estimate the order of correlation directly; the inferred lower bound might not always correspond to the maximal order of correlation present in a given data set.

