# OpenReview forum: "Heterogeneous Neuronal and Synaptic Dynamics for Spike-Efficient Unsupervised Learning: Theory and Design Principles"
_ICLR.cc/2023/Conference — ICLR 2023 poster_

### Official Review · Reviewer_rFWy · 2022-10-24

**Confidence:** 4
**Clarity, Quality, Novelty And Reproducibility:** The paper is clearly written with mod…
**Correctness:** 3
**Technical Novelty And Significance:** 2
**Empirical Novelty And Significance:** 2
**Recommendation:** 6

**Strength And Weaknesses:**

Strength:
1. The manuscript is clearly written with sufficient details.
2. The idea of quantifying the capacity and heterogeneity is interesting.

Weakness:
Although this topic appears to be interesting, there seem to lack sufficient evidence to support the claim. Specifically, I have the following concerns.
1. The capacity can be intuitively understood as a measurement of how much variance in the past are still maintained in the current state. However, what's the relationship between this capacity and the model performance or generalization ability? As is known for the deep learning setup, overfit on the training dataset is a typical phenomenon. Thus increasing only the capacity is not enough for such cases.

2. The definition of spike-efficiency is a bit random especially the factor 1000. Is there any reasoning on why we should define it as a ratio? Since both the memory capacity and spiking rate are unit-free variables, their relationship can be composed in multiple ways. Thus it would make more sense if this definition is derived under a certain criterion rather than simply claimed.

3. Intuitively one may expect that if we improve the heterogeneity of parameters, it would lead to more complex models thus higher ability in memorizing certain modes. For the results in Table 2, soly increasing heterogeneity of LIF does not necessarily improve the efficiency or not. How should we understand such difference with STDP? Specially for Lemma 3.2.1, the $\Delta Var$ is not defined. Also, it seems that the proof here is a directive corralory from the heavy-tail claim, I am not 100% sure about the reliability here.

4. The experiments are on fairly simple cases. To validate the capacity and the assumption, experiments on some more complex static/dynamic dataset may be included.

**Summary Of The Paper:**

In this paper, the authors proposed the spike-efficiency measurement for spiking neural networks and showed the relationship between spike-efficiency and the heterogeneity. They also performed some empirical experiments to demonstrate the significance of neuronal and synaptic heterogeneity in reducing spiking activity.

**Summary Of The Review:**

Overall, this is a paper that proposes an interesting question of measuring the performance of RSNN. However, the discussion remains a bit superficial.

---

> ### Author Response · Authors · 2022-11-19
> **Detailed Response to Weaknesses**
>
> We thank the reviewer for their feedback. In view of their comments, we present the following rebuttal:
>
> > The capacity can be intuitively understood as a measurement of how much variance in the past are still maintained in the current state. However, what's the relationship between this capacity and the model performance or generalization ability? As is known for the deep learning setup, overfit on the training dataset is a typical phenomenon. Thus increasing only the capacity is not enough for such cases.
>
> We thank the reviewer for their interesting feedback.
> We have performed additional experiments on the four ablation models for the classification task on the SHD dataset to understand the impact of heterogeneous LIF neurons and heterogenous STDP on generalizability (along with accuracy). The complete results are reported in Supplementary Section A (also reproduced below).
>
> A model of only heterogeneous LIF neurons increases the memory capacity. As the reviewer correctly pointed out, although a higher memory capacity improves performance (accuracy), it may lead to overfitting the training data, thereby increasing generalization error. On the other hand, we observed that adding heterogeneity in STDP dynamics leads to reduces generalization error of the HRSNN model.
>
> In summary, heterogeneity in the LIF neurons helps increase memory capacity and learn more feature representations which leads to improved performance (accuracy). In contrast, heterogeneity in the STDP dynamics improves the model's generalizability. Therefore, adding heterogeneity on both neuronal and synaptic dynamics allow us to improve performance with similar (or better) generalizability.
>
> |                  | Testing Accuracy | Generalization Error (Training - Testing Accuracy) |
> |:----------------:|:----------------:|:--------------------------------------------------:|
> | Hom LIF Hom STDP |       73.58      |                        13.75                       |
> | Hom LIF Het STDP |       74.03      |                        11.28                       |
> | Het LIF Hom STDP |       78.87      |                        16.42                       |
> | Het LIF Het STDP |       80.49      |                        13.83                       |
>
>
> In addition to this, we have trained the models with limited training data. We observe that the HRSNN model with heterogeneous LIF and STDP dynamics not only has better testing accuracy but also shows better generalization behavior when compared to other homogeneous RSNN or the other ablation heterogeneous models (with heterogeneity in only one of them). Also, we see that the HRSNN model with only heterogeneous STDP (model A) shows distinctly better generalization ability than the generalization ability of HRSNN with only heterogeneous LIF neurons (model B). On the other hand, model B showcases significantly higher training and testing accuracy than model A. This can be interpreted as follows: since heterogeneous LIF dynamics increase the memory capacity, it leads to an overfitting of the data. Heterogeneous STDP dynamics help in obtaining more generalizable solutions from this. Each has its pros and cons; however, using HRSNN with both heterogeneous LIF and STDP dynamics shows better performance and generalization abilities. The detailed results are discussed in Supplementary Section A.
>
>
>
> > The definition of spike-efficiency is a bit random especially the factor 1000. Is there any reasoning on why we should define it as a ratio? Since both the memory capacity and spiking rate are unit-free variables, their relationship can be composed in multiple ways. Thus it would make more sense if this definition is derived under a certain criterion rather than simply claimed.
>
> We agree with the reviewer on their suggestion. We initially introduced the ‘1000’ term for easier readability but have removed them from the definition to avoid confusion.
>
> To make spike efficiency more well-defined, we have added certain constraints in memory capacity and spike rate definitions. We specified that the number of spikes (in the recurrent layer of the HRSNN network) reported in the paper is counted until the first appearance of a spike in the final readout layer.

---

> > ### Author Response · Authors · 2022-11-19
> > **Detailed Response to Weakness III**
> >
> > > The experiments are on fairly simple cases. To validate the capacity and the assumption, experiments on some more complex static/dynamic dataset may be included.
> >
> > The focus of this paper is to develop a theoretical foundation for HRSNN; empirical results are primarily used to re-verify the theoretical predictions. In this light, we agree that the time series classification dataset used in the paper (SHD) is simpler than what is often used in the DNN baselines, but SHD is used quite often in the SNN literature. [1-6]
> >
> > The key task used in this paper is prediction. We would also like to highlight that the Lorenz96 is a complex dynamical system that has been widely used as a prototype for multiscale chaotic variability of the weather and climate system and a useful test bed for novel methods. Our paper uses a large-scale forcing that makes the system highly chaotic and tuned to produce appropriate spatiotemporal variability. We focus on predicting the Y axes, which have relatively moderate amplitudes compared to X, and Z, and demonstrate high-frequency variability and intermittency, which makes the prediction problem difficult. A snippet of the time series is shown in Supplementary Section A to demonstrate the chosen data's complexity.
> >
> >
> > [1] Yao, M., Gao, H., Zhao, G., Wang, D., Lin, Y., Yang, Z. and Li, G., 2021. Temporal-wise attention spiking neural networks for event streams classification. In Proceedings of the IEEE/CVF International Conference on Computer Vision (pp. 10221-10230).
> > [2] Yin, B., Corradi, F. and Bohté, S.M., 2020, July. Effective and efficient computation with multiple-timescale spiking recurrent neural networks. In International Conference on Neuromorphic Systems 2020 (pp. 1-8).
> > [3] Eshraghian, J.K. and Lu, W.D., 2022. The fine line between dead neurons and sparsity in binarized spiking neural networks. arXiv preprint arXiv:2201.11915.
> > [4] Perez-Nieves, N. and Goodman, D., 2021. Sparse spiking gradient descent. Advances in Neural Information Processing Systems, 34, pp.11795-11808.
> > [5] Yin, B., Corradi, F. and Bohté, S.M., 2021. Accurate and efficient time-domain classification with adaptive spiking recurrent neural networks. Nature Machine Intelligence, 3(10), pp.905-913.
> > [6] Perez-Nieves, N., Leung, V.C., Dragotti, P.L. and Goodman, D.F., 2021. Neural heterogeneity promotes robust learning. Nature communications, 12(1), pp.1-9.

---

> > > ### Comment · Reviewer_rFWy · 2022-11-20
> > > **Response to the rebuttal**
> > >
> > > Thank you for answering my question here. My concern about raising this question is based on the same logic above, i.e. how the proposed measurement took care of the model performance in terms of model capacity (remembering more variance) and model generalization (adapted prediction accuracy or other definition). As some of the proof or demonstration is simplified, it is necessary to demonstrate that the same reasoning procedure, which is not included in the definition, can work on complex scenarios as well. To me, this can not be naively derived based on the proposed material in the current state, and it is why I ask for more demonstrations to support the claims.

---

> > > > ### Comment · Area_Chair_yr7Z · 2022-11-23
> > > > **Following up on rebuttal and comeback by reviewer**
> > > >
> > > > Dear authors,
> > > >
> > > > do you have a response to the points raised by the reviewer after the rebuttal?
> > > >
> > > > Cheers,
> > > > Your AC

---

> > > > > ### Author Response · Authors · 2022-11-24
> > > > > **General Response to the AC, Reviewer rFWy**
> > > > >
> > > > > Sorry for the delay in the response. We were waiting for some of the results to complete. We have added our responses to the reviewer's points and hope we have addressed their concerns. Thank you!

---

> > > > > ### Comment · Reviewer_MSRY · 2022-11-29
> > > > > **thanks for the clarifications**
> > > > >
> > > > > I remain positive about the paper and maintain my score.

---

> > > > ### Author Response · Authors · 2022-11-24
> > > > **Response to the Reviewer**
> > > >
> > > > We thank the reviewer for their feedback. We have trained the model on more complex datasets (SSC)[1], and CIFAR10-DVS[2], and the complete results are shown in Table 1.
> > > >
> > > > |                  |                        | SHD                  |                               |                        |        SSC          |                              |                        | CIFAR10 DVS          |                               |
> > > > |------------------|------------------------|----------------------|-------------------------------|------------------------|----------------------|-------------------------------|------------------------|----------------------|-------------------------------|
> > > > |                  | Training  Accuracy (A) | Testing Accuracy (B) | Generalization Error \|A -B\| | Training  Accuracy (A) | Testing Accuracy (B) | Generalization Error \|A -B\| | Training  Accuracy (A) | Testing Accuracy (B) | Generalization Error \|A -B\| |
> > > > | Hom LIF Hom STDP |    $86.92 \pm 1.35$    |   $72.89 \pm 1.85$   |       $14.03 \pm 1.67$        | $74.69 \pm 1.72$       | $47.94 \pm 1.94$     | $26.75 \pm 1.42$              | $82.41 \pm 1.8$        | $65.33 \pm 3.41$     | $17.08 \pm 1.35$              |
> > > > | Hom LIF Het STDP |    $85.76 \pm 1.27$    |   $73.91 \pm 1.49$   |        $11.85 \pm 1.25$       | $76.79\pm 1.58$        | $52.96 \pm 1.73$     | $23.86 \pm 1.29$              | $83.48 \pm 1.52$       | $67.06 \pm 2.97$     | $16.42 \pm 1.24$              |
> > > > | Het LIF Hom STDP |    $95.29 \pm 1.16$    |   $78.36 \pm 1.42$   |        $16.93 \pm 1.13$       | $84.26 \pm 1.33$       | $55.11\pm 1.65$      | $29.15 \pm 1.12$              | $86.93 \pm 1.79$       | $68.37 \pm 3.05$     | $18.56 \pm 1.42$              |
> > > > | Het LIF Het STDP |    $94.07 \pm 1.03$    |   $80.01 \pm 1.13$   |        $14.06 \pm 1.02$       | $86.41 \pm 1.49$       | $59.28 \pm 1.35$     | $27.13 \pm 0.97$              | $87.49 \pm 1.76$       | $70.54 \pm 1.82$     | $16.95 \pm 1.38$              |
> > > >
> > > >
> > > >
> > > > [1] Cramer, B., Stradmann, Y., Schemmel, J. and Zenke, F., 2020. The heidelberg spiking data sets for the systematic evaluation of spiking neural networks. IEEE Transactions on Neural Networks and Learning Systems.
> > > > [2] Li, H., Liu, H., Ji, X., Li, G. and Shi, L., 2017. Cifar10-dvs: an event-stream dataset for object classification. Frontiers in neuroscience, 11, p.309.

---

> > ### Author Response · Authors · 2022-11-19
> > **Detailed Response to Weakness II**
> >
> > > Intuitively one may expect that if we improve the heterogeneity of parameters, it would lead to more complex models thus higher ability in memorizing certain modes. For the results in Table 2, soly increasing heterogeneity of LIF does not necessarily improve the efficiency or not. How should we understand such difference with STDP?
> >
> > The reviewer has raised a very interesting question. From Table 2, we observed that heterogeneity in the STDP does not significantly impact the mode's performance but significantly decreases the average spike rate. This may be interpreted as the recurrent layer acting as an orthogonal bases function where inputs are projected onto these bases. Thus, having orthogonal bases can efficiently map inputs without much loss. Thus, heterogeneity in STDP helps us orthogonalize the recurrent layer, which can help us achieve an efficient representation of the input spike patterns with fewer spikes. While heterogeneous LIF neurons help us increase the number of principal components, thereby enabling us to store a greater subclass of features, heterogeneous STDP helps us efficiently encode this orthogonalization of the recurrent layer. Thus, in effect, heterogeneous STDP parameters can learn the output more precisely, which is projected back into the recurrent network. One of the primary reasons why heterogeneous STDP helps project the input to orthogonal activations of the recurrent network can be attributed to the distribution of LTD dynamics, as this increases the competition and assists in distributing the projection of the inputs to multiple principal components.
> > Again, based on the reviewer's suggestion and new experiments on the generalization properties (shown in the table above), we can see that heterogeneity in STDP helps improve the model's generalizability. This falls in place with our hypothesis that heterogeneity in STDP helps us orthogonalize the recurrent layer since an orthogonalized state space means the network can identify the principal components of unknown data more efficiently. This leads to a network with better generalization characteristics, as we see in this case also.
> >
> >
> > > Specially for Lemma 3.2.1, the $\Delta Var$ is not defined. Also, it seems that the proof here is a directive corralory from the heavy-tail claim, I am not 100% sure about the reliability here.
> >
> > We thank the reviewers for pointing out this in Lemma 3.2.1, and we agree that proving a lower variance in the neuronal states does not necessarily mean fewer spike counts. As such, we have taken a different approach to prove that heterogeneity in the STDP reduces the spike count. We model the neural spike trains as a multivariate nonlinear Hawkes process model with inhibition. Hence, we show that using heterogeneous STDP dynamics leads to increased synaptic noise, which leads to greater inhibition of post-synaptic spikes during the neural oscillation, which leads to a decrease in the average firing rate of the model. Further details and the complete proof are shown in Supplementary Section B. We believe this is a much more robust proof that also addresses the main limitations of the previous proof. We would again like to thank the reviewers for pointing this out, as it leads to a more robust and complete proof.

---

> > > ### Comment · Reviewer_rFWy · 2022-11-20
> > > **Response to the rebuttal**
> > >
> > > Thanks for the interpretation. My question here is whether there is any proof/demonstration of the claim of the happening here. The idea of explaining the phenomenon from PCA or kernel perspectives is interesting but requires some additional validation here.

---

> > > > ### Author Response · Authors · 2022-11-24
> > > > **Response to the reviewer**
> > > >
> > > > We thank the reviewer for their feedback. To validate the idea that heterogeneity helps us increase the number of principal components, thereby enabling us to store a greater subclass of features, we plotted the t-SNE plots for the MRSNN and the HRSNN models for the CIFAR10—DVS dataset and is shown in [Fig. tsne2](https://anonymous.4open.science/r/ICLR23-8EDB/tsne2.jpg).  The figure qualitatively indicates that the HRSNN model can classify better than MRSNN model. It is to be noted here that these are some preliminary results, and further research needs to be done in the future.
> > > >
> > > > Fig. tsne2 URL:  https://anonymous.4open.science/r/ICLR23-8EDB/tsne2.jpg

---

> > ### Comment · Reviewer_rFWy · 2022-11-20
> > **Response to the rebuttal**
> >
> > Thank you for the response to my questions. But it seemed that the authors might misunderstand my concerns here. My question is what's the relationship between the capacity defined in the manuscript and the model performance or generalization ability rather than how the heterogeneity in LIF or STDP contributes. I ask this question because the authors were attempting to investigate the model performance through the proposed measurement. Thus it would be supportive if they explained the observation associated with my questions through their proposed methodology. Also, as a minor suggestion, in terms of the table shown above, considering that the numbers are close, it would make more sense if they provide additional details of the training accuracy and the errorbars trained multiple times and multiple datasets since this is a quantitative demonstration rather than a proof here.

---

> > > ### Author Response · Authors · 2022-11-24
> > > **Response to the Reviewer**
> > >
> > > We thank the reviewer for raising an interesting question. The relationship between memory capacity and model performance is greatly studied in the literature [1-7]. In this work, we see an empirical association between memory capacity and model performance. The model performance improves when we use memory capacity as the objective function for Bayesian Optimization. To validate our hypothesis that we can improve model performance by controlling memory capacity, we did the following empirical studies:
> > > 1. We study the t-SNE plots for MRSNN networks as we increase the memory capacity by optimizing the model parameters. We perform the experiment on the CIFAR10-DVS dataset. The result is shown in [Fig. tsne1](https://anonymous.4open.science/r/ICLR23-8EDB/tsne1.jpg) From the figure, we see that there is better separability between the clusters as we increase the memory capacity, which leads to better classification results. This highlights that there might be a correlation between memory capacity and model performance.
> > >
> > > 2. To further validate the hypothesis, we perform a synthetic control experiment to study the causality between the two variables. Using Fisher’s exact test, we can statistically show that when the memory capacity of the model increases, the accuracy of the model also increases. The details of the experiment are given below.
> > >
> > > We aim to understand the causal relationship between memory capacity and performance. In particular, we aim to test whether an increase in memory capacity increases the model performance. We leverage a statistical causal inference technique called Synthetic Control [8,9] to demonstrate this. We use a small RSNN neural network with 100 neurons, and the synapses between these neurons are chosen randomly with a fixed probability. We instantiate 10 such random networks and fix the model architectures. Next, we use Fisher’s Exact Tests to do inference with synthetic control using a simple seq-MNIST dataset. The experimental setup is as follows:
> > >
> > > 1. We use BO to optimize the hyper-parameters of an RSNN (Note this is a standard BO where we search for the individual parameters of all the neurons and the synapses). The objective function of the BO is to maximize the model's generalizability, keeping its memory capacity fixed. The generalization ability is measured using its Hausdorff Dimension, as discussed in previous works [10,11], which is orthogonal to the model performance. Basically, this BO method regularizes the model to have flatter minimas.
> > >
> > > 2. We evaluate the BO for 20 evaluations and observe the memory capacity and the model performance for each of these evaluations. We observe that as we force the BO to keep the memory capacity fixed, it does not vary much. However, it is interesting to note here that the model performance at each of these steps also does not vary significantly. (defined in point 3)
> > >
> > > 3. We count the number of times the memory capacity and the model performance increase. We define that a variable (either memory capacity or model performance) ‘increases’ if the difference in the value of the variable at that instant and the moving average of the variable from the past instances is more than 1% (chosen arbitrarily). Hence, we count the number of instances when each of these metrics satisfies this criterion.
> > >
> > > 4. After 20 evaluations, there are 2 different BO problems that we use:
> > > - Case 1: The control group: We keep running this BO where we keep the memory capacity constant and optimize the generalizability of the model
> > > - Case 2: The treatment group: We change the BO to optimize the memory capacity of the model
> > >
> > > 5. In each of the two cases, we observe the accuracy and the memory capacity and calculate if, at each time, the value deviates from the mean (as discussed in step 3). We take the average of the counts and tabulate it as follows:
> > >
> > > |                                                   |     First 20 steps  (MC constant)    |     Last 20 steps  (MC maximized)    |
> > > |:-------------------------------------------------:|:------------------------------------:|:------------------------------------:|
> > > |     Acc increase   (avg.   over 10 iterations)    |                   2                 |                   15               |
> > > |       No Acc Increase   (avg. over 10 iterations)     |                   18                 |                   5                  |
> > >
> > > 6. From this table, we calculate the p-value to test against the null hypothesis that the accuracy does not increase with an increase in memory capacity. The p-value calculated is less than 0.0001, which indicates that increasing memory capacity increases the accuracy of the model. The results are shown in [Fig. fig_causal](https://anonymous.4open.science/r/ICLR23-8EDB/fig_causal.jpg)
> > >
> > > Figure tsne1 URL: https://anonymous.4open.science/r/ICLR23-8EDB/tsne1.jpg
> > > Figure fig_causal URL: https://anonymous.4open.science/r/ICLR23-8EDB/fig_causal.jpg

---

> > > > ### Author Response · Authors · 2022-11-24
> > > > **References**
> > > >
> > > > > Also, as a minor suggestion, in terms of the table shown above, considering that the numbers are close, it would make more sense if they provide additional details of the training accuracy and the errorbars trained multiple times and multiple datasets since this is a quantitative demonstration rather than a proof here.
> > > >
> > > > We thank the reviewer for their suggestion. We have updated the table as follows:
> > > >
> > > > |                  |                        | SHD                  |                               |                        | SSC                  |                               |                        | CIFAR10 DVS          |                               |
> > > > |------------------|------------------------|----------------------|-------------------------------|------------------------|----------------------|-------------------------------|------------------------|----------------------|-------------------------------|
> > > > |                  | Training  Accuracy (A) | Testing Accuracy (B) | Generalization Error \|A -B\| | Training  Accuracy (A) | Testing Accuracy (B) | Generalization Error \|A -B\| | Training  Accuracy (A) | Testing Accuracy (B) | Generalization Error \|A -B\| |
> > > > | Hom LIF Hom STDP |    $86.92 \pm 1.35$    |   $72.89 \pm 1.85$   |       $14.03 \pm 1.67$        | $74.69 \pm 1.72$       | $47.94 \pm 1.94$     | $26.75 \pm 1.42$              | $82.41 \pm 1.8$        | $65.33 \pm 3.41$     | $17.08 \pm 1.35$              |
> > > > | Hom LIF Het STDP |    $85.76 \pm 1.27$    |   $73.91 \pm 1.49$   |        $11.85 \pm 1.25$       | $76.79\pm 1.58$        | $52.96 \pm 1.73$     | $23.86 \pm 1.29$              | $83.48 \pm 1.52$       | $67.06 \pm 2.97$     | $16.42 \pm 1.24$              |
> > > > | Het LIF Hom STDP |    $95.29 \pm 1.16$    |   $78.36 \pm 1.42$   |        $16.93 \pm 1.13$       | $84.26 \pm 1.33$       | $55.11\pm 1.65$      | $29.15 \pm 1.12$              | $86.93 \pm 1.79$       | $68.37 \pm 3.05$     | $18.56 \pm 1.42$              |
> > > > | Het LIF Het STDP |    $94.07 \pm 1.03$    |   $80.01 \pm 1.13$   |        $14.06 \pm 1.02$       | $86.41 \pm 1.49$       | $59.28 \pm 1.35$     | $27.13 \pm 0.97$              | $87.49 \pm 1.76$       | $70.54 \pm 1.82$     | $16.95 \pm 1.38$              |
> > > >
> > > >
> > > >
> > > > [1] Lun, S.X., Yao, X.S. and Hu, H.F., 2016. A new echo state network with variable memory length. Information Sciences, 370, pp.103-119.
> > > > [2] Rodan, A. and Tino, P., 2010. Minimum complexity echo state network. IEEE transactions on neural networks, 22(1), pp.131-144.
> > > > [3] Sun, X., Li, T., Li, Q., Huang, Y. and Li, Y., 2017. Deep belief echo-state network and its application to time series prediction. Knowledge-Based Systems, 130, pp.17-29.
> > > > [4] Fusi, S., 2021. Memory capacity of neural network models. arXiv preprint arXiv:2108.07839.
> > > > [5] Rodriguez, N., Izquierdo, E. and Ahn, Y.Y., 2019. Optimal modularity and memory capacity of neural reservoirs. Network Neuroscience, 3(2), pp.551-566.
> > > > [6] Farkaš, I., Bosák, R. and Gergeľ, P., 2016. Computational analysis of memory capacity in echo state networks. Neural Networks, 83, pp.109-120.
> > > > [7] Mejias, J.F. and Torres, J.J., 2009. Maximum memory capacity on neural networks with short-term synaptic depression and facilitation. Neural computation, 21(3), pp.851-871.
> > > > [8] Xu, Y., 2017. Generalized synthetic control method: Causal inference with interactive fixed effects models. Political Analysis, 25(1), pp.57-76.
> > > > [9] Hernán, M.A. and Robins, J.M., 2010. Causal inference.
> > > > [10] Chakraborty, B. and Mukhopadhyay, S., 2021. Characterization of Generalizability of Spike Timing Dependent Plasticity Trained Spiking Neural Networks. Frontiers in Neuroscience, 15, p.695357.
> > > > [11] Simsekli, U., Sener, O., Deligiannidis, G. and Erdogdu, M.A., 2020. Hausdorff dimension, heavy tails, and generalization in neural networks. Advances in Neural Information Processing Systems, 33, pp.5138-5151.

---

> > > > > ### Comment · Reviewer_rFWy · 2022-11-25
> > > > > **Response to rebuttal**
> > > > >
> > > > > Thank you for the further explanation. I am happy to increase my rating to 6.

---

> > > > > > ### Author Response · Authors · 2022-11-26
> > > > > > **Thank You.**
> > > > > >
> > > > > > Thank you so much for the positive feedback which helped us to improve the quality of our paper.

---

> ### Author Response · Authors · 2022-11-19
> **Summary of Changes**
>
> We thank the reviewer for their feedback. We hope we have addressed their concerns and questions regarding the paper and hope they will reconsider their rating. Taking into account the feedback from the reviewer, the summary of the changes is listed as follows:
> 1. We studied the generalizability of the models and added experimental results to show that heterogeneous LIF improves the model performance and heterogeneous STDP improves the generalizability of the model.
> 2. To show the robustness of the HRSNN model, we have trained the models with limited training data. The detailed results are discussed in Supplementary Section A.
> 3. We have formalized the definitions of memory capacity and spike efficiency, adding specific constraints on the inputs and conditions for calculating spike count.
> 4. We have added an intuitive explanation behind the effect of heterogeneous STDP in HRSNN.
> 5. We proved lemma 3.2.1 using a different approach to prove that heterogeneity in the STDP reduces the spike count. We model the neural spike trains as a multivariate nonlinear Hawkes process model with inhibition. We believe this is a much more robust proof that also addresses the main limitations of the previous proof. We would again like to thank the reviewers for pointing this out, as it leads to a more robust and complete proof.

---

### Official Review · Reviewer_MSRY · 2022-10-24

**Confidence:** 3
**Correctness:** 4
**Technical Novelty And Significance:** 4
**Empirical Novelty And Significance:** 4
**Recommendation:** 8

**Clarity, Quality, Novelty And Reproducibility:**

I found the writing very clear, precise and easy to follow

would be useful to expand slightly on the bayesian optimization procedure, in particular the  parameter distributions being optimized over

**Strength And Weaknesses:**

Strengths:
+ biological motivation
+ what in hardware may be traditionally deemed as a bug becomes a computational feature of the system
+ rigorous analysis of implications
+ numerics are solid

Weakness:
- capacity measure somewhat restricting as a measure of computational power



**Summary Of The Paper:**

The paper investigates the role of neural and synaptic heterogeneity in improving the computational capacity and/or energy efficiency of spiking recurrent neural nets. It also provides an optimization procedure for determining the optimal degree of heterogeneity for a problem.

**Summary Of The Review:**

Overall, a solid contribution with clear problem framing, theoretical and numerical support and robust numerical results.

---

> ### Author Response · Authors · 2022-11-19
> **Response to Weaknesses and Questions**
>
> We thank the reviewer for their feedback, and we appreciate their vote of confidence in our paper. In view of some of the questions raised by the reviewer, we present the following rebuttal:
>
> > capacity measure somewhat restricting as a measure of computational power
>
> We agree with the reviewer’s comment. Though memory capacity might be a restricting measure to estimate computational power, it is commonly used as an estimator for the model's performance [1-3]. In this work, we see an empirical association between memory capacity and model performance. The model performance improves when we use memory capacity as the objective function for Bayesian Optimization. However, it must be noted here that deriving a good measure for estimating the computational power of an RSNN is an interesting and open research question but is out of the scope of this paper. However, it might be an interesting future research direction to work on.
>
>
> > would be useful to expand slightly on the bayesian optimization procedure, in particular the parameter distributions being optimized over
>
> We thank the reviewer for their comment. We have added more details about the Bayesian Optimization process in Supplementary Section A, including the initial parameter values and the final searched distributions. We have also added further discussion about the methodology of the modified BO and how the Matern kernel is calculated using the Wasserstein distance. We have also added Table 6, which shows the optimized hyperparameters and the average parameters of the optimal distributions obtained using the BO. The details are given in Supplementary Section B.
>
>
>
>
> [1] Lun, S.X., Yao, X.S. and Hu, H.F., 2016. A new echo state network with variable memory length. Information Sciences, 370, pp.103-119.
> [2] Rodan, A. and Tino, P., 2010. Minimum complexity echo state network. IEEE transactions on neural networks, 22(1), pp.131-144.
> [3] Sun, X., Li, T., Li, Q., Huang, Y. and Li, Y., 2017. Deep belief echo-state network and its application to time series prediction. Knowledge-Based Systems, 130, pp.17-29.

---

> > ### Comment · Area_Chair_yr7Z · 2022-11-23
> > **Following up on rebuttal**
> >
> > Dear reviewer,
> >
> > given the other reviews as well as the rebuttal, do you maintain your opinion?
> >
> > Cheers,
> > Your AC

---

> > ### Author Response · Authors · 2022-11-24
> > **Further Analysis and Results**
> >
> > > capacity measure somewhat restricting as a measure of computational power
> >
> > To further validate our hypothesis that we can improve model performance by controlling memory capacity, we did the following empirical studies:
> > 1. We study the t-SNE plots for MRSNN networks as we increase the memory capacity by optimizing the model parameters. We perform the experiment on the CIFAR10-DVS dataset. The result is shown in [Fig. tsne1](https://anonymous.4open.science/r/ICLR23-8EDB/tsne1.jpg) From the figure, we see that there is better separability between the clusters as we increase the memory capacity, which leads to better classification results. This highlights that there might be a correlation between memory capacity and model performance.
> >
> > 2. To further validate the hypothesis, we perform a synthetic control experiment to study the causality between the two variables. Using Fisher’s exact test, we can statistically show that when the memory capacity of the model increases, the accuracy of the model also increases. The details of the experiment are given below.
> >
> > We aim to understand the causal relationship between memory capacity and performance. In particular, we aim to test whether an increase in memory capacity increases the model performance. We leverage a statistical causal inference technique called Synthetic Control [8,9] to demonstrate this. We use a small RSNN neural network with 100 neurons, and the synapses between these neurons are chosen randomly with a fixed probability. We instantiate 10 such random networks and fix the model architectures. Next, we use Fisher’s Exact Tests to do inference with synthetic control using a simple seq-MNIST dataset. The experimental setup is as follows:
> >
> > 1. We use BO to optimize the hyper-parameters of an RSNN (Note this is a standard BO where we search for the individual parameters of all the neurons and the synapses). The objective function of the BO is to maximize the model's generalizability, keeping its memory capacity fixed. The generalization ability is measured using its Hausdorff Dimension, as discussed in previous works [10,11], which is orthogonal to the model performance. Basically, this BO method regularizes the model to have flatter minimas.
> >
> > 2. We evaluate the BO for 20 evaluations and observe the memory capacity and the model performance for each of these evaluations. We observe that as we force the BO to keep the memory capacity fixed, it does not vary much. However, it is interesting to note here that the model performance at each of these steps also does not vary significantly. (defined in point 3)
> >
> > 3. We count the number of times the memory capacity and the model performance increase. We define that a variable (either memory capacity or model performance) ‘increases’ if the difference in the value of the variable at that instant and the moving average of the variable from the past instances is more than 1% (chosen arbitrarily). Hence, we count the number of instances when each of these metrics satisfies this criterion.
> >
> > 4. After 20 evaluations, there are 2 different BO problems that we use:
> > - Case 1: The control group: We keep running this BO where we keep the memory capacity constant and optimize the generalizability of the model
> > - Case 2: The treatment group: We change the BO to optimize the memory capacity of the model
> >
> > 5. In each of the two cases, we observe the accuracy and the memory capacity and calculate if, at each time, the value deviates from the mean (as discussed in step 3). We take the average of the counts and tabulate it as follows:
> >
> > |                                                   |     First 20 steps  (MC constant)    |     Last 20 steps  (MC maximized)    |
> > |:-------------------------------------------------:|:------------------------------------:|:------------------------------------:|
> > |     Acc increase   (avg.   over 10 iterations)    |                   2                 |                   15               |
> > |       No Acc Increase   (avg. over 10 iterations)     |                   18                 |                   5                  |
> >
> > 6. From this table, we calculate the p-value to test against the null hypothesis that the accuracy does not increase with an increase in memory capacity. The p-value calculated is less than 0.0001, which indicates that increasing memory capacity increases the accuracy of the model. The results are shown in [Fig. fig_causal](https://anonymous.4open.science/r/ICLR23-8EDB/fig_causal.jpg)
> >
> > Figure tsne1 URL: https://anonymous.4open.science/r/ICLR23-8EDB/tsne1.jpg
> > Figure fig_causal URL: https://anonymous.4open.science/r/ICLR23-8EDB/fig_causal.jpg

---

### Official Review · Reviewer_LwLn · 2022-10-27

**Confidence:** 3
**Correctness:** 2
**Technical Novelty And Significance:** 2
**Empirical Novelty And Significance:** 2
**Recommendation:** 3

**Clarity, Quality, Novelty And Reproducibility:**

* Clarity:
The paper is written clearly

Quality:
The quality of the theory is hard to evaluate because the analytical results are directly tested analytically.
The numerical evaluation seems to be done carefully, I liked e.g. convergence analysis on the supplement.


Novelty:
The main theoretical results seem to be a slight extension of some previous work (Aceituno 2020). Unfortunately, the main analytical result (Equations 21 and 22) seems more of a heuristic nature. An actual analytical theory of the quantitative effects of heterogeneity of membrane time constant and thresholds in the capacity is missing, so maybe one could write it as a conjecture instead. Also, a discussion of the assumptions and limitations of the analytical derivation is missing. When is it expected to fail? Are the assumptions tested numerically?

Reproducibility:
Because no code is provided, the results are unfortunately not fully reproducible in the current form.


**Strength And Weaknesses:**

Strength:
* It is a rare case of an analytical study of the computational capabilities of a recurrent spiking neural network.

Weaknesses:
* The main finding is that heterogeneity of membrane time constants increases the capacity, however, in biological neurons, the membrane time constant is not a free parameter but is tightly controlled by the resistance across the membrane and the capacitance of the membrane. It is thus not biologically plausible that the brain has the freedom to choose a heterogeneous distribution of membrane time constants. A more plausible candidate for that would be the heterogeneity of synaptic time constants.
* A direct comparison of analytical results and numerical simulations is missing. While figure 3 shows a similar trend as expected from theory, it would be desirable to actually directly compare them.
* table 2: it would be better to measure the firing rates of the neurons in 1/s or 1/tau_m and not as 'spike count'.
* It would be nice to relate this work to other work with recurrent networks of LIF neurons, e.g. Brunel 2000, Jahnke/Memmesheimer/Timme 2009, Deneve/Machens 2016.
* The theoretical part seems to rely on some kind of mean-field assumption that allows to express the covariance of the activity in terms of linearized dynamics and a stationarity assumption. As LIF networks can have a very diverse set of states (e.g. oscillations, synchronization, splay-states, asynchronous irregular states, etc. see e.g. Brunel 2000), it is not clear how this assumption is justified here. Also, no direct numerical tests of the analytical results are being presented, which makes it hard to evaluate them.
* Details the Bayesian Optimization are missing, the numerical experiments are not fully reproducible in the current version.

**Summary Of The Paper:**

The submission studies discrete-time recurrent spiking neural networks of LIF neurons with STDP both analytically and numerically.  The main analytical finding is that heterogeneity of neuron parameters increases the memory capacity of the recurrent spiking neural network. This finding is based on the fact that with increasing heterogeneity of neuron parameters (threshold and membrane time constant), the covariance is decreasing ints covariance. A lower covariance results in a higher capacity. The numerical analysis confirms that. Moreover, a SAGE-attribution analysis reveals that the main contribution to the improved performance comes from heterogeneous membrane time constants.

**Summary Of The Review:**

The submission studies the role of the heterogeneity of membrane time constants and threshold heterogeneity on the capacity of recurrent spiking neural networks.
The two main weaknesses of the submission are a lack of biological plausibility and a lack of discussion of the scope and rigor of the "analytical" theory (which seems to be a slight extension of  (Aceituno 2020), i.e (including discussion of the implicit assumptions and when it would break down) and details (and code) that would make the findings reproducible.

Overall, I cannot recommend the manuscript in the current form for publication.

---

> ### Author Response · Authors · 2022-11-19
> **Response to Weaknesses I**
>
> > The main finding is that heterogeneity of membrane time constants increases the capacity, however, in biological neurons, the membrane time constant is not a free parameter but is tightly controlled by the resistance across the membrane and the capacitance of the membrane. It is thus not biologically plausible that the brain has the freedom to choose a heterogeneous distribution of membrane time constants. A more plausible candidate for that would be the heterogeneity of synaptic time constants.
>
> We agree with the reviewer’s comment and apologize for the confusion regarding when/how the membrane potentials are allowed to change. We are loosely inspired by the neuroscience literature [1-4] that suggests the plausibility of having multiple time constants for neurons and synaptic dynamics in the human brain.   We also apologize if it is not clear from the text, but our model already considers heterogeneity in the STDP parameters (as the reviewer suggested), where we introduce variance in the synaptic time constants.
> However, we also note that although our heterogeneous recurrent spiking neural network is inspired by the brain, the paper's main aim is not to develop a computational neuroscience model of the brain or a bio-mimicking network. We focus on developing an unsupervised machine-learning model for time-series classification and time-series prediction. We accomplish this within the construct of SNN as an ML model by using heterogeneity in the LIF neurons and the STDP dynamics and analytically proving their effects.
> Towards this goal, we also perform empirical studies on existing datasets and introduce the Bayesian Optimization step to search for hyperparameter distributions (membrane time constant for LIF neurons and synaptic time constants for STDP). This Bayesian Optimization process to search for the optimal hyperparameters of the model is performed before training and inference using the model and is generally equivalent to the network architecture search process used in deep learning. Once we have these optimal hyper-parameters, we freeze these hyperparameters, learn (unsupervised) the network parameters (i.e., synaptic weights) of the HRSNN while using the frozen hyperparameters, and generate the final HRSNN model for inference. In other words, the hyperparameters, like the distribution of membrane time constants or the distribution of synaptic time constants for STDP, are fixed during the learning and inference.
>
>
>
> [1] Hansel, D., Mato, G. and Meunier, C., 1995. Synchrony in excitatory neural networks. Neural computation, 7(2), pp.307-337.
> [2] Prescott, S.A., De Koninck, Y. and Sejnowski, T.J., 2008. Biophysical basis for three distinct dynamical mechanisms of action potential initiation. PLoS computational biology, 4(10), p.e1000198.
> [3] Machens, C.K., Romo, R. and Brody, C.D., 2010. Functional, but not anatomical, separation of “what” and “when” in prefrontal cortex. Journal of Neuroscience, 30(1), pp.350-360.
> [4] Douglass, J.K., Wilkens, L., Pantazelou, E. and Moss, F., 1993. Noise enhancement of information transfer in crayfish mechanoreceptors by stochastic resonance. Nature, 365(6444), pp.337-340.

---

> > ### Author Response · Authors · 2022-11-19
> > **Response to Weakness II**
> >
> > > A direct comparison of analytical results and numerical simulations is missing. While figure 3 shows a similar trend as expected from theory, it would be desirable to actually directly compare them.
> > > Also, no direct numerical tests of the analytical results are being presented, which makes it hard to evaluate them.
> >
> > We have added additional results showing the numerical results based on analytical derivations/models (See Supplementary Section B). Here is a brief summary of the key results.
> > For theorem 1, where we state that the memory capacity of a heterogeneous RSNN is greater than the memory capacity of an MRSNN, we evaluated the estimated memory capacity with respect to the heterogeneity in the neuronal parameters. We analytically study the variance of the estimated memory capacity with the change in the heterogeneity of neuronal parameters. We plot the change in the estimated memory capacity, calculated using Eq. 3. We plot this with respect to the neuronal heterogeneity, measured using the entropy of the neuronal parameters for the HRSNN model. The result is plotted in Fig. 7(a) (Supplementary Section B).  We use the HRSNN model with 1000 neurons and sequences of 4,000 random inputs chosen from U[-1; 1]. We see that, as predicted, the memory capacity of the model increases linearly with the increase in heterogeneity within the limits of the application, as proved in Theorem 1. The error bars in Fig. 7(a) represent the standard deviation of the observations. The details of the experiments and results are shown in Supplementary Section B.
> > Similarly, in Theorem 2, we show that the average firing rate for HRSNN is less than that of MRSNN. To analytically show that, we calculate the average firing rate of the heterogeneous spiking neural network for the prediction task during inference. We plot the product of the memory capacity and the mean firing rate from the model to estimate the efficiency, which is shown in Fig 7(b) (Supplementary Section B). From Table 2, we see that using heterogeneity in the STDP parameters reduces the average firing rate while keeping the memory capacity almost equal. This result shows that Heterogeneous STDP leads to sparse activation of neurons, as proved in Theorem 2.  The details of the experiment and the results obtained are shown in Supplementary Section B.
> > In addition to this, we have added a simplified reduction mean-field model of the HRSNN network in Supplementary Section B. However, it must be noted here that complete mean-field modeling of the HRSNN network with heterogeneous LIF neurons and heterogeneous STDP is a fascinating research question but beyond the scope of this paper.
> >
> >
> > > table 2: it would be better to measure the firing rates of the neurons in 1/s or 1/tau_m and not as 'spike count'.
> >
> > We have updated Table 2 to show the average firing rates (1/S) instead of the avg. spike counts shown before
> >
> > > It would be nice to relate this work to other work with recurrent networks of LIF neurons, e.g. Brunel 2000, Jahnke/Memmesheimer/Timme 2009, Deneve/Machens 2016.
> >
> > We thank the reviewer for the suggestion. We have added a section in Supplementary Section B discussing how our work relates to other recurrent networks with LIF neurons. In addition to this, we have also added a discussion on deriving a mean-field reduction model of the HRSNN model, which is added in Suppl. Sec. B. However, it must be noted here that complete mean-field modeling of the HRSNN network with heterogeneous LIF neurons and heterogeneous STDP is a fascinating research question but beyond the scope of this paper. This is also highlighted in the limitations added in the Conclusion.

---

> > > ### Author Response · Authors · 2022-11-19
> > > **Response to Weaknesses III**
> > >
> > > > The theoretical part seems to rely on some kind of mean-field assumption that allows to express the covariance of the activity in terms of linearized dynamics and a stationarity assumption. As LIF networks can have a very diverse set of states (e.g. oscillations, synchronization, splay-states, asynchronous irregular states, etc. see e.g. Brunel 2000), it is not clear how this assumption is justified here.
> > >
> > > We were intrigued by the reviewer’s questions on the relation between our approach and mean-field theory and have now performed a detailed investigation. We have added a subsection in Supplementary Section B to discuss the mean-field approximation of the HRSNN model in detail.
> > > We first point out that the derivation of the covariance of the activity does not directly use the mean-field theory with thermodynamic limit assuming infinite number of neurons. We only used the relationship between the trace of the covariance and the variance of the neuronal states to get the eigenvalues of the covariance.  However, we do realize that the analytical form for the neuronal states used in our derivation assumes an implicit mean-field model of the entire heterogeneous network. We also agree with the reviewer that mean-field assumption for individual may not be justified considering the diverse states exhibited by the heterogeneous population of LIF neurons. However, the implicit mean-field assumption in our approach is for the entire network and not for individual neurons; and hence, does not the above-mentioned limitation.
> > > We use the mean-field approximation to study the entire network, which consists of neurons in different states, as shown by Camera et al. [1]. In their paper, the authors described the mean-field approach ignores fluctuations in the interaction between the units defining the system. Similar results have also been shown by Dumont et al. [2], where the authors show that neural network dynamics emerge from the interaction of spiking cells, and they formulate the problem using the mean-field theory. There are similar other works that study the collective behavior of LIF neurons with different modes of operation.
> > >
> > > > Details the Bayesian Optimization are missing, the numerical experiments are not fully reproducible in the current version.
> > >
> > > We have added more details about the Bayesian Optimization process in Supplementary Section A, including the initial parameter values and the final searched distributions. We have also added further discussion about the methodology of the modified BO and how the Matern kernel is calculated using the Wasserstein distance. The details are given in Supplementary Section B.
> > >
> > >
> > > > Novelty: The main theoretical results seem to be a slight extension of some previous work (Aceituno 2020).
> > >
> > > We prove two main theoretical results in this paper. First, we show that HRSNN models with heterogeneous LIF neurons have a higher memory capacity and second, that HRSNN models with heterogeneous STDP dynamics have a lower spike firing rate when compared to MRSNN models.  We note that the second result has no connection to the Aceituno 2020 work.
> > > We do agree, and already mentioned in the paper, that our first result leverages Aceituno 2020. However, our result showed that heterogeneity in the LIF neurons decreases the neuronal correlations. On the other hand, Aceituno et al showed that decreasing neuronal correlation resulted in increasing the memory capacity of the model. Hence, when we connect our result with Aceituno et al.'s result, we show that parameters' heterogeneity reduces neuronal correlation (our contribution) which improves model's memory capacity (Aceituno et. al. 2020).  Hence, our contribution is not a simple extension or corollary of the work by Aceituno et al.; rather, we leverage their work to establish the connection between neuronal parameter and heterogeneity.
> > >
> > > To highlight the novelty of this work, we have added further explanations of both proofs in Supplementary Section B.
> > >
> > >
> > >
> > >
> > >
> > > [1] La Camera, G., 2022. The mean field approach for populations of spiking neurons. In Computational Modelling of the Brain (pp. 125-157). Springer, Cham.
> > > [2] Dumont, G. and Gabriel, P., 2020. The mean-field equation of a leaky integrate-and-fire neural network: measure solutions and steady states. Nonlinearity, 33(12), p.6381.

---

> > > > ### Author Response · Authors · 2022-11-19
> > > > **Response to Weaknesses IV**
> > > >
> > > > > Unfortunately, the main analytical result (Equations 21 and 22) seems more of a heuristic nature. An actual analytical theory of the quantitative effects of heterogeneity of membrane time constant and thresholds in the capacity is missing, so maybe one could write it as a conjecture instead.
> > > >
> > > > We have added a complete analytical derivation of erstwhile Eqs 21 and 22 for better understanding. Further, we have added an extensive discussion about the quantitative effects of the heterogeneity of membrane time constants and how $\mathcal{J}$ given in Eq. 46, using the eigenvalues of the covariance matrix, is a good estimator describing the heterogeneity of the neuronal parameters. However, it must be noted here that we do not intend to derive an analytical relation between memory capacity and the heterogeneity of the membrane time constants but simply intend to prove that heterogeneity in the neuronal parameters leads to a model with higher memory capacity.
> > > >
> > > >
> > > > >Also, a discussion of the assumptions and limitations of the analytical derivation is missing. When is it expected to fail? Are the assumptions tested numerically?
> > > >
> > > > We have added further discussion about the limitations of the paper in the Conclusion. Further, we have added a subsection discussing the general assumptions we used for the paper. The details about the assumptions are discussed in Section 2 of the paper.
> > > >
> > > > > Reproducibility: Because no code is provided, the results are unfortunately not fully reproducible in the current form.
> > > >
> > > > We plan to make the codes used in this project publicly available once the paper is accepted.

---

> > > > > ### Comment · Area_Chair_yr7Z · 2022-11-23
> > > > > **Following up on rebuttal**
> > > > >
> > > > > Dear reviewer,
> > > > >
> > > > > the authors tried to respond to your criticism regarding heterogeneity of membrane time constants, as well as comparisons. Has the rebuttal influenced your opinion?
> > > > >
> > > > > Cheers,
> > > > > Your AC

---

> > > > > ### Comment · Reviewer_LwLn · 2022-11-24
> > > > > **Response to rebuttal**
> > > > >
> > > > > We thank the author for the comprehensive response and all the changes in the manuscript. We find the clarity and quality of the submission has significantly improved, and we increase the recommendation/score.
> > > > >
> > > > > Here are further comments and issues:
> > > > > * The changes to preliminaries and definitions are very helpful.
> > > > > * Making the assumptions more transparent in the "proof" was also helpful.  Especially, that the proof assumes a Hawkes process. It would be important to make clear where the innovation compared to previous mean-field work on stochastic LIF network/Hawkes processes is, e.g. the mean-field work on stochastic LIF networks by Ocker (Dynamics of stochastic integrate-and-fire networks https://arxiv.org/abs/2202.07751) and (Training and Spontaneous Reinforcement of Neuronal Assemblies by Spike Timing Plasticity https://academic.oup.com/cercor/article/29/3/937/4836778) and the relevant work cited there.
> > > > > * The details on the Bayesian Optimization in the main text and supplement are also helpful.
> > > > > * The limitation paragraph is helpful.
> > > > >
> > > > > After reading the updated manuscript, the following issues
> > > > > * A direct comparison of theory and numerical experiments seems still to be missing. While
> > > > > * I am a bit confused about the units of the avg. firing rate. It is quantified as 1/\tilde S in table 2, however, I thought \tilde S is the average  spike count, so its inverse seems not to be a firing rate. Shouldn't firing rates have units of Hz or 1/\tau or spikes  per second? Right now, it is hard to understand if the reported firing rates in table 2 are meaningful.
> > > > > * "Percentange" should be "Percentage"
> > > > > * "The Memory capacity" should be "The memory capacity"
> > > > > * The references got a little bit mixed up e.g. "Deneve et al. Denève & Machens (2016)", "Brunel Brunel (2000)", Jaeger Jaeger (2002), Simsekli et al. Simsekli et al. (2020)Aceituno et al. Aceituno et al. (2020)
> > > > > * "Percentange" should be "Percentage"
> > > > > * It is helpful that in the supplement it is acknowledged that "a complete mean-field derivation of the HRSNN model with heterogeneous LIF neurons, and heterogeneous STDP dynamics" is still missing.
> > > > > * Figure 7 in the supplement is helpful as a qualitative confirmation of parts of the theory, but more quantitative comparisons are still missing to give evidence for the whole chain of arguments ("H increases → covariance decreases → neurons become less correlated").  E.g. depict covariances for different heterogeneity and show the effect of various other parameters, e.g. coupling strength, firing rate, LIF parameters etc. Also a numerical of the assumptions is missing ((i) finite size effects are negligible (N e/i ≫ 1) (ii) the firing rate of presynaptic neurons is governed by a Poisson
> > > > > process (iii) the population firing rate averaged over q and τ m is a good approximation to the average presynaptic input rate (iv) a single p.d.f. function is sufficient to describe the population behavior,) (finite N, .
> > > > > Also, it is unclear how well the theory works for coupled Hawkes process, how well for LIF networks.

---

> > > > > > ### Author Response · Authors · 2022-11-28
> > > > > > **Response to the Reviewer's Followup Questions 1**
> > > > > >
> > > > > > We thank the reviewer for their in-depth discussion about the paper. Their suggestions have helped us improve the quality of the paper greatly. The responses to the followup questions asked by the reviewer is discussed as follows:
> > > > > >
> > > > > > >  It would be important to make clear where the innovation compared to previous mean-field work on stochastic LIF network/Hawkes processes is, e.g. the mean-field work on stochastic LIF networks by Ocker (Dynamics of stochastic integrate-and-fire networks https://arxiv.org/abs/2202.07751) and (Training and Spontaneous Reinforcement of Neuronal Assemblies by Spike Timing Plasticity https://academic.oup.com/cercor/article/29/3/937/4836778) and the relevant work cited there.
> > > > > >
> > > > > > **>>**  We thank the reviewers for bringing these papers on mean-field theory for LIF networks/Hawkes processes. Ocker et al. [1] developed a statistical mean-field theory for networks of integrate-and-fire neurons with stochastic spike emission where they constructed the full joint probability density functional of a neuronal network's spike trains and membrane voltages. In the second work, Ocker et al. [2] developed a low-dimensional mean-field theory for STDP in recurrent networks and showed the emergence of assemblies of strongly coupled neurons with shared stimulus preferences. We have added these works in our references.
> > > > > > These papers presented important advancements in understanding the dynamics of LIF networks. However, the objective of these papers is very different from ours. The primary aim of our work is not to give a better mean-field model of heterogeneous recurrent spiking neural network models. Our focus is to develop an unsupervised spiking neural network model for prediction and classification tasks and show that using heterogeneity in the LIF neurons and the STDP dynamics helps us engineer better learning models since that helps us increase the memory capacity and reduce the number of spike activations. We introduce a Bayesian Optimization step to search for hyperparameter distributions (membrane time constant for LIF neurons and synaptic time constants for STDP) and present empirical studies on datasets. Therefore, the core innovations of our work lie in using heterogeneity to improve unsupervised SNN-based ML models for classification and prediction [1,2].
> > > > > > We also note that, deriving a mean-field model for a network of recurrently connected LIF neurons, [1] did not consider the STDP-based learning rule but instead studied the coupling effects to study their interactions. On the other hand, our paper uses heterogeneous LIF neurons and STDP learning rules to learn the synaptic weights (neuron coupling). In [2], the authors studied the mean-field effect of STDP on an RSNN model. However, they did not consider heterogeneity in the LIF parameters or the STDP dynamics. As mentioned above, our focus is on using heterogenous LIF and synaptic dynamics to improve memory capacity and spike-efficiency of RSNN.
> > > > > > We do believe that derivation of a complete mean-field model for an HRSNN with heterogeneous LIF neurons and heterogeneous STDP dynamics could be an interesting future direction of research, but out of the scope for this paper. We also feel that [2] could provide a foundation for such a future study. Hence, we will add these papers as references in the final version of our paper and add a sentence stating on this regard.
> > > > > >
> > > > > >
> > > > > > > A direct comparison of theory and numerical experiments seems still to be missing.
> > > > > >
> > > > > > **>>**  We have shown the relation between heterogeneity in LIF parameters and memory capacity. To validate this, we show that increasing heterogeneity in LIF parameters reduces the correlation between neurons. This is shown in [Fig](https://anonymous.4open.science/r/ICLR23-8EDB/iclr_reb_fig1.jpg). Further, previous works have shown that reducing correlation increases memory capacity. [3] Thus, these results add numerical confirmation to our proof.
> > > > > > Similarly, for theorem2, we showed that heterogeneity in STDP dynamics reduces the average neuron firing rate compared to the MRSNN network with homogeneous STDP. To numerically validate this, we evaluated the average firing rates of the neurons for the four cases viz., MRSNN (with homogeneous LIF neurons and homogeneous STDP), HRSNN (with heterogeneous LIF, homogeneous STDP), HRSNN (with homogeneous LIF, heterogeneous STDP), HRSNN (with heterogeneous LIF, heterogeneous STDP). We plot the results for a smaller network with 100 neurons and a Poisson input process. The results are shown in [Fig.](https://anonymous.4open.science/r/ICLR23-8EDB/reb_hist.jpg) We see that the heterogeneity in the STDP dynamics reduces the average firing rate of the neurons.

---

> > > > > > > ### Author Response · Authors · 2022-11-28
> > > > > > > **Response to Reviewer's Followup Questions 2**
> > > > > > >
> > > > > > > > I am a bit confused about the units of the avg. firing rate. It is quantified as 1/\tilde S in table 2, however, I thought \tilde S is the average spike count, so its inverse seems not to be a firing rate. Shouldn't firing rates have units of Hz or 1/\tau or spikes per second? Right now, it is hard to understand if the reported firing rates in table 2 are meaningful.
> > > > > > >
> > > > > > > **>>** We believe there was some misunderstanding. We have updated the table to include the average spike rate, calculated as the ratio of the moving average of the number of spikes in a time interval $T$. For this experiment, we choose $T= 4ms$ and a rolling time span of $2ms$, which is repeated until the first spike appears in the final layer. Following the works of Paul et al. [4], we show that the normalized average spike rate is the total number of spikes generated by all neurons in an RSNN averaged over the time interval $T$.  The updated table is given as follows:
> > > > > > > |                        |                      Method                     | SHD  (Classification) |                                                     |                   Spike  Efficiency                  | Lorenz System (Prediction) |                                                     |                                                                   |
> > > > > > > |:----------------------:|:-----------------------------------------------:|:---------------------:|-----------------------------------------------------|:----------------------------------------------------:|:----------------------------------:|-----------------------------------------------------|:-----------------------------------------------------------------:|
> > > > > > > |          Task          |                      Method                     |     Accuracy ($A$)    | Normalized Avg. Firing Rate ($\frac{\tilde{S}}{T}$) | Efficiency $\hat{\mathcal{E}} = \frac{A}{\tilde{S}}$ |                NRMSE               | Normalized Avg. Firing Rate ($\frac{\tilde{S}}{T}$) | Efficiency $\hat{\mathcal{E}} = \frac{1}{NRMSE \times \tilde{S}}$ |
> > > > > > > |     Unsupervised  RSNN |    MRSNN (Homogeneous LIF,  Homogeneous STDP)   |         73.58         | -0.508                                              |                $18.44 \times 10^{-3}$                |                0.395               | -0.768                                              |                       $0.787 \times 10^{-3}$                      |
> > > > > > > |                        |  HRSNN  (Heterogeneous LIF,  Homogeneous STDP)  |         78.87         | 0.277                                               |                $17.19 \times 10^{-3}$                |                0.203               | -0.143                                              |                       $1.302 \times 10^{-3}$                      |
> > > > > > > |                        |  HRSNN  (Homogeneous LIF,  Heterogeneous STDP)  |         74.03         | -1.292                                              |                $22.47 \times 10^{-3}$                |                0.372               | -1.102                                              |                       $0.932 \times 10^{-3}$                      |
> > > > > > > |                        | HRSNN  (Heterogeneous LIF,  Heterogeneous STDP) |         80.49         | -1.154                                              |                $24.35 \times 10^{-3}$                |                0.195               | -1.018                                              |                       $1.725 \times 10^{-3}$                      |
> > > > > > > |      RSNN  with BP     |         MRSNN-BP (Homogeneous LIF,  BP)         |         81.42         | 0.554                                               |                 $16.9 \times 10^{-3}$                |                0.182               | 0.857                                               |                       $1.16 \times 10^{-3}$                       |
> > > > > > > |                        |        HRSNN-BP (Heterogeneous LIF,  BP)        |         83.54         | 1.292                                               |                $15.42 \times 10^{-3}$                |                0.178               | 1.233                                               |                       $1.09 \times 10^{-3}$                       |
> > > > > > > |                        |             Adaptive SRNN \cite{yin}            |         84.46         | 0.831                                               |                $17.21 \times 10^{-3}$                |               0.174*               | 0.941                                               |                       $1.19 \times 10^{-3}$*                      |

---

> > > > > > > > ### Author Response · Authors · 2022-11-28
> > > > > > > > **Response to Reviewer's Followup Questions 3**
> > > > > > > >
> > > > > > > > >"Percentange" should be "Percentage".
> > > > > > > > "The Memory capacity" should be "The memory capacity"
> > > > > > > > The references got a little bit mixed up e.g. "Deneve et al. Denève & Machens (2016)", "Brunel Brunel (2000)", Jaeger Jaeger (2002), Simsekli et al. Simsekli et al. (2020)Aceituno et al. Aceituno et al. (2020)
> > > > > > > > "Percentange" should be "Percentage"
> > > > > > > >
> > > > > > > >
> > > > > > > > **>>**  We thank the reviewer for their suggestions and for pointing out these mistakes. We shall make the changes in the final version of the paper once it is accepted.
> > > > > > > >
> > > > > > > >
> > > > > > > > > Figure 7 in the supplement is helpful as a qualitative confirmation of parts of the theory, but more quantitative comparisons are still missing to give evidence for the whole chain of arguments ("H increases → covariance decreases → neurons become less correlated"). E.g. depict covariances for different heterogeneity and show the effect of various other parameters, e.g. coupling strength, firing rate, LIF parameters etc.
> > > > > > > >
> > > > > > > > **>>** As the reviewer suggested, we have performed further quantitative evaluations to support our theory. We will be happy to add these results in the appendix of the final paper. The results obtained are listed as follows:
> > > > > > > >
> > > > > > > >
> > > > > > > > **Impact of Heterogeneity on Covariance:**  We plot the covariance matrices for different levels of heterogeneity   \mathcal{J} (Eq. 46) for a small network with 50 neurons. The covariance matrix is calculated by taking the average neuronal states before the appearance of the first spike in the final layer. We see that as the heterogeneity in the neuronal parameters increases, the correlation between the neurons decreases. The results are shown in [Fig](https://anonymous.4open.science/r/ICLR23-8EDB/iclr_reb_fig1.jpg).
> > > > > > > >
> > > > > > > > **Impact of Heterogeneity on Principal Components:** From the covariance plots, we see that increasing $\mathcal{J}$. reduces the correlation between neurons. We also plot the probability density functions of the eigenvalues of the covariance matrix of the neurons with increasing heterogeneity in the neuronal parameters. We see that with higher heterogeneity in the neuronal parameters $\mathcal{J}$., the distribution of the eigenvalues of the covariance becomes flatter. This signifies that the covariance matrix has a lower variance for higher $\mathcal{J}$. A flatter distribution also indicates that a larger number of principal components are active. This supports our hypothesis that heterogeneity in the neuronal parameters increases the number of principal components and helps increase the model's memory capacity. The result is shown in this [Fig.](https://anonymous.4open.science/r/ICLR23-8EDB/iclr_het_eig.jpg)
> > > > > > > >
> > > > > > > > **Impact of Heterogeneity in STDP on Firing Rate:** We plot the mean firing rate of the neurons for the four types of HRSNNs and MRSNN with homogeneous LIF and STDP dynamics. We plot the results for a smaller network with 100 neurons and a Poisson input process. We see that the MRSNN model shows a much higher firing rate, especially at a higher frequency, demonstrating that MRSNN requires a significantly more number of spikes compared to the HRSNN model. The result is shown in [Fig.](https://anonymous.4open.science/r/ICLR23-8EDB/reb_hist.jpg)
> > > > > > > >
> > > > > > > > **Coupling Strength:** We note here that in this paper, we use (homogeneous or heterogeneous) STDP to learn the synaptic conductance connecting various neurons in the SNN. Therefore, we do not control the synaptic coupling strength as independent variables and hence, cannot perform control experiments with various extents of coupling strength. An interesting future extension of the results will be to quantify the coupling strength for HRSNN with heterogeneity in LIF and STDP dynamics. We can leverage McKenzie et al. [5], where the authors proposed statistical tools to estimate synaptic coupling dynamics from spike-spike correlations.

---

> > > > > > > > > ### Author Response · Authors · 2022-11-28
> > > > > > > > > **Response to Reviewer's Followup Questions 4**
> > > > > > > > >
> > > > > > > > > > Also a numerical of the assumptions is missing ((i) finite size effects are negligible (N e/i ≫ 1) (ii) the firing rate of presynaptic neurons is governed by a Poisson process (iii) the population firing rate averaged over $q$ and $\tau_m$ is a good approximation to the average presynaptic input rate (iv) a single p.d.f. function is sufficient to describe the population behavior,) (finite N, .)
> > > > > > > > >
> > > > > > > > > **>>**  We thank the reviewers for pointing this out. We will update the Suppl. Sect. B adding the numerical assumptions to improve comprehensibility. We also thank the reviewers for pointing out some of these assumptions.
> > > > > > > > >
> > > > > > > > > > Also, it is unclear how well the theory works for coupled Hawkes process, how well for LIF networks.
> > > > > > > > >
> > > > > > > > > **>>**  The notion of using point process models, especially the interactive Hawkes processes, to model the spiking dynamics of LIF network dynamics has been studied in the literature previously [6-10]. We leverage these results to prove that heterogeneity in the synaptic dynamics can help reduce the spike count, as already discussed in the paper. We highlighted the key assumptions used in deriving the results in the Suppl. Sec. C. We apologize if there is still confusion, and we will add more in-depth discussion in the final manuscript as discussed below.
> > > > > > > > > In their paper, Löcherbach et al. [6] provide a survey of some aspects of the study of Hawkes processes in high dimensions to model biological neural systems and study their long-term behavior. Galves et al. provided an overview of point processes used as stochastic models for interacting neurons both in discrete and continuous time [7]. Similarly, Hawkes processes have met a recent interest in the mathematical neuroscience literature for their ability to model the dependence of the activity of a neuron in the history of the network [8-13]. Other works have also used a nonlinear interactive Hawkes process to model spiking neural networks with excitatory and inhibitory neurons [14-18]. Taking inspiration from these works, we use a microscopic model describing a large network of interacting neurons that can generate oscillations in a macroscopic frame. In the model, the activity of each neuron is represented by a point process indicating the successive times at which the neuron emits a spike, where each realization of this point process is the spike train. We take the spiking intensity of a neuron as the probability of emitting a spike during the next time instant, depending on the past history of the neuron and the activity of other neurons in the network. The neurons interact through their synapses. This means that a spike of a pre-synaptic neuron leads to an increase of the membrane potential of the post-synaptic neuron if the synapse is excitatory or to a decrease if the synapse is inhibitory, possibly after some delay, like the process of synaptic integration. When the membrane potential reaches a certain upper threshold, the neuron fires a spike. Thus, excitatory inputs from the neurons in the network increase the firing intensity, and inhibitory inputs decrease it. Hawkes processes provide good models of this synaptic integration phenomenon by the structure of their intensity processes. In this paper, we use a general class of mean-field interacting Hawkes processes is considered, modeling the reciprocal interactions between a population of excitatory neurons and a population of inhibitory neurons.

---

> > > > > > > > > > ### Author Response · Authors · 2022-11-28
> > > > > > > > > > **References**
> > > > > > > > > >
> > > > > > > > > > 1. Ocker, G.K., 2022. Dynamics of stochastic integrate-and-fire networks. arXiv preprint arXiv:2202.07751.
> > > > > > > > > > 2. Ocker, G.K. and Doiron, B., 2019. Training and spontaneous reinforcement of neuronal assemblies by spike timing plasticity. Cerebral Cortex, 29(3), pp.937-951.
> > > > > > > > > > 3. Aceituno, P.V., Yan, G. and Liu, Y.Y., 2020. Tailoring echo state networks for optimal learning. iscience, 23(9), p.101440.
> > > > > > > > > > 4. Paul, A., Wagner, S. and Das, A., 2022. Learning in Feedback-driven Recurrent Spiking Neural Networks using full-FORCE Training. arXiv preprint arXiv:2205.13585.
> > > > > > > > > > 5. McKenzie, S., Huszár, R., English, D.F., Kim, K., Christensen, F., Yoon, E. and Buzsáki, G., 2021. Preexisting hippocampal network dynamics constrain optogenetically induced place fields. Neuron, 109(6), pp.1040-1054.
> > > > > > > > > > 6. Löcherbach, E., 2017. Spiking neurons: interacting Hawkes processes, mean field limits and oscillations. ESAIM: Proceedings and Surveys, 60, pp.90-103.
> > > > > > > > > > 7. Galves, A. and Löcherbach, E., 2016. Modeling networks of spiking neurons as interacting processes with memory of variable length. Journal de la Société Française de Statistique, 157(1), pp.17-32.
> > > > > > > > > > 8. Mascart, C., 2021. Efficient simulation of point processes with applications to neurosciences (Doctoral dissertation, Université Côte d'Azur, Nice, France).
> > > > > > > > > > 9. Pfaffelhuber, P., Rotter, S. and Stiefel, J., 2022. Mean-field limits for non-linear Hawkes processes with excitation and inhibition. Stochastic Processes and their Applications, 153, pp.57-78.
> > > > > > > > > > 10. Galves, A. and Löcherbach, E., 2016. Modeling networks of spiking neurons as interacting processes with memory of variable length. Journal de la Société Française de Statistique, 157(1), pp.17-32.
> > > > > > > > > > 11. Gerhard, F., Deger, M. and Truccolo, W., 2017. On the stability and dynamics of stochastic spiking neuron models: Nonlinear Hawkes process and point process GLMs. PLoS computational biology, 13(2), p.e1005390.
> > > > > > > > > > 12. Zhou, F., Zhang, Y. and Zhu, J., 2020. Efficient inference of flexible interaction in spiking-neuron networks. arXiv preprint arXiv:2006.12845.
> > > > > > > > > > 13. Duval, C., Luçon, E. and Pouzat, C., 2022. Interacting Hawkes processes with multiplicative inhibition. Stochastic Processes and their Applications, 148, pp.180-226.
> > > > > > > > > > 14. Chevallier, J., Cáceres, M.J., Doumic, M. and Reynaud-Bouret, P., 2015. Microscopic approach of a time elapsed neural model. Mathematical Models and Methods in Applied Sciences, 25(14), pp.2669-2719.
> > > > > > > > > > 15. Chornoboy, E.S., Schramm, L.P. and Karr, A.F., 1988. Maximum likelihood identification of neural point process systems. Biological cybernetics, 59(4), pp.265-275.
> > > > > > > > > > 16. Hansen, N.R., Reynaud-Bouret, P. and Rivoirard, V., 2015. Lasso and probabilistic inequalities for multivariate point processes. Bernoulli, 21(1), pp.83-143.
> > > > > > > > > > 17. Reynaud-Bouret, P., Rivoirard, V., Grammont, F. and Tuleau-Malot, C., 2014. Goodness-of-fit tests and nonparametric adaptive estimation for spike train analysis. The Journal of Mathematical Neuroscience, 4(1), pp.1-41.
> > > > > > > > > > 18. Colombani, L., 2022. On asymptotic behaviour of stochastic processes on neuroscience (Doctoral dissertation, Université Paul Sabatier-Toulouse III).

---

> > > > > > ### Author Response · Authors · 2022-12-02
> > > > > > **Follow-up discussion**
> > > > > >
> > > > > > We would appreciate it if you could please give us feedback on whether our response has addressed your concerns. We will be more than happy to discuss this further with you if you have any other concerns regarding our rebuttal. We want to thank you again for your valuable time and feedback in improving the quality of the paper.
> > > > > > Also, we observed that the score was changed to 5 after the first round of rebuttal, but we see it is changed back to 3 again. We believe it might be a bug in openreview. If not, we would really appreciate it if the reviewer pointed out any issues they may have observed based on the responses to the follow-up questions. We will be happy to answer them to the best of our ability.
> > > > > >
> > > > > > Thank You.
> > > > > > Paper 4721 Authors

---

> > > > > > > ### Comment · Reviewer_LwLn · 2022-12-05
> > > > > > > **No bug in openreview**
> > > > > > >
> > > > > > > After careful consideration of the long constructive discussion and the most recent version of the manuscript, I cannot recommend the manuscript in its current form for publication at ICLR. My reasons are already pointed out above, but I'll repeat them here:
> > > > > > >
> > > > > > >
> > > > > > > * The initial "analytical proof for spiking networks" contained assumptions that cannot be applied to trained spiking networks, as the authors confirmed. In order to fix this, the authors change their model from leaky integrate-and-fire neurons to Hawkes processes (which are mathematically much more easily treatable than spiking neural networks). I interpret this modification as an admission that the prior proof was insufficient. Unfortunately, LIF networks are not Hawkes processes after training, therefore it is not obvious how well the proposed "theory" accounts for the numerical findings. The theory is at most a weak basis for the claim because it makes no quantitatively testable predictions and offers no numerical comparison to computer simulations.
> > > > > > >
> > > > > > > * The authors admitted that the variety of membrane time constants is not biologically plausible.
> > > > > > >
> > > > > > > * Because the code is currently missing, it is impossible to replicate the study's main conclusions.
> > > > > > >
> > > > > > > I'd be happy to change my score, but I think at least the first two points are difficult to fix at this point.

---

> > > > > > > > ### Author Response · Authors · 2022-12-06
> > > > > > > > **Response to Clarify the Reviewer's Concerns**
> > > > > > > >
> > > > > > > > We respect the reviewers’ opinions and appreciate that they have clarified the reasons behind reversing the scores providing us the opportunity to respond and clarify our views.
> > > > > > > >
> > > > > > > > > The initial "analytical proof for spiking networks" contained assumptions that cannot be applied to trained spiking networks, as the authors confirmed. In order to fix this, the authors change their model from leaky integrate-and-fire neurons to Hawkes processes (which are mathematically much more easily treatable than spiking neural networks). I interpret this modification as an admission that the prior proof was insufficient. Unfortunately, LIF networks are not Hawkes processes after training, therefore it is not obvious how well the proposed "theory" accounts for the numerical findings. The theory is at most a weak basis for the claim because it makes no quantitatively testable predictions and offers no numerical comparison to computer simulations.
> > > > > > > >
> > > > > > > > **>>**  We would like to clarify that we do not change the neuron model from leaky integrate and fire neurons to the Hawkes process. Our results and proofs are based on a network of LIF neurons.
> > > > > > > > We model the spikes generated by the LIF neurons using an interactive Hawkes Process as studied in previous works [1-4]. We use a general class of mean-field interacting nonlinear Hawkes Process to model the reciprocal interaction between the excitatory and inhibitory neuronal populations of LIF neurons. Moreover, we don’t use a standard Hawkes process owing to the difficulty of modifying it to accommodate inhibition which has been studied extensively, as the reviewer mentioned [5-6]. We use the interacting Hawkes process with multiplicative inhibition, which has been shown (theoretically and empirically) to model voltage-based LIF models [4,7].
> > > > > > > > We also note that the use of the Hawkes process is not a key contribution of the paper but was simply adopted to model the spikes in a network of LIF neurons and show how heterogeneity in STDP dynamics in the synapses can lead to a lesser firing rate. The key findings of the paper are that (i) heterogeneity in neuronal parameters increases memory capacity, and (ii) heterogeneity in STDP reduces the average firing rate. We have presented numerical results to verify this trend.
> > > > > > > >
> > > > > > > > References:
> > > > > > > > 1. Lambert, R.C., Tuleau-Malot, C., Bessaih, T., Rivoirard, V., Bouret, Y., Leresche, N. and Reynaud-Bouret, P., 2018. Reconstructing the functional connectivity of multiple spike trains using Hawkes models. Journal of neuroscience methods, 297, pp.9-21.
> > > > > > > > 2. Löcherbach, E., 2017. Spiking neurons: interacting Hawkes processes, mean field limits and oscillations. ESAIM: Proceedings and Surveys, 60, pp.90-103.
> > > > > > > > 3. Pfaffelhuber, P., Rotter, S. and Stiefel, J., 2022. Mean-field limits for non-linear Hawkes processes with excitation and inhibition. Stochastic Processes and their Applications, 153, pp.57-78.
> > > > > > > > 4. Duval, C., Luçon, E. and Pouzat, C., 2022. Interacting Hawkes processes with multiplicative inhibition. Stochastic Processes and their Applications, 148, pp.180-226.
> > > > > > > > 5. Chornoboy, E.S., Schramm, L.P. and Karr, A.F., 1988. Maximum likelihood identification of neural point process systems. Biological cybernetics, 59(4), pp.265-275.
> > > > > > > > 6. Brémaud, P. and Massoulié, L., 1996. Stability of nonlinear Hawkes processes. The Annals of Probability, pp.1563-1588.
> > > > > > > > 7. Bonnet, A., Herrera, M.M. and Sangnier, M., 2022. Inference of multivariate exponential Hawkesprocesses with inhibition and application toneuronal activity. arXiv preprint arXiv:2205.04107.

---

> > > > > > > > > ### Author Response · Authors · 2022-12-06
> > > > > > > > > **Response to Clarify the Reviewer's Concerns Part II**
> > > > > > > > >
> > > > > > > > > >The authors admitted that the variety of membrane time constants is not biologically plausible.
> > > > > > > > >
> > > > > > > > > **>>** We believe this is maybe a misunderstanding. We did not admit that the variety of membrane time constants is not biologically plausible; rather, we mentioned that there is a biological inspiration behind using multiple membrane time constants, as mentioned in prior works [1-2].
> > > > > > > > > We mentioned that we do not change the membrane time constants with time (as the reviewer had initially asked) during learning or inference using an HRSNN. We said that the membrane time constants are fixed for training and inference after the Bayesian Optimization step. In other words, we envision RSNN to have various neurons with different time constants, but a given neuron has the same time constant over time.
> > > > > > > > >
> > > > > > > > > Ultimately, we would like to reiterate that we do. not aim (and claim) to develop a biologically plausible model. We seek an unsupervised ML model that leverages spike-based information processing and STDP-based unsupervised learning. Our goal is to explore whether adding heterogeneity in the neuronal and synaptic parameters improves the performance of the ML model. We apologize if this has not been clear before.
> > > > > > > > >
> > > > > > > > > [1] Koch, C. and Laurent, G., 1999. Complexity and the nervous system. Science, 284(5411), pp.96-98.
> > > > > > > > > [2] Gjorgjieva, J., Drion, G. & Marder, E. Computational implications of biophysical diversity and multiple timescales in neurons and synapses for circuit performance. Curr. Opin. Neurobiol. 37, 44–52 (2016).
> > > > > > > > >
> > > > > > > > >
> > > > > > > > >
> > > > > > > > > > Because the code is currently missing, it is impossible to replicate the study's main conclusions.
> > > > > > > > >
> > > > > > > > > **>>**  We will upload the full source codes with proper documentation once the paper is accepted and de-anonymized.

---

> ### Author Response · Authors · 2022-11-19
> **Summary of Changes**
>
> We thank the reviewer for their feedback. We hope we have addressed their concerns and questions regarding the paper and hope they will reconsider their rating. Taking into account the feedback from the reviewer, the summary of the changes is listed as follows:
>
> 1. We added further explanations for both theorems 1 and 2 to better explain the derivations and make the paper easier to read
> 2. We added a subsection discussing some analytical simulation results to demonstrate the effect of heterogeneity in LIF neurons vs. memory capacity. We also demonstrate how heterogeneity in STDP dynamics improves the spike efficiency of the model.
> 3. We added a simplified mean-field reduction model of the HRSNN network and derived an approximate mean-field firing rate.
> 4. We updated Table 2 to report the mean firing rate measured as (1/S) instead of the number of spikes.
> 5. We discussed how the HRSNN network relates to similar recurrent LIF networks and balanced spiking networks used in the current literature.
> 6. We added further discussions about the BO including the details of the parameters used and the final distributions of the hyperparameters obtained.
> 7. We added a discussion on the limitations of the current work, which is presented in the Conclusion of the main text.
> 8. We discussed the assumptions we made and when the theoretical analysis failed. We present the discussion in Supp Sec B.

---

### Official Review · Reviewer_Rdb2 · 2022-11-03

**Confidence:** 4
**Clarity, Quality, Novelty And Reproducibility:** Some notations are a bit confusing. T…
**Correctness:** 3
**Technical Novelty And Significance:** 3
**Empirical Novelty And Significance:** Not applicable
**Recommendation:** 6

**Strength And Weaknesses:**

**Pros**

Although using a heterogeneous time constant is not novel in this area, the elaboration upon its effect on the memory capacity of SNNs is clearly novel to me.

**Cons**

1. There are some things that need to be clarified or properly placed notations.
2. The statement in Lemma 3.2.1, that is, heterogeneous dynamics reduce the variance of membrane voltages and thus reduce the rate of spikes, seems to omit the discussion of the mean of membrane voltages.

**Questions**

1. The notations are a bit confusing even when the authors have made a table of notations. Here is a non-exhaustive lists
   -  ${\bf{w}}^{\rm{in}}$ first appears in the statement of Lemma 3.1.1, but is first defined in short proof.
   -  In Eq.2, the authors choose $[\cdot]$ to express discretized variable and distinguish it from continuous one $(\cdot)$. However, in Eq.4 and many following equations with integer $\tau$, they still use $(\cdot)$.
   - When using $\Delta$, only the meanings of $\Delta w$ and $\Delta t$ are clearly given. Notations like $\Delta\text{Var}$ in Lemma 3.2.1 are not defined and can be confusing.
2. The statement in Lemma 3.2.1 that a lower variance of membrane voltage distribution leads to a lower firing rate is not sufficient. A zero variance (constant) voltage but higher than the threshold leads to persistent firing.

**Summary Of The Paper:**

The paper makes an endeavor to enhance the memory capacity $\mathcal{C}$ with heterogeneity in neuronal dynamics, which refers to different membrane time constants. The authors also show heterogeneous STDP, which means using a probabilistic synaptic time constant, helps to suppress the firing rate while the memory capacity is maintained. The results are both analytical and empirical.

**Summary Of The Review:**

Many contents are confusing, and need to be clarified.

---

> ### Author Response · Authors · 2022-11-19
> **Response to Cons & Questions**
>
> We thank the reviewer for their feedback. We hope we have addressed their concerns and questions regarding the paper and hope they will reconsider their rating.
>
> > There are some things that need to be clarified or properly placed notations.
> > The notations are a bit confusing even when the authors have made a table of notations.
>
> We thank the reviewer for their feedback. We have carefully revised the paper and corrected the inconsistencies in the notations to the best of our abilities.
>
> > The statement in Lemma 3.2.1, that is, heterogeneous dynamics reduce the variance of membrane voltages and thus reduce the rate of spikes, seems to omit the discussion of the mean of membrane voltages.
>
> > The statement in Lemma 3.2.1 that a lower variance of membrane voltage distribution leads to a lower firing rate is not sufficient. A zero variance (constant) voltage but higher than the threshold leads to persistent firing.
>
> We thank the reviewers for pointing this out about Lemma 3.2.1. In light of the reviews, we have improved the proof of Lemma 3.2.1. We now model the neural spike trains as a multivariate nonlinear Hawkes process model with inhibition. Hence, we show that using heterogeneous STDP dynamics leads to increased synaptic noise, which leads to greater inhibition of post-synaptic spikes during the neural oscillation, which leads to a decrease in the average firing rate of the model. Further details and the complete proof are shown in Supplementary Section B. We believe this is much more robust proof that also addresses the main concerns that arose from the previous proof. We would again like to thank the reviewers for pointing this out, as we believe it leads to a more robust and complete proof.
>
>
> > $\mathbf{w}^{in}$  first appears in the statement of Lemma 3.1.1, but is first defined in short proof.
>
> We thank the reviewer for pointing this out. We have made the necessary corrections in the paper.
>
> > In Eq.2, the authors choose $[.]$ to express discretized variable and distinguish it from continuous one $(.)$. However, in Eq.4 and many following equations with integer $\tau$, they still use $(.)$.
>
> We are sorry for this confusion; the discretized version of the voltage was shown to describe how the voltage function was implemented in the software code; the analytical derivation of the paper does not use discretization. To make the paper more readable and to remove confusion, we have moved this equation and relevant discussion on discretization of LIF dynamics for software implementation to Supplementary Section A. In general, we followed the previous works of Perez-Nieves [1] and Cramer et al. [2] for software implementation.
>
>
>
> > When using $\Delta$, only the meanings of $\Delta w$  and $\Delta t$ are clearly given. Notations like $\Delta Var$ in Lemma 3.2.1 are not defined and can be confusing.
>
> We thank the reviewer for pointing this out. However, in view of the feedback from the reviewers, we have taken a different approach to prove that heterogeneity in the STDP reduces the spike count. We model the neural spike trains as a multivariate nonlinear Hawkes process model with inhibition. In this new approach, we did not use the $\Delta Var$ notation but have taken care to define all the notations used in the derivation. We are sorry for this confusion, but thanks to the reviewers' feedback, we believe the new approach leads to a more robust and complete proof.
>
>
>
> [1] Perez-Nieves, N., Leung, V.C., Dragotti, P.L. and Goodman, D.F., 2021. Neural heterogeneity promotes robust learning. Nature communications, 12(1), pp.1-9.
> [2] Cramer, B., Stradmann, Y., Schemmel, J. & Zenke, F. The heidelberg spiking datasets for the systematic evaluation of spiking neural networks. In IEEE Transactions on Neural Networks and Learning Systems 1–14 (2020).

---

> > ### Comment · Area_Chair_yr7Z · 2022-11-23
> > **Following up on the rebuttal**
> >
> > Dear reviewer,
> >
> > the authors tried to address your concerns, most notably on the lack of clarity regarding notation and the firing rate in the lemma. Are you happy with the response, or you still have doubts?
> >
> > Cheers,
> > Your AC

---

> > > ### Comment · Reviewer_Rdb2 · 2022-11-24
> > > **Post-rebuttal**
> > >
> > > The authors have made efforts to enhance the paper presentation. Based on the authors' responses, I would like to increase the score to 6.

---

> > > > ### Author Response · Authors · 2022-11-24
> > > > **Thank you.**
> > > >
> > > > Thank you so much for the positive feedback which helped us to improve the quality of our paper.

---

### Author Response · Authors · 2022-11-19
**Summary of Changes**

We thank the reviewers for their feedback and insightful comments. With regard to the reviews, the major changes in the updated paper are listed as follows:
1. We improved the writing to improve the readability and reduced notational inconsistencies and errors to the best of our abilities
2. We added further explanations for both theorems 1 and 2 to better explain the derivations and make the paper easier to read.
3. We added further discussions about the BO, including the details of the parameters used and the final distributions of the hyperparameters obtained.
4. We discussed the assumptions we made and when the theoretical analysis failed. We present the discussion in Supp Sec B.
5. We proved lemma 3.2.1 using a different approach to prove that heterogeneity in the STDP reduces the spike count. We model the neural spike trains as a multivariate nonlinear Hawkes process model with inhibition. We believe this is a much more robust proof that also addresses the main limitations of the previous proof. We would again like to thank the reviewers for pointing this out, as it leads to a more robust and complete proof.
6. We studied the generalizability of the models and added experimental results to show that heterogeneous LIF improves the model performance and heterogeneous STDP improves the model's generalizability.
7. To show the robustness of the HRSNN model, we have trained the models with limited training data. The detailed results are discussed in Supplementary Section A.
8. We have formalized the definitions of memory capacity and spike efficiency, adding specific constraints on the inputs and conditions for calculating spike count.
9. We added a subsection discussing some analytical simulation results to demonstrate the effect of heterogeneity in LIF neurons vs. memory capacity. We also demonstrate how heterogeneity in STDP dynamics improves the spike efficiency of the model.
10. We added a simplified mean-field reduction model of the HRSNN network and derived an approximate mean-field firing rate.
11. We discussed how the HRSNN network relates to similar recurrent LIF networks and balanced spiking networks used in the current literature.
12. We updated Table 2 to report the mean firing rate measured as (1/S) instead of the number of spikes.
13. We have added an intuitive explanation behind the effect of heterogeneous STDP in HRSNN.
14. We added a discussion on the limitations of the current work, which is presented in the Conclusion of the main text.


We hope we have addressed their concerns and questions regarding the paper and hope they will reconsider their rating.

---

### Decision · Program_Chairs · 2023-01-20

**Decision:**

Accept: poster

**Justification For Why Not Higher Score:**

There are significant changes that needed for the camera ready. Also, it is not entirely clear if the theoretical motivation is completely bullet-proof.

**Justification For Why Not Lower Score:**

There is a good interplay between theoretical backing and empirical justification. I found the paper an interesting and useful read.

**Metareview: Summary, Strengths And Weaknesses:**

The paper focuses on the neuronal and synaptic dynamics in spike-efficient unsupervised learning, specifically showing by theory and experiments that optimizing for heterogeneity of the time constant in the STDP learning rule using Bayesian Optimization brings significant boost to accuracy. The paper describes theorems that show that the heterogeneity increases memory capacity, decreases average spiking firing rate, and overall maximizes their ration. By this account, the authors train recurrent spiking neural networks that obtain high accuracies.

There was disagreement with reviewers, mostly on whether the theorem is indeed correctly proven or not, and whether the paper relies on Hawkes processes or not to formally prove its claims. Ultimately, after reading the paper myself, I find the theoretical backing good enough at this stage for publication, and the overall contribution of the paper to be of interest, as most reviewers are positive as well.

**Note From Pc:**

if the above contains the word "oral" or "spotlight" please see: "oral" presentation means -> notable-top-5% and "spotlight" means -> notable-top-25%. As stated in our emails, we are disassociating presentation type from AC recommendations